JCB Journal of Cell Biology

# UBAP2L ensures homeostasis of nuclear pore complexes at the intact nuclear envelope

Yongrong Liao[1,2,3,4], Leonid Andronov[2,3,4,5], Xiaotian Liu[1,2,3,4], Junyan Lin[1,2,3,4], Lucile Guerber[1,2,3,4], Linjie Lu[1,2,3,4], Arantxa Agote-Arán[1,2,3,4], Evanthia Pangou[1,2,3,4], Li Ran[1,2,3,4], Charlotte Kleiss[1,2,3,4], Mengdi Qu[1,2,3,4], Stephane Schmucker[1,2,3,4], Luca Cirillo[6,7], Zhirong Zhang[1,2,3,4], Daniel Riveline[1,2,3,4], Monica Gotta[6,7], Bruno P. Klaholz[2,3,4,5], and Izabela Sumara[1,2,3,4]

Assembly of macromolecular complexes at correct cellular sites is crucial for cell function. Nuclear pore complexes (NPCs) are large cylindrical assemblies with eightfold rotational symmetry, built through hierarchical binding of nucleoporins (Nups) forming distinct subcomplexes. Here, we uncover a role of ubiquitin-associated protein 2-like (UBAP2L) in the assembly and stability of properly organized and functional NPCs at the intact nuclear envelope (NE) in human cells. UBAP2L localizes to the nuclear pores and facilitates the formation of the Y-complex, an essential scaffold component of the NPC, and its localization to the NE. UBAP2L promotes the interaction of the Y-complex with POM121 and Nup153, the critical upstream factors in a well-defined sequential order of Nups assembly onto NE during interphase. Timely localization of the cytoplasmic Nup transport factor fragile X-related protein 1 (FXR1) to the NE and its interaction with the Y-complex are likewise dependent on UBAP2L. Thus, this NPC biogenesis mechanism integrates the cytoplasmic and the nuclear NPC assembly signals and ensures efficient nuclear transport, adaptation to nutrient stress, and cellular proliferative capacity, highlighting the importance of NPC homeostasis at the intact NE.

## Introduction

Nuclear pore complexes (NPCs) are among the most intricate multiprotein assemblies in eukaryotic cells. They are crucial for cellular function, serving as sole communication gateways between the nucleus and cytoplasm. Multiple copies of around 30 different nucleoporins (Nups) form various subcomplexes which can subsequently coassemble, following a hierarchical principle, into functional NPCs (Onischenko et al., 2020). The mature NPCs contain a scaffold that surrounds and anchors the Nups with disordered domains forming the inner passage channel (so-called phenylalanine–glycine repeat Nups or FG-Nups), as well as two asymmetric complex components, the cytoplasmic filaments facing the cytoplasmic side of the nuclear envelope (NE) and the nuclear basket pointing toward the inside of the nucleus. How these NPC architectural elements are assembled and stabilized at the intact NE represents an intriguing and unresolved biological question.

Previous studies using biochemical and high-resolution structural techniques revealed the eightfold rotational symmetry as a feature of the NPC three-dimensional organization (Beck and Hurt, 2017; Grossman et al., 2012; Hampoelz et al., 2019; Knockenhauer and Schwartz, 2016; Lin and Hoelz, 2019). The evolutionarily conserved Y-complex (also known as Nup107–160 complex) is an important component of the scaffold, forming the cytoplasmic and the nuclear rings that encompass the inner ring of the NPC (von Appen et al., 2015). In metazoans, the Y-complex is critical for NPC assembly (Doucet et al., 2010; Walther et al., 2003a) while FG-Nups can also contribute to the biogenesis of the NPC in yeast (Onischenko et al., 2017). In metazoan cells, NPCs are formed concomitantly with the reassembly of the NE during mitotic exit, and during interphase, NPCs can be formed de novo and are inserted into intact NE through an inside-out mechanism (Otsuka et al., 2016). Nup153 and POM121 are the upstream components in a well-defined sequential order of Nups assembly onto the interphase nuclei (Otsuka et al., 2016; Weberruss and Antonin, 2016).

Multiple non-nucleoporin factors have been also reported to regulate interphase NPC assembly, such as nuclear lamins

[1]Department of Development and Stem Cells, Institute of Genetics and Molecular and Cellular Biology, Illkirch, France;   [2]Centre National de la Recherche Scientifique UMR 7104, Strasbourg, France;   [3]Institut National de la Santé et de la Recherche Médicale U964, Strasbourg, France;   [4]Université de Strasbourg, Strasbourg, France;   [5]Department of Integrated Structural Biology, Centre for Integrative Biology, Institute of Genetics and Molecular and Cellular Biology, Illkirch, France;   [6]Department of Cell Physiology and Metabolism, Faculty of Medicine, University of Geneva, Geneva, Switzerland;   [7]iGE3 Institute of Genetics and Genomics of Geneva, Geneva, Switzerland.

Correspondence to Izabela Sumara: sumara@igbmc.fr

L. Andronov's current affiliation is Department of Chemistry, Stanford University, Stanford, CA, USA.   A. Agote-Arán's current affiliation is Department of Biology, Institute of Biochemistry, ETH Zürich, Zürich, Switzerland.   L. Cirillo's current affiliation is The Institute of Cancer Research, London, UK.

(Kittisopikul et al., 2021), torsin AAA+ proteins (Rampello et al., 2020), lipid saturation factors (Romanauska and Köhler, 2023), and nuclear transport receptors/regulators (Davis et al., 2022; Walther et al., 2003b; Mosammaparast and Pemberton, 2004). Interestingly, defects in DNAJB6 and Ran induce annulate lamellae (AL) in the cytoplasm (Walther et al., 2003b; Kuiper et al., 2022), which are structures containing partly assembled NPCs embedded in the endoplasmic reticulum (ER) membrane sheets, a feature associated with perturbed NPC biogenesis (Hampoelz et al., 2016). In addition to these well-established pathways, fragile X-related protein 1 (FXR1) was described to interact with Y-complex Nups in the cytoplasm and to facilitate their localization to the NE during interphase through a microtubule- and dynein-dependent mechanism, contributing to the NPC homeostasis during early interphase (Agote-Aran et al., 2020; Agote-Arán et al., 2021; Holzer and Antonin, 2020).

However, the crosstalk between different determinants of the NPC assembly during interphase, in particular between the nuclear (POM121, Nup153) and the cytoplasmic (FXR1) signals, as well as the pathways governing the formation and stability of the essential NPC subcomplexes (such as the Y-complex) at the intact NE, remained unexplored. Likewise, it is unknown what are the signaling pathways defining the eightfold-symmetrical organization of the NPC. Here, we uncover a mechanism based on UBAP2L protein by which human cells can build and stabilize functional NPCs at the NE during interphase.

## Results

### UBAP2L localizes to the NPCs and interacts with Nups and NPC assembly factors

NPC assembly during interphase is particularly active as cells grow during early G1 phase where an increase in NPC biogenesis has been observed after NE reformation (Dultz and Ellenberg, 2010; Rampello et al., 2020). The number of NPCs can be also modulated in response to cellular needs, such as differentiation or carcinogenesis when the NPC density augments dramatically (Kau et al., 2004). UBAP2L (also known as NICE-4) has been associated with various cancer types (Chai et al., 2016; He et al., 2018; Li and Huang, 2014; Ye et al., 2017; Zhao et al., 2015; Guerber et al., 2022), but the cellular mechanisms underlying its oncogenic potential remain unknown. In search of additional biological functions of UBAP2L, we analyzed its subcellular localization. Consistent with published findings (Cirillo et al., 2020; Youn et al., 2018; Huang et al., 2020; Maeda et al., 2016), endogenous UBAP2L localized to stress granules (SGs) upon exposure to stress, but a weaker UBAP2L signal was also found in the nucleus (Fig. 1, A and B) as demonstrated previously (Asano-Inami et al., 2023). In cells not treated with sodium arsenite, we observed a fraction of endogenous (Fig. 1, A and B) as well as ectopically expressed GFP- (Fig. S1 A) and Flag-UBAP2L (Fig. S1 B) to be localized at the NE during interphase. Moreover, UBAP2L accumulated in the nucleus upon treatment with the Leptomycin B (inhibitor of nuclear export factor Exportin 1), similar to MPS1 (also known as TTK), known to shuttle between the nucleus and cytoplasm (Jia et al., 2015) (Fig. S1, C–E). Cellular fractionation experiments confirmed that UBAP2L could be

found in the nucleus in interphase (Fig. S1 F), in accordance with our published findings (Guerber et al., 2023). NE localization of endogenous UBAP2L was detected in early prophase, late telophase, and in G1 cells (Fig. 1 C), suggesting a role of this protein at the sealed NE.

Because a portion of endogenous UBAP2L colocalized with the Nups detected by the monoclonal antibody mAb414, which recognizes Nup358, Nup214, Nup153, and Nup62 (hereafter named "mAb414-reactive Nups") (Fig. 1 B), we aimed to dissect the nuclear UBAP2L localization more precisely using multicolor ratiometric single-molecule localization microscopy (splitSMLM) (Andronov et al, 2021, 2022). This analysis revealed that UBAP2L can be localized at the NPCs embedded in the NE, where it was found both in the central channel labeled by Nup62 and surrounding the nuclear and cytoplasmic rings labeled by Nup96 (Fig. 1, D–F). Due to technical limitations, it was not possible to perform 3D imaging of the entire cell/nucleus and to conclude if UBAP2L could be localized to all or just a subset of NPCs. Nevertheless, quantification of the images indicated that UBAP2L is frequently localized at the side of the Nup96-positive nuclear ring (Fig. 1 F). Given that the used technique generates fluorescence images with a resolution in a 20-nm range (Andronov et al., 2022), our results suggest that UBAP2L colocalizes with several Nups and building elements of the NPCs at the NE.

To test any possible interactions of UBAP2L with Nups and NPC-assembling factors, we performed immunoprecipitations (IPs). As expected, ectopically expressed GFP-Nup85 in HeLa cells interacted with endogenous Nup133 and SEC13 (Doucet et al., 2010; Walther et al., 2003a), and with POM121 and Nup153, responsible for targeting Y-complexes to the NE (Otsuka et al., 2016; Weberruss and Antonin, 2016) and with the cytoplasmic Nup transporter FXR1 (Agote-Aran et al., 2020). GFP-Nup85 also bound endogenous UBAP2L in this analysis (Fig. 2 A). In addition, endogenous UBAP2L interacted with FXR1, FXR2, and fragile X messenger ribonucleoprotein (FMRP) (Fig. 2 B) as previously shown (Huang et al., 2020; Marmor-Kollet et al., 2020; Sanders et al., 2020) and with some mAb414-reactive FG-Nups that are known to contribute to the biogenesis of the NPC in yeast (Onischenko et al., 2017) (Fig. 2 B). Finally, ectopically expressed GFP-FXR1 interacted with Y-complex Nups and with UBAP2L (Fig. 2 C). Taken together, the interaction of UBAP2L with Nups and NPC assembly factors suggests a possible function of UBAP2L on NPC assembly and/or stability.

### UBAP2L regulates Nups localization

To understand if UBAP2L regulates Nups, we used two clonal HeLa cell lines with CRISPR/Cas9-mediated deletion of the *UBAP2L* gene, which we previously characterized (Guerber et al., 2023). As expected (Cirillo et al., 2020; Huang et al., 2020; Youn et al., 2018), deletion of UBAP2L inhibited formation of SGs upon stress (Fig. S1 G) and abolished nuclear localization of endogenous UBAP2L (Fig. S1 H), confirming the specificity of UBAP2L antibodies. Relative to isogenic control cell line (wild type [WT]), both UBAP2L knock-out (KO) cell lines revealed accumulation of foci containing Nups (Y-complex Nup133, cytoplasmic filaments

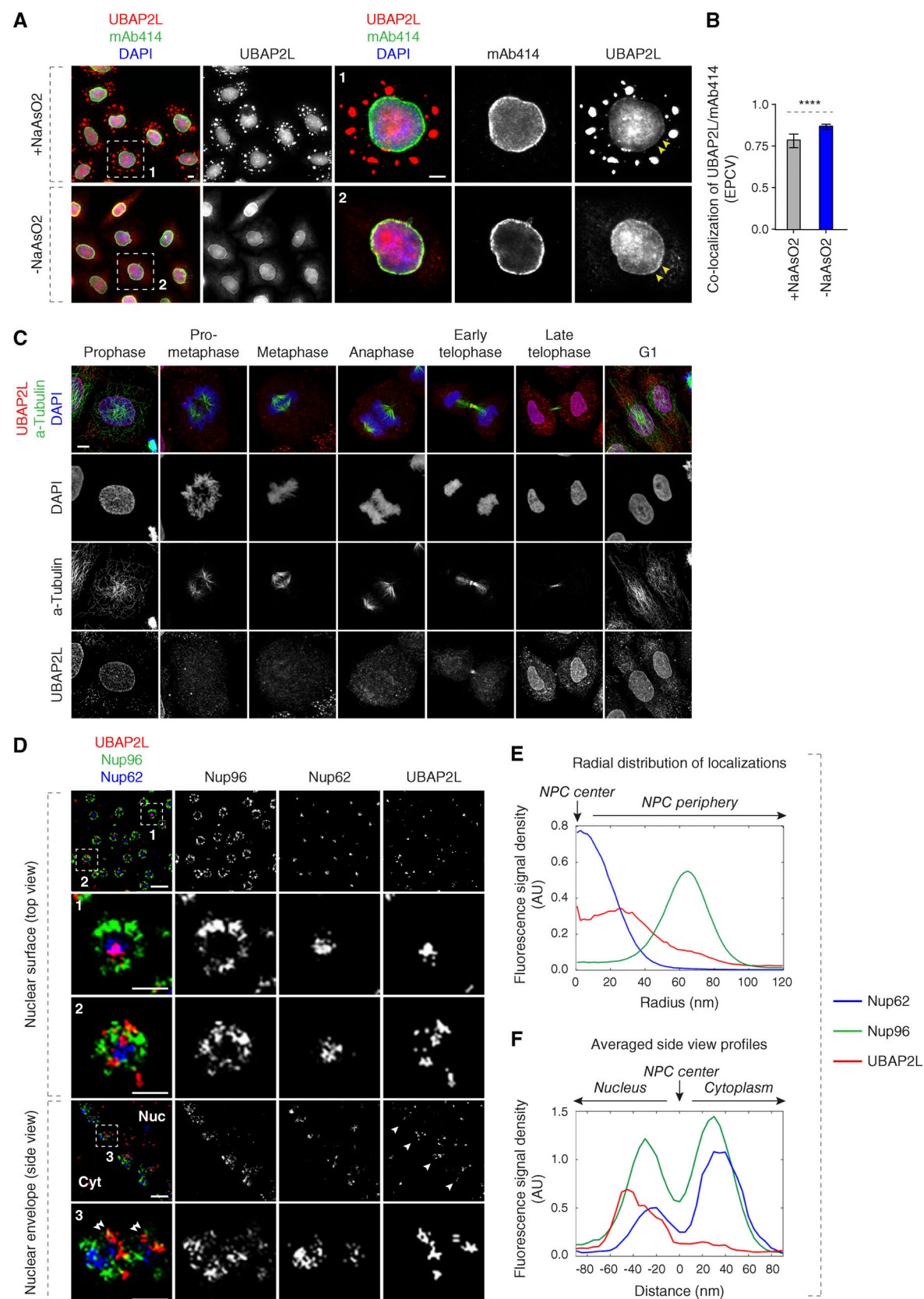

Figure 1.  **UBAP2L localizes to the NE and NPCs. (A and B)** Representative images of the localization of UBAP2L and Nups in HeLa cells with/without NaAsO2 treatment shown by immunofluorescence microscopy with UBAP2L and mAb414 antibodies. Nuclei were stained with DAPI. The arrowheads indicate the NE localization of endogenous UBAP2L. The magnified framed regions are shown in the corresponding numbered panels. Scale bars, 5 µm (A). The co-localization (EPCV, events per cell view) of UBAP2L and mAb414 in A was measured by CellProfiler (mean ± SD, ****P < 0.0001, unpaired two-tailed t test; 175

cells for NaAsO$_2$ treatment and 110 cells without NaAsO$_2$ treatment were counted) (B). **(C)** Representative immunofluorescence images depicting the localization of UBAP2L in HeLa cells after chemical pre-extraction of the cytoplasm using 0.01% of Triton X-100 for 90 s in indicated cell cycle stages and visualized by UBAP2L antibody. Nuclei and chromosomes were stained with DAPI. Scale bar, 5 μm. **(D–F)** Representative super-resolution immunofluorescence images of Nup96-GFP KI U2OS cells acquired using multicolor SMLM with a dichroic image splitter (splitSMLM) show NPCs on the nuclear surface (top view) and in the cross-section of the NE (side view). Nup96 signal labels the cytoplasmic and nuclear ring of the NPC and the localization of the central channel NPC component is analyzed by Nup62 antibody. The nuclear (Nuc) and cytoplasmic (Cyt) sides of the NE are indicated in the side view. The magnified framed regions are shown in the corresponding numbered panels. Note that UBAP2L can localize to both structures within the NPCs (framed regions 1 and 2 in the top view) and is found preferentially at the nuclear ring labeled with Nup96 (double arrowheads in framed region 3 in the side view). Scale bars, 300 and 100 nm, respectively (D). Radial distribution of localizations of Nup62, Nup96, and UBAP2L in D was obtained by averaging 1932 NPC particles (E). Averaged "side view" profiles of Nup62, Nup96, and UBAP2L in D were obtained by alignment of 83 individual NPCs (F). Orientation bars point to the NPC center (central channel middle point) as well as the cytoplasmic and nuclear sides (E and F).

RanBP2, mAb414-reactive Nups, and inner ring Nup205) as well as Importin-β and Exportin-1 in the cytoplasm, but did not show defects in the localization of the NPC basket component Nup153 (Fig. 3, A–F). UBAP2L KO cells also displayed cytoplasmic granules containing both Importin-β and Nup133, and likewise, RanBP2-containing granules colocalized with mAb414-reactive

Figure 2. **UBAP2L interacts with Nups and NPC assembly factors. (A)** HeLa cells lysates expressing GFP alone or 3XGFP-Nup85 for 27 h were immunoprecipitated using agarose GFP-Trap A beads (GFP-IP), analyzed by western blot, and signal intensities were quantified (shown a mean value, **P < 0.01, ***P < 0.001, unpaired two-tailed *t* test; *n* = 3 independent experiments). Molecular weight markers are indicated in kilodalton (kDa). Please note that in all shown experiments, a specific band corresponding to Nup153 and recognized by both Nup153 and mAb414 antibodies displayed an atypical migration pattern of around 250 kDa size, probably due to usage of Tris-acetate gradient gels (Materials and methods section). **(B)** HeLa cells lysates were immunoprecipitated using UBAP2L antibody or unspecific rabbit IgG, analyzed by western blot, and signal intensities were quantified (shown a mean value, **P < 0.01, ***P < 0.001, unpaired two-tailed *t* test; *n* = 3 independent experiments). The arrow indicates the band corresponding to the IgG heavy chain (HC). **(C)** Lysates of HeLa cells expressing GFP alone or GFP-FXR1 for 27 h were immunoprecipitated using agarose GFP-Trap A beads (GFP-IP), analyzed by western blot, and signal intensities were quantified (shown a mean value, *P < 0.05, **P < 0.01, unpaired two-tailed *t* test; *n* = 3 independent experiments). Source data are available for this figure: SourceData F2.

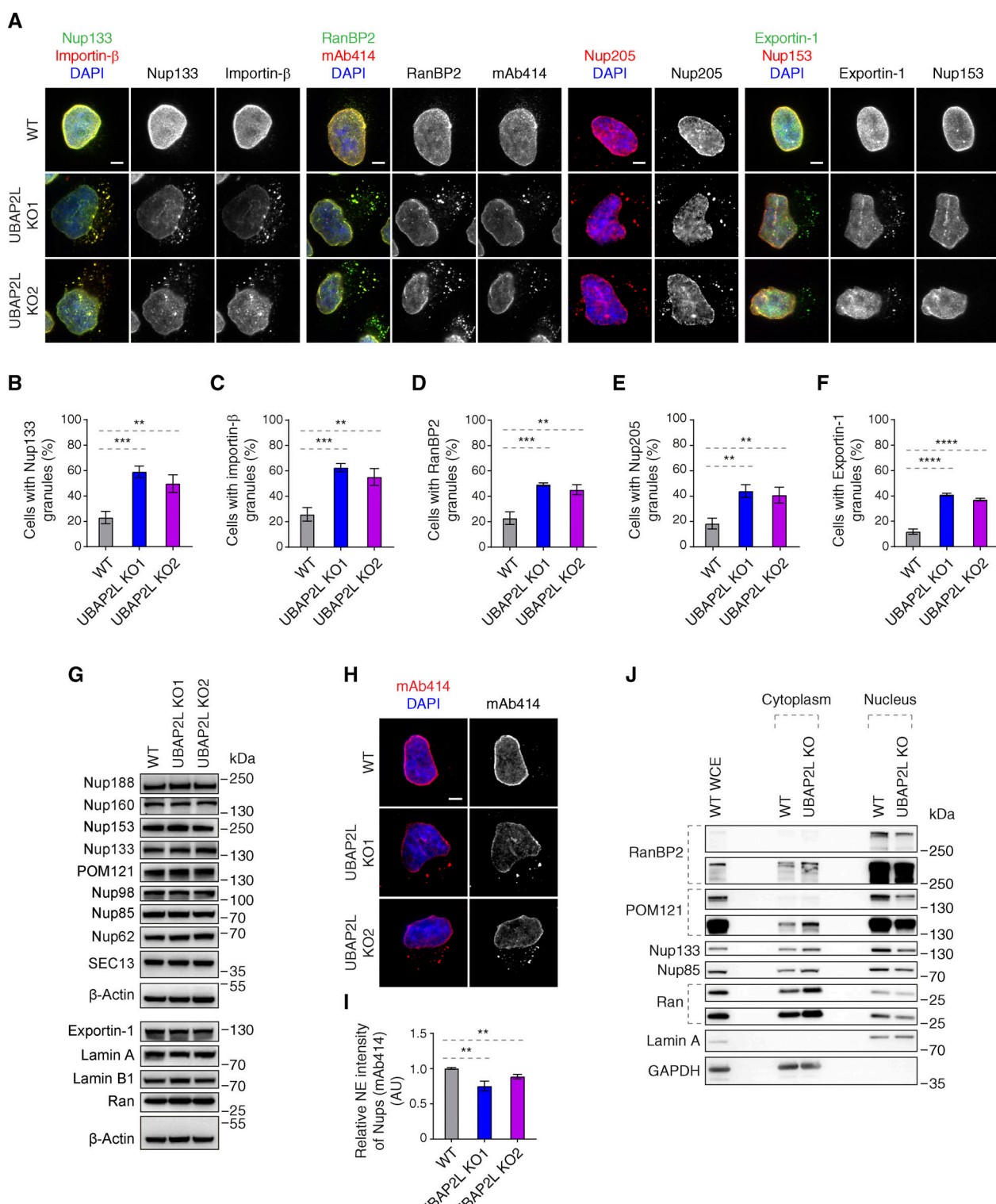

Figure 3. **UBAP2L regulates Nups localization. (A–F)** Representative immunofluorescence images depicting the localization of Nups and NPC-associated factors in WT and UBAP2L KO HeLa cells synchronized in interphase by DTBR at 12 h. Nuclei were stained with DAPI. Scale bars, 5 μm (A). The percentage of cells with the cytoplasmic granules containing Nup133 (B), Importin-β (C), RanBP2 (D), Nup205 (E), and Exportin-1 (F) in A were quantified. At least 100 cells per condition were analyzed (mean ± SD, **P < 0.01, ***P < 0.001, ****P < 0.0001, unpaired two-tailed $t$ test, $n$ = 3 independent experiments). **(G)** The protein levels of Nups and NPC-associated factors in WT and UBAP2L KO HeLa cells synchronized in interphase by DTBR at 12 h were analyzed by western blot. **(H and I)** Representative immunofluorescence images of FG-Nups (mAb414) at the NE in WT and UBAP2L KO HeLa cells in interphase cells synchronized by DTBR at 12 h. Nuclei were stained with DAPI. Scale bar, 5 μm (H). The NE intensity of Nups (mAb414) in H was quantified (I). At least 150 cells per condition were analyzed (mean ± SD, **P < 0.01, unpaired two-tailed $t$ test, $n$ = 3 independent experiments). **(J)** The nuclear and cytoplasmic protein levels of Nups and NPC transport-associated factors in WT and UBAP2L KO HeLa cells synchronized in the G1/S transition phase by thymidine 18 h were analyzed by western blot. WCE indicates whole cell extract. Source data are available for this figure: SourceData F3.

Nups (Fig. 3 A). This phenotype strongly resembles the defects observed upon downregulation of the factors required for the assembly of NPCs at the NE such as FXR1 (Agote-Aran et al., 2020). We were unable to detect any changes in protein levels of several Nups as well as in Exportin-1 and Lamin A and B1 (Fig. 3 G) in the whole-cell extracts, but deletion of UBAP2L led to reduced NE intensity of mAb414-reactive Nups (Fig. 3, H and I). Fractionation experiments confirmed moderately reduced levels of Nups in the nucleus and an increased pool of cytoplasmic Nups upon deletion of UBAP2L (Fig. 3 J), suggesting that UBAP2L does not regulate total protein levels but rather Nups localization during interphase. Owing to the fact that UBAP2L deletion can delay mitotic exit (Guerber et al., 2023), which could theoretically influence the length of the G1 phase and, indirectly, the localization of Nups, we have arrested UBAP2L KO cells in G1 using lovastatin (Rao et al., 1999) and observed accumulation of cytoplasmic Nup foci and reduced NE intensity of mAb414-reactive Nups (Fig. S1, I–K), but no changes in nuclear size (Fig. S1 L). The same results were obtained in G0/G1-arrested cells using Psoralidin (Gulappa et al., 2013) (Fig. S1, M–P). Lovastatin led to a decrease in nuclear size (Fig. S1 L), as previously demonstrated (Iida et al., 2022), relative to Psoralidin (Fig. S1 P) and untreated HeLa cells (Fig. S3 E) but no significant differences could be detected between WT and UBAP2L KO cells upon both treatments and under untreated conditions (Fig. S1, L and P; and Fig. S3 E), which is in agreement with our published findings (Guerber et al., 2023). These results suggest that UBAP2L regulates Nups localization and possibly their assembly without affecting the size of the nucleus or the length of the G1 phase.

## UBAP2L regulates localization of Nups in interphase but not during mitotic exit

Two distinct pathways of NPC assembly at the NE have been described in higher eukaryotic cells (Weberruss and Antonin, 2016). In the postmitotic pathway, NPC assembly occurs on segregated chromosomes, while during interphase, both Nup153 and POM121 drive de novo assembly of NPCs into an enclosed NE (D'Angelo et al., 2006; Doucet et al., 2010; Vollmer et al., 2015), supported by several non-Nup assembly factors (Dultz et al., 2022). The interphase pathway can be facilitated by FXR1 and microtubule-dependent transport of cytoplasmic Nups toward NE (Agote-Aran et al., 2020; Agote-Arán et al., 2021; Holzer and Antonin, 2020). Given the strong interaction of UBAP2L with FXR1 (Fig. 2, B and C), we hypothesized that UBAP2L may selectively affect Nups assembly during interphase. Indeed, accumulation of cytosolic Nup foci could be first observed during late telophase, early G1 as well as in phospho-Rb–positive cells (mid-late G1, S, and G2 phases), but not during anaphase and early telophase stages (Fig. 4, A–F). mAb414-reactive Nups assembled normally on segregating chromosomes in anaphase and on decondensing chromatin during early telophase (Fig. 4 G) upon deletion of UBAP2L, but reduced NE levels of Nups were observed in early G1 and in phospho-Rb–positive cells in the absence of UBAP2L (Fig. 4, H and I). The percentage of cells in mid-late G1, S, and G2 phases was not affected by UBAP2L deletion (Fig. 4 J), further suggesting that progression through interphase occurred normally in UBAP2L KO cells. We conclude that UBAP2L drives Nups localization to NE during interphase but not in cells exiting mitosis.

## UBAP2L may facilitate the assembly of the NPC scaffold elements and the biogenesis of NPCs

Our data demonstrate that UBAP2L deletion leads to decreased Nup levels at the NE and to the formation of Nup foci in the cytoplasm. However, can UBAP2L also regulate the assembly of functional NPCs at the NE? The splitSMLM analysis revealed that deletion of UBAP2L decreased the density of the NPCs at the NE (Fig. 5, A and B) and confirmed the presence of cytoplasmic assemblies containing RanBP2 and mAb414-reactive Nups (Fig. S2 A), which often (depending on the optical view) displayed linear-like organization with symmetrical RanBP2 distribution (Fig. S2 A), contrary to non-symmetrical localization at the cytoplasmic side of the NE (Fig. 5 A). These cytosolic assemblies also contained preassembled NPCs with Nup133-positive rings surrounding the central channel labeled by Nup62 (Fig. S2 A), suggesting that they may represent AL-like structures. Overexpression of Flag-UBAP2L in interphase HeLa cells was sufficient to moderately increase the density of NPCs at the NE (Fig. S2 B and Fig. 5 C), suggesting that UBAP2L might be required for NPC biogenesis onto intact NE. Flag-UBAP2L also occasionally colocalized with the cytoplasmic assemblies of mAb414-reactive Nups (Fig. S2, B and C). The alignment and segmentation analysis of Nup133 further suggested that the organization of the NE-localized NPCs was slightly altered upon deletion of UBAP2L (Fig. 5, A and D) where a moderately reduced number of NPCs structures with an eightfold symmetrical organization was detected (Fig. 5, A and D). At present, it cannot be formally excluded that observed differences are the result of insufficient labeling, and future ultrastructural approaches will be required to formally address the regulation of NPC symmetry by UBAP2L.

Two clonal U2OS cell lines with CRISPR/Cas9-mediated deletion of UBAP2L gene with stably integrated Nup96-GFP (Nup96-GFP knock-in [KI]) (Fig. S2, D and E) likewise showed the accumulation of cytoplasmic Nup foci (Nup96-GFP and mAb414-reactive Nups) (Fig. S2, F–H) and reduced density of the NPCs at the NE (Fig. 5, E and F).

Moreover, deletion of UBAP2L in HeLa cells reduced the interaction of GFP-Nup85 with other components of the Y-complex, Nup133, and SEC13 in both unsynchronized (Fig. 5 G) and G1/S-synchronized cells (Fig. S2 I) as well as decreased the interaction of GFP-Nup85 with the two Nups, Nup153 and POM121 (Fig. 5 G), involved in the assembly of the NPCs at the enclosed NE (Funakoshi et al., 2011; Vollmer et al., 2015). IP of endogenous Nup96 from Nup96-GFP KI U2OS cells also demonstrated reduced interaction of Y-complex components Nup85 and SEC13 and inhibition of Nup96-GFP binding to Nup153 and POM121 in the absence of UBAP2L (Fig. 5 H). Interestingly, the interaction of endogenous Nup85 with other components of the Y-complex appeared moderately increased in G1/S cells relative to cells arrested in prometaphase using Eg5 inhibitor S-trityl-L-cysteine (STLC) (Fig. S2 J), suggesting that Y-complex assembly may also take place during interphase. In addition, the interaction of FXR1 with both GFP-Nup85 and

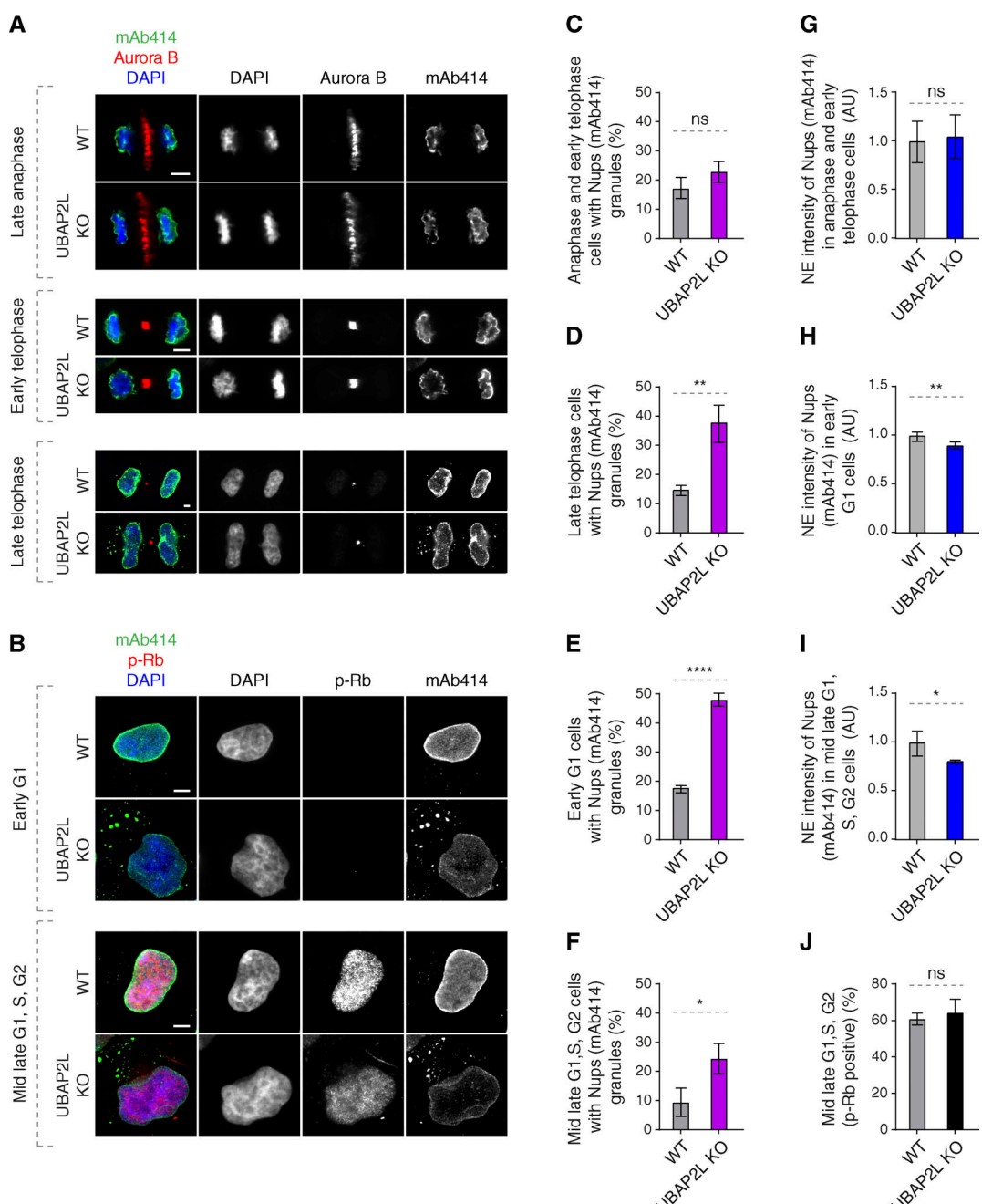

Figure 4. **UBAP2L regulates localization of Nups in interphase but not during mitotic exit. (A and B)** Representative immunofluorescence images depicting the localization of Nups (mAb414) in WT and UBAP2L KO HeLa cells in different cell cycle stages. Mitotic cells were labeled by Aurora B (A) while p-Rb was used to distinguish between early G1 (p-Rb–negative cells) and mid-late G1, S, and G2 (p-Rb–positive cells) stages (B). Nuclei and chromosomes were stained with DAPI. Scale bars, 5 μm. **(C–F)** The percentage of cells with the cytoplasmic granules of Nups (mAb414) in anaphase and early telophase (C), late telophase (D), early G1 (E), and mid-late G1, S, G2 (F) in A and B were quantified. At least 150 cells per condition were analyzed (mean ± SD, ns: non-significant, *P < 0.05, **P < 0.01, ****P < 0.0001, unpaired two-tailed $t$ test, $n$ = 3 independent experiments). **(G–I)** The NE intensity of Nups (mAb414) in anaphase and early telophase cells (G), early G1 cells (H), and mid-late G1, S, G2 cells (I) in A and B were quantified. At least 100 cells per condition were analyzed (mean ± SD, ns: non-significant, *P < 0.05, **P < 0.01, unpaired two-tailed $t$ test, $n$ = 3 independent experiments). **(J)** The percentage of p-Rb–positive cells in B was quantified. At least 150 cells per condition were analyzed (mean ± SD, ns: non-significant, unpaired two-tailed $t$ test, $n$ = 3 independent experiments).

Nup96-GFP was reduced in the absence of UBAP2L (Fig. 5, G and H), and UBAP2L deletion inhibited the binding of GFP-FXR1 to Nup85, SEC13, and with the components of the dynein complex, dynactin p150$^{Glued}$ and BICD2 (Fig. S2 K), that work with FXR1 to transport Nups along microtubules toward NE

during interphase (Agote-Aran et al., 2020). These results demonstrate that UBAP2L might be involved in the biogenesis (or stability) of NPCs at the NE during interphase possibly by facilitating the assembly of the Y-complex and its interaction with both nuclear (Nup153, POM121) as well as

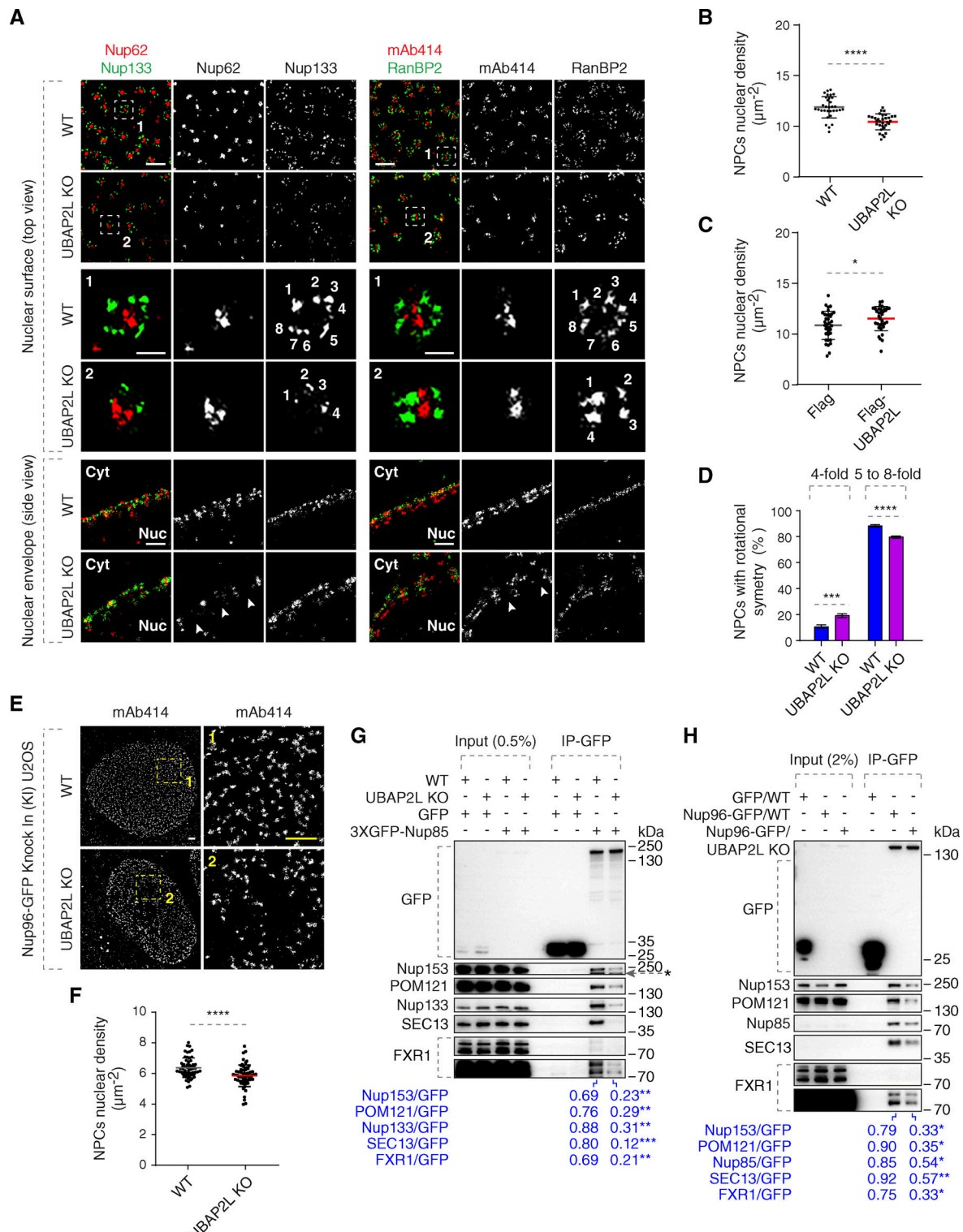

**Figure 5. UBAP2L may facilitate the assembly of the NPC scaffold elements and the biogenesis of NPCs. (A)** Representative splitSMLM images depicting several NPC components on the nuclear surface (top view) and in the cross-section of the NE (side view) in WT and UBAP2L KO HeLa cells synchronized in early interphase by DTBR at 12 h. Nup133 signal labels the cytoplasmic and nuclear rings of the NPC, the localization of the central channel is visualized by Nup62 and mAb414 antibodies, and the cytoplasmic filaments are labeled by RanBP2. The magnified framed regions are shown in the corresponding numbered panels. The nuclear (Nuc) and cytoplasmic (Cyt) side of the NE are indicated in the side view. The arrowheads indicate the disrupted localization of Nup62 or mAb414 at NE in UBAP2L KO HeLa cells and the numbers point to the individual identified spokes of the NPC. Scale bars, 300 and 100 nm, respectively. **(B and C)** The nuclear density of NPCs (mAb414 and RanBP2) in cells shown in A was quantified (B) (mean ± SD, ****P < 0.0001, unpaired two-tailed *t* test; counted 32 cells per cell line). The nuclear density of NPCs (mAb414) in HeLa cells expressing Flag alone or Flag-UBAP2L for 35 h and synchronized in interphase by DTBR at 12 h was quantified (C) (mean ± SD, *P < 0.05, unpaired two-tailed *t* test; counted 35 cells for Flag and 32 cells for Flag-UBAP2L). The corresponding

representative images are shown in Fig. S2 B. **(D)** The rotational symmetry of NPCs in A was quantified by alignment of Nup133 particles and segmentation analysis (mean ± SD, ***P < 0.001, ****P < 0.0001, unpaired two-tailed *t* test; counted 851 NPCs for WT HeLa cell line and 559 NPCs for UBAP2L KO HeLa cell line). **(E and F)** Representative SMLM immunofluorescence images of FG-Nups (mAb414) at the nuclear surface in Nup96-GFP KI U2OS WT and UBAP2L KO cells in interphase cells synchronized by DTBR at 12 h. Scale bars, 1 µm (E). The nuclear density of NPCs in cells shown in E was quantified in F (mean ± SD, ****P < 0.0001, unpaired two-tailed *t* test; counted 60 cells per cell line). **(G)** Lysates of interphase WT and UBAP2L KO HeLa cells expressing GFP alone or 3XGFP-Nup85 for 27 h were immunoprecipitated using agarose GFP-Trap A beads (GFP-IP), analyzed by western blot, and signal intensities were quantified (shown a mean value, **P < 0.01, ***P < 0.001, unpaired two-tailed *t* test; *n* = 3 independent experiments). The asterisk indicates a non-specific, faster migrating band. **(H)** Lysates of interphase U2OS cells expressing GFP alone for 27 h and Nup96-GFP KI U2OS WT and UBAP2L KO cells were immunoprecipitated using agarose GFP-Trap A beads (GFP-IP), analyzed by western blot, and signal intensities were quantified (shown a mean value, *P < 0.05, **P < 0.01, unpaired two-tailed *t* test; *n* = 3 independent experiments) (H). Source data are available for this figure: SourceData F5.

with cytoplasmic (FXR1, dynein complex) NPC assembly signals.

## UBAP2L regulates localization of the Nup transporting factor FXR1

How can UBAP2L help fuel the assembly of cytoplasmic Nups onto NE? The cellular phenotypes on Nups upon deletion of UBAP2L strongly resemble downregulation of FXR1, which drives transport of the cytoplasmic Nups to the NE during early interphase (Agote-Aran et al., 2020; Agote-Arán et al., 2021). The fact that UBAP2L not only facilitated the interaction of the Y-complex with Nup153 and POM121 but also with FXR1 and the dynein complex (Fig. 5, G and H; and Fig. S2 K) and that FXRPs strongly interacted with UBAP2L (Fig. 2, B and C) prompted us to analyze the dynamics of FXRPs in more detail. In contrast to WT cells where FXR1 was localized at the NE and diffusely in the cytoplasm, as reported previously (Agote-Aran et al., 2020), both UBAP2L KO cell lines displayed reduced NE localization of FXR1 and formation of cytoplasmic FXR1-containing granules (Fig. S3, A and C) in addition to mAb414-reactive Nups foci, which did not co-localize with FXR1 in the cytoplasm (Fig. S3, A and B). Both UBAP2L KO cell lines also showed irregular nuclear shape (Fig. S3 D), but no changes in the nuclear size (Fig. S3 E) could be observed, in agreement with our previous findings (Guerber et al., 2023). Deletion of UBAP2L moderately reduced levels of FXR1 in the nuclear fractions of both G1-synchronized (Fig. S3 F) and unsynchronized interphase cells (Fig. S3 G), similar to Nups and the nuclear transport factor Ran (Fig. S3, F and G). The same phenotype was observed for FMRP (Fig. S3, H and I), but UBAP2L deletion did not affect protein levels of any of the three FXRPs (Fig. S3 J). Downregulation of UBAP2L using specific siRNAs confirmed the cellular phenotypes of UBAP2L KO cells and displayed accumulation of FXR1 foci, cytoplasmic Nups-containing granules, and irregular nuclear shape as also observed upon depletion of FXR1 and in contrast to control cells (Fig. S3, L–N). These results suggest that FXR1 cytoplasmic granules are not the result of any possible compensation effects due to stable deletion of UBAP2L in KO cells. Since UBAP2L was previously demonstrated to contribute to the assembly of SGs (Cirillo et al., 2020; Huang et al., 2020; Youn et al., 2018) and FXRPs and Nups are able to localize to these structures (Huang et al., 2020; Zhang et al., 2018), we studied if observed phenotypes could be linked to cellular stress signaling. As expected (Cirillo et al., 2020; Huang et al., 2020; Youn et al., 2018), deletion of UBAP2L inhibited formation of SGs (Fig. S1 G) upon stress but the SG components G3BP1 and TIA-1 did not localize to

FXR1-containing granules under normal growing conditions in UBAP2L KO cells (Fig. S3, O and P), suggesting that FXR1 foci are distinct from SGs and that UBAP2L-mediated regulation of Nups might be independent of UBAP2L's function on SGs. Importantly, UBAP2L not only facilitates the interaction of FXRPs with the scaffold Nups but also helps to localize FXRPs to the NE, thereby fueling the assembly of Nups from the cytoplasm to the nucleus.

## Arginines within the RGG domain of UBAP2L mediate the function of UBAP2L on Nups and FXRPs

To dissect the molecular basis of the UBAP2L-FXR1-Nup pathway and to understand if the function of UBAP2L on cytoplasmic Nups and FXRPs is specific, we performed rescue experiments. In contrast to GFP, ectopic expression of GFP-UBAP2L efficiently rescued Nup and FXR1 granules as well as the irregular nuclei phenotypes in both UBAP2L KO cell lines (Fig. S4, A–E). GFP-UBAP2L protein fragment encompassing 98–430 aa was required (Fig. S4, F and G) and sufficient (Fig. S4 H) for the interaction with FXR1 in the IP experiments. Interestingly, this fragment contains the arginine-glycine-glycine repeat (RGG) domain (Fig. S4 F), which often engages in interactions with mRNAs and mediates UBAP2L's function in protein translation and RNA stability (Luo et al., 2020). Surprisingly, GFP-tagged UBAP2L (Fig. S4 I) and endogenous UBAP2L (Fig. S4 J) interacted with endogenous FXR1 and FMRP despite the absence of RNAs after RNase A treatment, suggesting that the role of UBAP2L on FXRPs-Nups pathway may be, to a large extent, RNA independent. The arginines present in the RGG domains were previously demonstrated to regulate localization of other proteins also in an RNA-independent manner (Thandapani et al., 2013) and to be asymmetrically dimethylated (ADMA) by the protein arginine methyltranferase PRMT1 (Huang et al., 2020; Maeda et al., 2016). Indeed, the mutant form of UBAP2L, where all 19 arginines were exchanged to alanines (UBAP2L R131–190A), did not interact with endogenous PRMT1 and showed reduced ADMA signal as expected (Fig. 6 A). The R131–190A mutation also reduced the interaction of UBAP2L with Nups and FXR1 (Fig. 6 A), suggesting the role of arginines within the RGG domain of UBAP2L in Nups assembly. The GFP-UBAP2L protein fragment encompassing 98–430 aa could rescue localization defects of Nups and FXR1 in UBAP2L KO cells, in a manner similar to the full-length UBAP2L protein (Fig. S4, K–N) but the UBAP2L R131–190A mutant was unable to restore the FXR1 and Nups localization defects and irregular nuclear shape in UBAP2L KO cells (Fig. 6, B–E). We conclude that the function of UBAP2L on the regulation of FXRPs

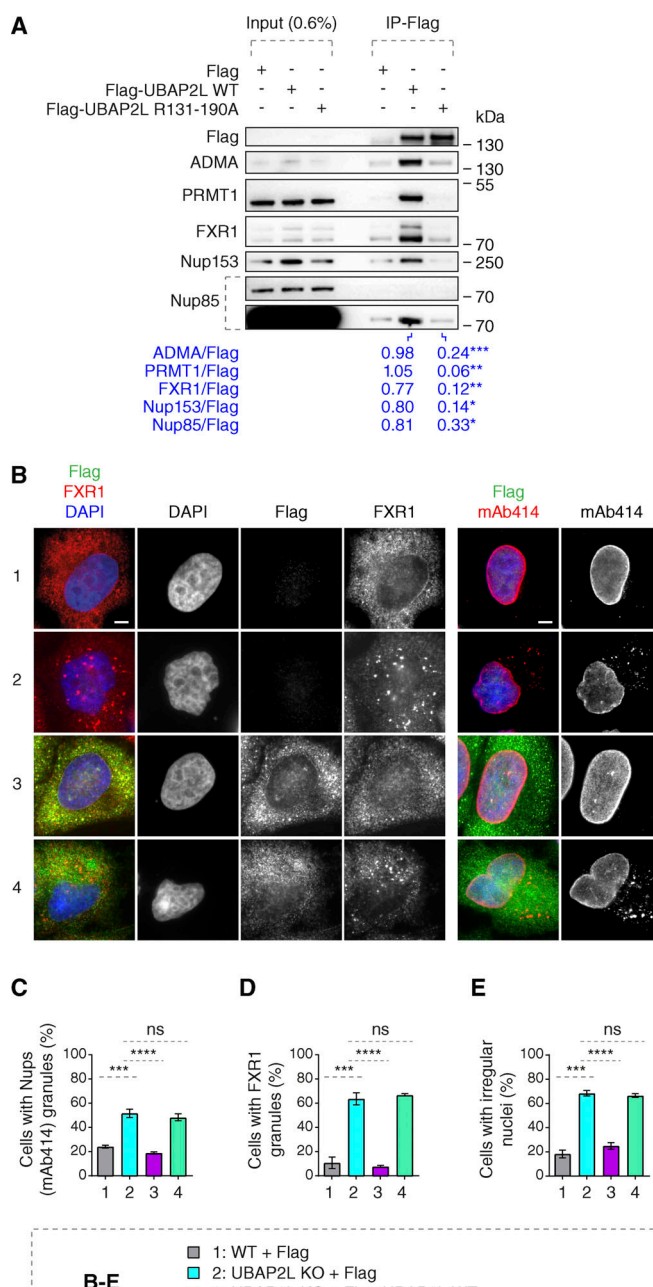

**Figure 6. Arginines within the RGG domain of UBAP2L mediate the function of UBAP2L on Nups and FXRPs. (A)** Lysates of interphase HeLa cells expressing Flag alone, Flag-UBAP2L WT, or mutated Flag-UBAP2L version where 19 arginines located in the RGG domain were replaced by alanines (R131–190A) for 27 h were immunoprecipitated using Flag beads (Flag-IP), analyzed by western blot, and signal intensities were quantified (shown a mean value, *P < 0.05, **P < 0.01, ***P < 0.001, unpaired two-tailed *t* test; *n* = 3 independent experiments). **(B–E)** Representative immunofluorescence images depicting nuclear shape and localization of FXR1 and Nups (mAb414) in WT and UBAP2L KO HeLa cells expressing Flag alone or Flag-UBAP2L (WT or R131–190A) for 60 h and synchronized in interphase by DTBR at 12 h. Nuclei were stained with DAPI. Scale bars, 5 µm (B). The percentage of cells with the cytoplasmic granules of Nups (mAb414) (C) and of FXR1 (D) and irregular nuclei (E) shown in B were quantified. At least 200 cells per condition were analyzed (mean ± SD, ns: not significant, ***P < 0.001, ****P < 0.0001, unpaired two-tailed *t* test, *n* = 3 independent experiments). Source data are available for this figure: SourceData F6.

and Nups is mediated through the arginines present within the RGG domain.

## UBAP2L regulates localization of FXR1 to the NE

How can the function of UBAP2L on Nups be linked to the observed subcellular localization of FXR1? And how and when can FXRPs form cytoplasmic assemblies in the absence of UBAP2L? Although UBAP2L regulates mitotic exit (Guerber et al., 2023; Maeda et al., 2016), the Nups localization defects could be also observed in UBAP2L KO cells arrested in G1 (Fig. S1, I–P). In addition, inhibition of Polo-like kinase 1 (PLK1) activity, the downstream target of UBAP2L during mitosis, was reported to rescue the mitotic defects observed in the absence of UBAP2L (Guerber et al., 2023) but it could not reverse the Nup defects in the same experimental setting (Fig. S5, A and B), arguing that UBAP2L-dependent regulation of Nups could be largely uncoupled from its role in mitotic progression.

Importantly, the increased numbers of FXR1-containing foci were also observed in UBAP2L KO late telophase cells when compared with the corresponding WT cells synchronized in the same cell cycle stage (Fig. S5, C and D). The average size of the FXR1-containing granules was likewise increased in late telophase synchronized UBAP2L KO relative to WT cells (0.346 and 0.218 µm², respectively) (Fig. S5, C and E). Reduced NE localization of FXR1 and formation of cytoplasmic granules were observed in early and mid-late G1, S, and G2 phases in UBAP2L KO relative to WT cells (Fig. S5, F–I). In addition, endogenous UBAP2L could interact with endogenous FXR1 and FMRP in asynchronous cells as well as in cells synchronized during mitosis and in interphase (Fig. S5 J). Interestingly, the effect of UBAP2L deletion on the percentage of FXR1 granules-containing cells, the number of granules per cell, and the size of FXR1 granules were the most evident in early G1 compared with other cell cycle stages analyzed (Fig. S5, F–I), in line with our findings suggesting that UBAP2L preferentially regulates Nups localization to NE during early G1 (Fig. 4, A–I). The fact that FXR1-containing granules are also observed in the WT late telophase cells, although to a lesser extent as compared with UBAP2L KO cells (Fig. S5, C–E), suggests that they do not form de novo upon deletion of UBAP2L but may originate from some similar assemblies existing before mitotic exit.

For this reason, we analyzed FXR1 and FMRP localization during mitosis in cells synchronized in prometaphase-like stage using Nocodazole or Eg5 inhibitor STLC where strong accumulation of granules containing both FXR1 and FMRP was observed (Fig. 7 A). Time-lapse analysis using live video spinning disk microscopy of cells expressing GFP-FXR1 revealed its dynamic localization during mitotic progression and confirmed the presence of GFP-FXR1–containing granules in mitotic cells (Fig. 7, B–D) starting from late prophase and throughout prometaphase, metaphase, and anaphase stages. Interestingly, unlike in control cells where GFP-FXR1 mitotic granules spread out in the vicinity of the NE concomitant with the nuclei reformation during mitotic exit, in UBAP2L-deleted cells, these granules remained in the cytoplasm, surrounding the nucleus and GFP-FXR1 localization at the NE appeared to be reduced (Fig. 7, B–D). Accordingly, both the number as well as the average size of

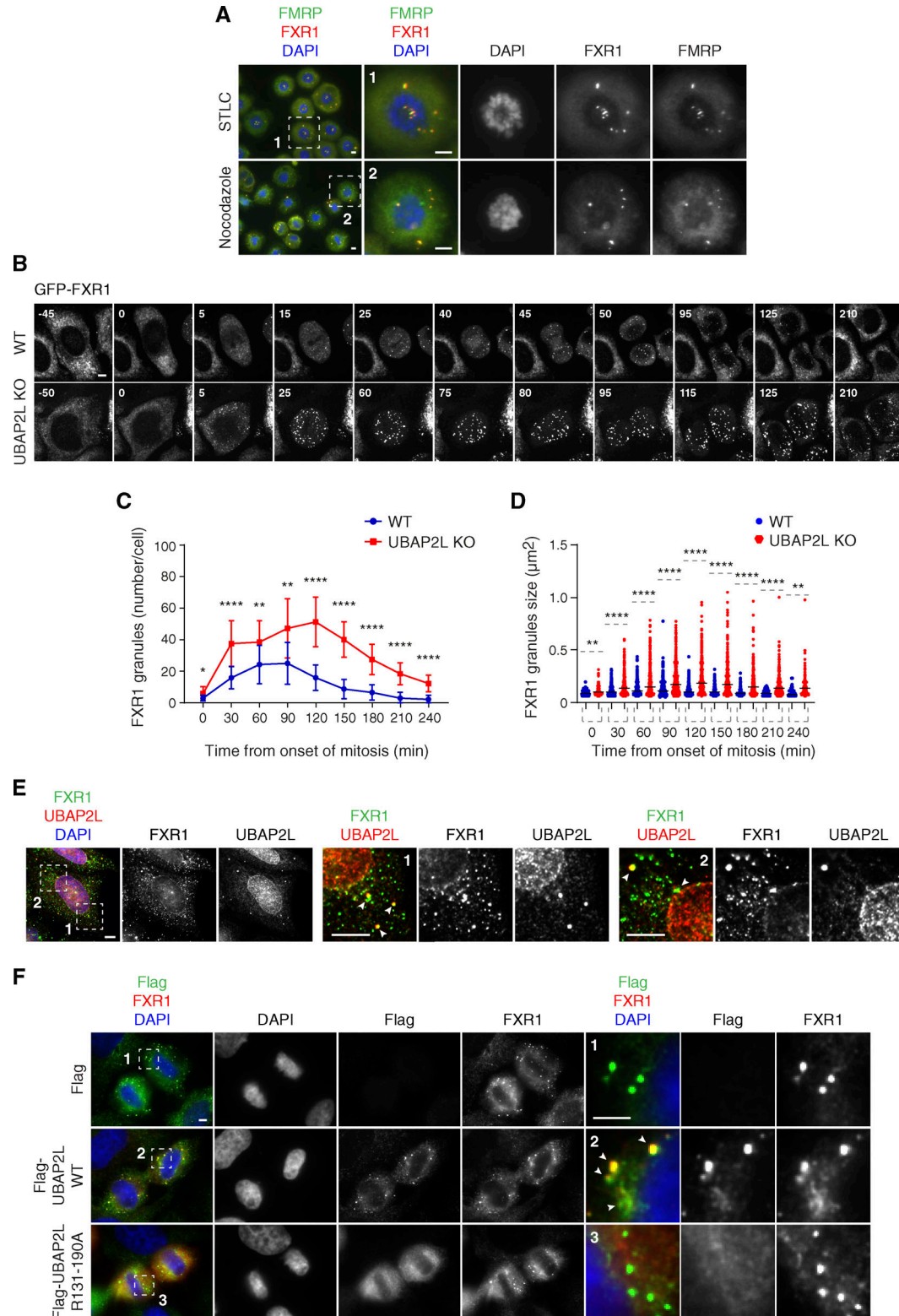

Figure 7. **UBAP2L remodels FXR1-protein assemblies in the cytoplasm and drives localization of FXR1 to the NE. (A)** Representative immunofluorescence images depicting the localization of FXR1 and FMRP in HeLa cells synchronized in prometaphase using STCL 16 h or nocodazole 16 h. Chromosomes were stained with DAPI. Scale bars, 5 μm. **(B–D)** WT and UBAP2L KO HeLa cells expressing GFP-FXR1 were synchronized by DTBR and analyzed by live video spinning disk confocal microscopy. The selected representative frames of the movies are depicted, and time is shown in minutes. Timepoint 0 indicates mitotic entry during prophase. Scale bar, 5 μm (B). GFP-FXR1 granules number (number/cell) shown in B at indicated times during mitotic progression were quantified (C). GFP-FXR1 granules sizes (granule ≥ 0.061 μm²) shown in B at indicated timepoints during mitotic progression were quantified (D). 16 WT and 11 UBAP2L KO HeLa cells were counted in C and D, respectively (mean ± SD, *P < 0.05, **P < 0.01, ****P < 0.0001, unpaired two-tailed *t* test ). **(E)** Representative immunofluorescence images depicting the cytoplasmic and NE localization of endogenous UBAP2L and FXR1 in interphase HeLa cells. Nuclei were stained with

DAPI. The magnified framed regions are shown in the corresponding numbered panels. The arrowheads indicate co-localization of UBAP2L and FXR1 foci in the cytoplasm. Scale bar, 5 μm. **(F)** Representative immunofluorescence images depicting the localization of FXR1, Flag alone, and Flag-UBAP2L (WT or R131–190A) in late telophase in HeLa cells. Nuclei were stained with DAPI. The magnified framed regions are shown in the corresponding numbered panels. Note that Flag-UBAP2L WT (arrowheads) but not Flag alone and Flag-UBAP2L R131–190A is localized to FXR1 containing granules in proximity of NE. Scale bar, 5 μm.

FXR1-containing granules were increased in dividing UBAP2L-deficient cells relative to WT cells (Fig. 7, C and D). These results suggest that UBAP2L may remodel FXR1 protein assemblies present in mitotic cells to restrict their timely localization to the vicinity of the NE after completion of mitosis, when they could interact with Nups and transport them toward NE allowing for the formation of mature NPCs during early interphase. Indeed, endogenous UBAP2L and FXR1 can localize to NE and occasionally colocalize in the cytoplasmic assemblies in the proximity of NE in early interphasic cells (Fig. 7 E). In addition, Flag-tagged WT, but not the R131–190A mutant form of UBAP2L, frequently colocalized to FXR1-containing granules in the proximity of NE in late telophase cells (Fig. 7 F) and WT but not R131–190A mutant UBAP2L was able to disperse endogenous FXR1-containing mitotic granules (Fig. S5, K–M). Similar observations were made when either the full length or the 98–430 aa UBAP2L fragment fused to GFP, but not GFP alone, were expressed in STLC-synchronized mitotic cells (Fig. S5, N–P), suggesting that UBAP2L may chaperone and/or remodel FXR1 to ensure its interaction with Nups and their timely localization to the NE. The exact molecular mechanism underlying UBAP2L-mediated remodeling of FXR1 will have to be investigated in the future, but it is interesting that DNAJB6, a molecular chaperone of the heat shock protein network, which was demonstrated to prevent aggregation of Nups and promote their NE assembly during interphase (Kuiper et al., 2022), could also interact with endogenous UBAPL2 in our hands (Fig. S5 Q), further corroborating the role of UBAP2L in the assembly of cytoplasmic Nups. Collectively, our results identify UBAP2L as an important component of the FXRPs-Nups pathway that promotes assembly or stability of NPCs during early interphase by regulating the localization of FXR1 and Nups to the NE during early G1.

### UBAP2L regulates the function of NPCs on nuclear transport and cellular proliferation

Next, it was important to understand the physiological relevance and functional implications of the UBAP2L-mediated regulation of NPCs at the NE. Our data so far demonstrated that deletion of UBAP2L leads to the cytoplasmic sequestration of some mAb414-reactive FG-Nups (Fig. 3, A and H), which constitute the selective permeability barrier of NPCs as well as of Importin-β and Exportin-1 (Fig. 3, A, C, and F), the essential components of the nucleocytoplasmic transport system (Pemberton and Paschal, 2005). UBAP2L KO cells also display a reduced number of NPCs at the intact NE (Fig. 5, A, B, E, and F). To understand if these defects affect the function of nuclear pores in UBAP2L-deficient cells, we measured the rates of nucleocytoplasmic transport of an ectopic import/export reporter plasmid XRGG-GFP that shuttles to the nucleus (accumulating in the nucleoli) when induced with dexamethasone as previously described (Agote-Aran et al., 2020; Love et al., 1998). Deletion of UBAP2L

decreased the rates of XRGG-GFP nuclear import (Fig. 8, A and B) and its nuclear export (Fig. 8, C and D) relative to WT cells, suggesting that UBAP2L is important for the transport function of NPCs. To corroborate these observations using a marker that does not localize at specific structures, we analyzed the gradient of endogenous Ran, a guanine nucleotide triphosphatase, as shown previously (Coyne et al., 2020; Zhang et al., 2015). Since most of Ran protein is actively imported to the nucleus with the help of transport factors (Ribbeck et al., 1998; Smith et al, 1998, 2002), we analyzed the nuclear–cytoplasmic (N/C) distribution of Ran and observed significant reduction in the N/C ratio of Ran in UBAP2L KO cells (Fig. 8, E and F). Together, with our analysis in living cells, and with the reduced nuclear levels of Ran in fractionation experiments (Fig. 3 J and Fig. S3, F and G), these results suggest that UBAP2L may facilitate the nucleocytoplasmic transport across the NE.

Interestingly, in the live video analysis, we observed that UBAP2L-deficient cells displaying strong transport defects may undergo cellular death (Fig. 8 C) in accordance with the previous reports demonstrating an essential role of transport across NE for cell viability (Hamada et al., 2011). Colony formation assays showed that the long-term proliferation capacity of both UBAP2L KO cell lines was reduced relative to WT cells (Fig. S5, R–V) in agreement with our published study (Guerber et al., 2023) and propidium iodide (PI) labeling and flow cytometry indicated reduced viability of UBAP2L KO cells (Fig. S5, W and X). Future studies will have to address whether UBAP2L-dependent regulation of NPCs can directly promote cell survival or if the effects of UBAP2L deletion on NPC function and viability are circumstantial.

### UBAP2L-dependent regulation of Nups facilitates adaptation to nutrient stress

Because the Y-complex can selectively affect survival of cancer cells in response to the presence of nutrients (such as high serum and growth factors) (Sakuma et al., 2020), and changes in nutrient availability can lead to NPC reorganization (clustering) in fission yeast (Varberg et al., 2022), we studied how UBAP2L-dependent regulation of Nups can be affected by nutrient deprivation in human cells.

Serum deprivation led to reduced NE levels of Nups and accumulation of Nups foci (Fig. 9, A–C). Interestingly, NE localization and protein levels of UBAP2L were moderately reduced upon serum deprivation (Fig. 9, A, D, and E) but the total protein levels of several tested Nups were unaffected under nutrient-poor conditions (Fig. 9 D). Serum starvation further potentiated inhibition of cell viability in a UBAP2L-dependent manner (Fig. S5, W and X) but did not lead to more severe Nups defects in UBAP2L KO cells (Fig. 9, F and G), suggesting that additional pathways may contribute to UBAP2L-dependent cell survival under serum poor conditions. To exclude the possibility that

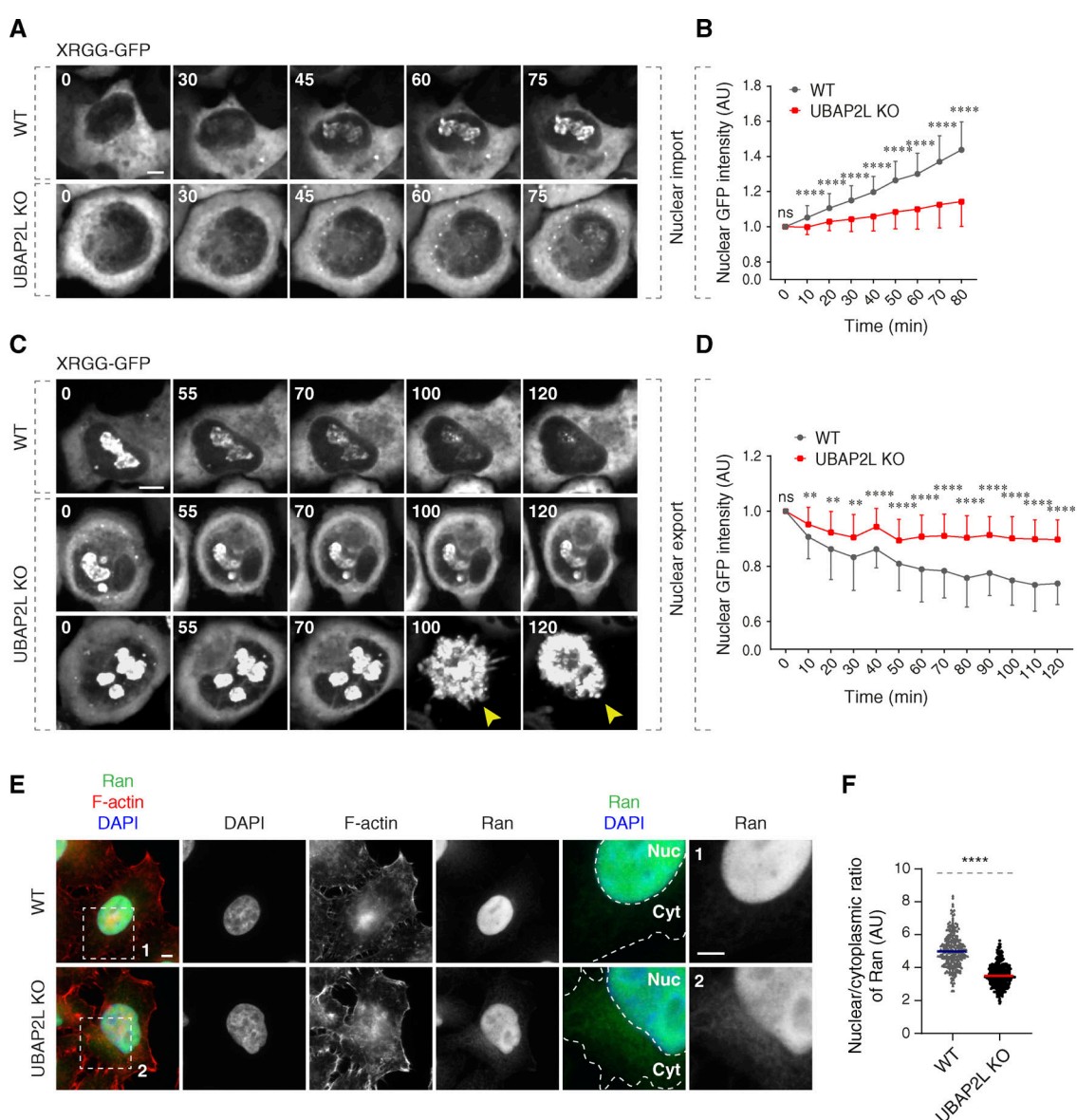

Figure 8. **UBAP2L regulates nucleocytoplasmic transport. (A–D)** WT and UBAP2L KO HeLa cells expressing reporter plasmid XRGG-GFP for 30 h were analyzed by live video spinning disk confocal microscopy. The selected representative frames of the movies are depicted, and time is shown in minutes. Timepoint 0 in the top panel (nuclear import of XRGG-GPF) indicates that dexamethasone (0.01 µM) was added, while timepoint 0 in the bottom panel (nuclear export of XRGG-GPF) indicates that dexamethasone was washed out. The arrowheads indicate dead cells in UBAP2L KO cells. Scale bars, 5 µm (A and C). The nuclear intensity (fold change) of XRGG-GFP (to DNA labeled by SiR-DNA probe) in top panel (nuclear import) (B) and in bottom panel (nuclear export) (D) shown in A and C were quantified. At least 10 cells per condition were analyzed (mean ± SD, ns: not significant, **P < 0.01, ****P < 0.0001, unpaired two-tailed *t* test, *n* = 3 independent experiments). **(E and F)** Representative immunofluorescence images depicting the nuclear (Nuc) and cytoplasmic (Cyt) localization of Ran in asynchronously proliferating WT and UBAP2L KO HeLa cells. Actin filaments (also known as F-actin) were stained with phalloidin. The magnified framed regions are shown in the corresponding numbered panels. Nuclei were stained with DAPI. Scale bar, 5 µm (E). The N/C ratio of Ran shown in E was quantified (F) (mean ± SD, ****P < 0.0001, unpaired two-tailed *t* test; counted 277 cells for WT and 306 cells for UBAP2L KO).

serum starvation induced cell cycle arrest where UBAP2L is not operational, we analyzed Nups localization in early and late G1 cells. Both early G1 as well as phospho-Rb–positive cells (mid-late G1, S, and G2 phases) displayed increased cytoplasmic Nup foci in response to serum deprivation (Fig. 9, H–J), similar to the results obtained in UBAP2L KO cells (Fig. 4, B, E, and F) and despite reduced percentage of phospho-Rb–positive cells (Fig. 9 K).

Serum starvation could also lead to reduced density of NPCs at the NE, a phenotype that could be partially rescued by overexpression of GFP-UBAP2L (Fig. 9, L and M), suggesting that the presence of UBAP2L is important for NPC homeostasis also under serum stress conditions. Finally, deprivation of serum (Fig. 9, N and O) and amino acids (Fig. 9, P and Q) could induce the formation of the cytoplasmic Nup granules, which were rescued by Flag-UBAP2L overexpression also upon inhibition of active protein translation (using cycloheximide, CHX), suggesting that UBAP2L-mediated NPC regulation under nutrient stress conditions is, at least partially, independent of production

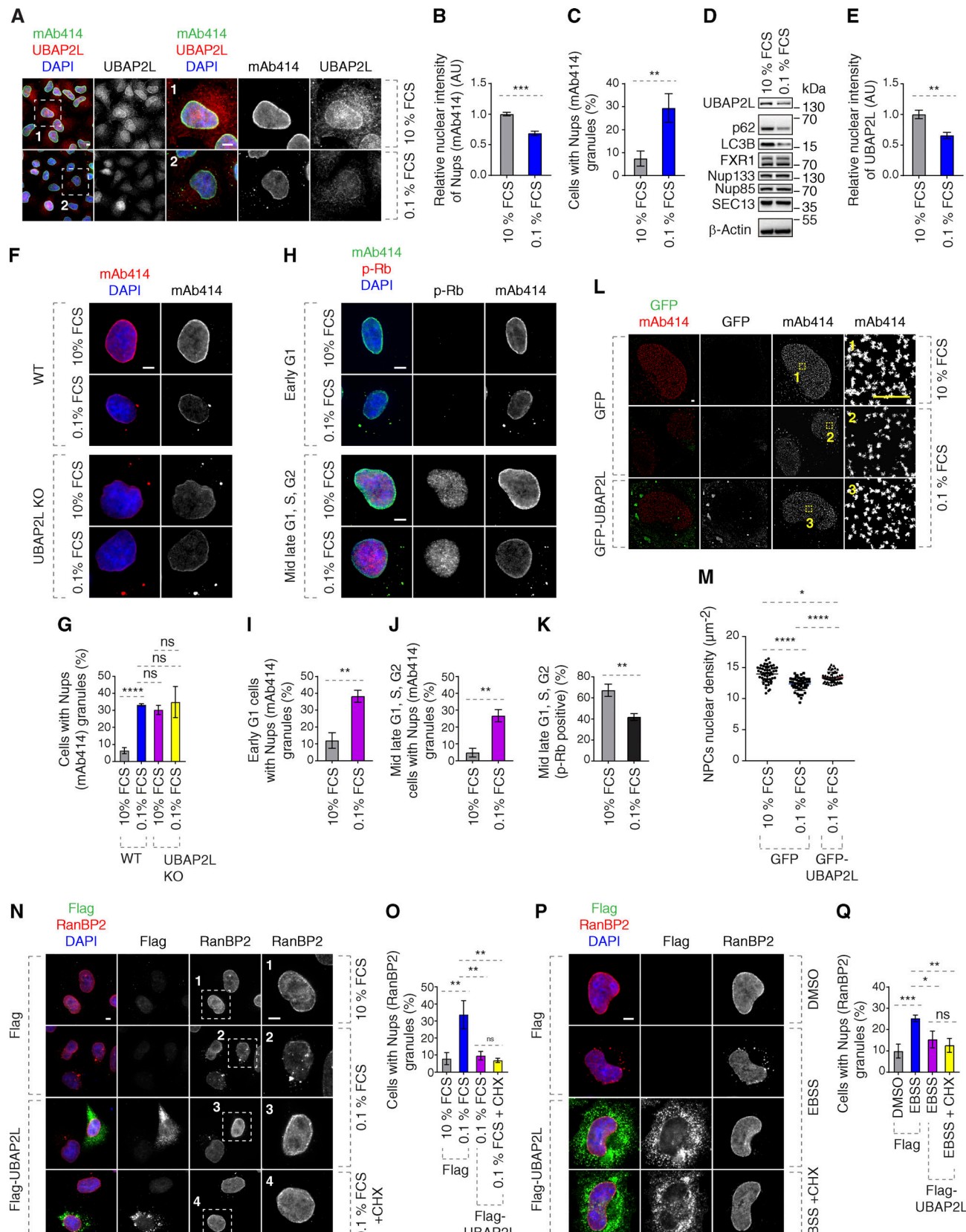

**Figure 9. UBAP2L-dependent regulation of Nups facilitates adaptation to nutrient stress. (A–E)** Representative immunofluorescence images depicting the localization of UBAP2L and Nups (mAb414) in HeLa cells cultured in the indicated concentrations of serum for 72 h. Nuclei were stained with DAPI. Scale bars, 5 µm (A). The protein levels of UBAP2L, Nups, FXR1, and other indicated factors shown in A were analyzed by western blot (D). The nuclear intensity of Nups (mAb414) (B) and the percentage of cells with the cytoplasmic granules of Nups (mAb414) (C) and the nuclear intensity of UBAP2L (E) shown in A were

quantified. At least 100 cells per condition were analyzed (mean ± SD, **P < 0.01, ***P < 0.001, unpaired two-tailed t test, n = 3 independent experiments). **(F and G)** Representative immunofluorescence images depicting the localization of Nups (mAb414) in WT and UBAP2L KO HeLa cells cultured in the indicated concentrations of serum for 72 h. Nuclei were stained with DAPI. Scale bar, 5 µm (F). The percentage of cells with the cytoplasmic granules of Nups (mAb414) (G) shown in F was quantified. At least 100 cells per condition were analyzed (mean ± SD, ns: not significant, ****P < 0.0001, unpaired two-tailed t test, n = 3 independent experiments). **(H–K)** Representative immunofluorescence images depicting the localization of p-Rb and Nups (mAb414) in HeLa cells cultured in the indicated concentrations of serum for 72 h. Nuclei were stained with DAPI. Scale bars, 5 µm (H). The percentage of cells with the cytoplasmic granules of Nups (mAb414) in early G1 (I) and mid-late G1, S, G2 (J), and the percentage of p-Rb–positive cells (K) shown in H were quantified. At least 100 cells per condition were analyzed (mean ± SD, **P < 0.01, unpaired two-tailed t test, n = 3 independent experiments). **(L and M)** Representative SMLM immuno-fluorescence images of FG-Nups (mAb414) at the nuclear surface in interphase HeLa cells expressing GFP alone or GFP-UBAP2L WT for 48 h cultured in the indicated concentrations of serum for 72 h. The magnified framed regions are shown in the corresponding numbered panels. Scale bar, 1 µm (L). The nuclear density of NPCs (mAb414) in cells shown in L was quantified (M) (mean ± SD, *P < 0.05, ****P < 0.0001, unpaired two-tailed t test; counted 51 cells per cell line). **(N and O)** Representative immunofluorescence images depicting the localization of RanBP2 in HeLa cells expressing Flag alone or Flag-UBAP2L for 30 h cultured in the indicated concentrations of serum for 72 h. Note that CHX was used at a concentration of 0.1 mg/ml for 8 h prior to sample collection. The magnified framed regions are shown in the corresponding numbered panels. Nuclei were stained with DAPI. Scale bar, 5 µm (N). The percentage of cells with the cytoplasmic granules containing RanBP2 shown in N was quantified (O). At least 100 cells per condition were analyzed (mean ± SD, ns: not significant, **P < 0.01, unpaired two-tailed t test, n = 3 independent experiments). **(P and Q)** Representative immunofluorescence images depicting the localization of RanBP2 in HeLa cells expressing Flag alone or Flag-UBAP2L for 28 h and then cultured in the Earle's Balanced Salt Solution (EBSS) medium for 4 h. Note that CHX was used at a concentration of 0.1 mg/ml for 4 h prior to sample collection. Nuclei were stained with DAPI. Scale bar, 5 µm (P). The percentage of cells with the cytoplasmic granules containing RanBP2 shown in P was quantified (Q). At least 100 cells per condition were analyzed (mean ± SD, ns: not significant, *P < 0.05, **P < 0.01, ***P < 0.001, unpaired two-tailed t test, n = 3 independent experiments). Source data are available for this figure: SourceData F9.

of new proteins during early interphase. The possible regulation of NPC biogenesis and/or stability by UBAP2L in response to nutrient-poor conditions or upon induction of autophagy will have to be investigated in the future. Taken together, our data are consistent with the hypothesis that the role of UBAP2L on NPCs is important for adaptation to nutrient stress.

### UBAP2L may contribute to stability and/or repair of NPCs at the NE

Our data could support the conclusion that UBAP2L functions specifically in de novo assembly of NPCs at the NE. However, an alternative interpretation exists where UBAP2L could also contribute to NPC stability during early interphase. To test this possibility and avoid any possible compensatory effects in KO cells due to prolonged UBAP2L absence, we used siRNA-mediated downregulation of UBAP2L and inhibited protein translation by CHX. In agreement with the previous results (Fig. S3, K–N), downregulation of UBAP2L led to a significant increase in the Nups granules in the cytoplasm and to reduction of Nup intensity at the NE (Fig. 10, A–D). These effects were moderately modulated by translation inhibition, where CHX decreased the presence of Nup foci in UBAP2L-depleted cells but not in control cells (Fig. 10, A and C) as well as decreased the NE Nups levels in both groups (Fig. 10, A and D). These observations suggest that de novo assembly of Nups is partially dependent on UBAP2L, which might also be involved in the regulation of NPC stability at the NE.

Since Flag-UBAP2L overexpression could rescue Nups defects in nutrient-stressed cells also upon inhibition of active protein translation (Fig. 9, N–Q), we further aimed to investigate the possible role of UBAP2L in NPC stability using SNAP-Nup85 stable cell line (Fig. 10, E and F), which allowed for a pulse la-beling of the "old" pool of existing Nup85 prior to extensive washes to prevent subsequent labeling of a newly made pool of Nup85 and downregulation of UBAP2L or Nup153, previously implicated in de novo interphase NPC assembly (Vollmer et al., 2015) (Fig. 10, G). Surprisingly, downregulation of UBAP2L or Nup153 led to cytoplasmic mislocalization and reduced NE

intensity of SNAP-Nup85 compared with control cells (Fig. 10, H–K). Relative to UBAP2L, downregulation of Nup153 further decreased NE intensity of SNAP-Nup85. Although we cannot fully exclude the possibility that some labeling of new Nup85 pool took place during the course of the experiment, these re-sults suggest that in addition to their role in the NPC assembly de novo, UBAP2L, and unexpectedly Nup153, may also regulate stability or repair of NPCs during interphase. The exact molec-ular mechanisms and additional factors supporting this dual role of UBAP2L on NPC homeostasis will be the subject of future investigations.

## Discussion
NPCs are large eightfold symmetrical assemblies composed of multiple copies of 30 different Nups. Nups assemble into bio-chemically stable subcomplexes that form eight identical pro-tomer unit, known as "spokes," radially arranged around the central channel. Although deviations from typical eightfold ro-tational symmetry have been observed (Hinshaw and Milligan, 2003) and NPCs can dilate their inner ring by moving the spokes away from each other (Mosalaganti et al., 2018; Taniguchi et al., 2024, Preprint), the molecular pathways defining NPC structural organization are largely unknown.

Our data suggest a model (Fig. 10 L) where UBAP2L localizes to the NE and NPCs and may facilitate Y-complex formation and its interaction with NE-targeting Nups Nup153 and POM121. It also remodels FXRP proteins, restricting their timely localization to the NE and interaction with the Y-complex. Thus, UBAP2L integrates nuclear and cytoplasmic NPC assembly signals to ensure homeostasis of NPCs during interphase (Fig. 10 L). Our data are consistent with the role of UBAP2L in the biogenesis of new NPCs but we also present some evidence that UBAP2L may regulate a repair mechanism of existing NPCs possibly through its function on Y-complex Nups (Fig. 10, A–K).

The Y-complex is an essential scaffold component of the NPC that oligomerizes head to tail in double-ring arrangements in each cytoplasmic and nuclear outer ring (Bui et al., 2013). The

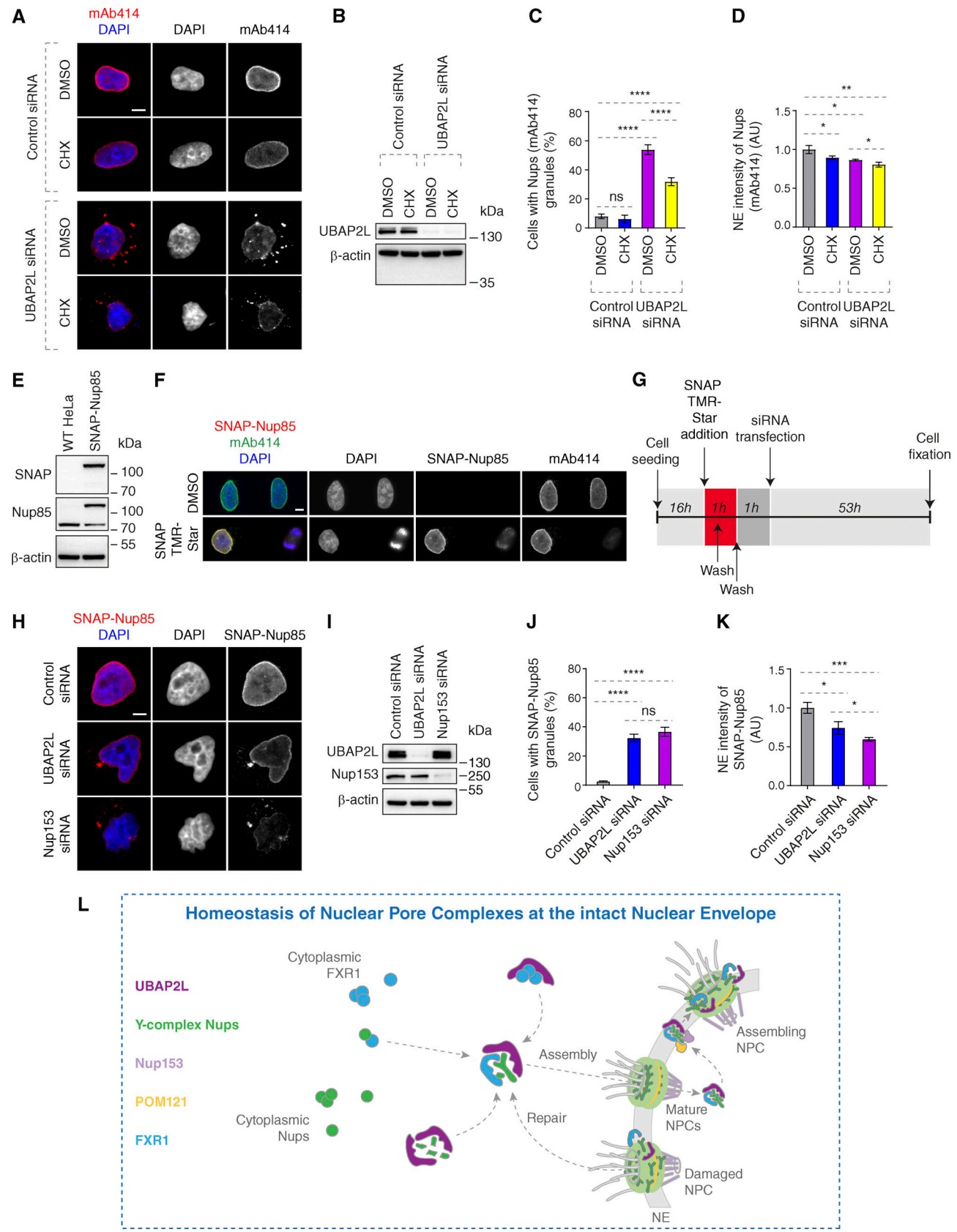

Figure 10. **UBAP2L regulates homeostasis of NPCs at the intact NE. (A–D)** Representative immunofluorescence images depicting localization of Nups (mAb414) in HeLa cells treated with indicated siRNAs and synchronized in interphase by DTBR at 12 h. Note that CHX was used at a concentration of 0.1 mg/ml for 2 h 40 min prior to sample collection. Nuclei were stained with DAPI. Scale bar, 5 µm (A). UBAP2L protein levels in A were analyzed by western blot (B). The

percentage of cells with the cytoplasmic granules of Nups (mAb414) (C) and the NE intensity of Nups (mAb414) (D) shown in A were quantified. At least 100 cells per condition were analyzed (mean ± SD, ns: not significant, *P < 0.05, **P < 0.01, ****P < 0.0001, unpaired two-tailed *t* test, n = 3 independent experiments). **(E and F)** Validation of SNAP-Nup85 HeLa cells by western blot (E) and immunofluorescence microscopy (F). SNAP-Cell TMR-Star was used according to the established protocols. Note that SNAP-Nup85 cells were incubated with SNAP-Cell TMR-Star for 30 min, washed extensively, and the medium was exchanged two times to remove any unreacted SNAP-tag substrate before sample collection. Nuclei and chromosomes were stained with DAPI. Scale bar, 5 μm. **(G–K)** Scheme of the experimental setup of the SNAP-Nup85 experiment (G). Representative immunofluorescence images depicting localization of SNAP-Nup85 in unsynchronized SNAP-Nup85 HeLa cells treated with indicated siRNAs. Nuclei were stained with DAPI. Scale bar, 5 μm (H). UBAP2L and Nup153 protein levels in H were analyzed by western blot (I). The percentage of cells with the cytoplasmic granules of SNAP-Nup85 (J) and the NE intensity of SNAP-Nup85 (K) shown in H were quantified. At least 100 cells per condition were analyzed (mean ± SD, ns: not significant, *P < 0.05, ***P < 0.001, ****P < 0.0001, unpaired two-tailed *t* test, n = 3 independent experiments). **(L)** Hypothetical model of how UBAP2L regulates the homeostasis of NPCs at the intact NE. In the proximity of the NE, UBAP2L (dark purple) interacts with cytoplasmic Y-complex Nups (green) and may facilitate the formation of Y-complex. UBAP2L also interacts with the transporting factor of Nups in the cytoplasm, FXR1 (blue), and restricts its localization to NE during early G1 phase, and promotes its interaction with Nups to fuel assembly NPCs. UBAP2L also regulates the interaction of Y-complex Nups with Nup153 (light purple) and POM121 (yellow), which facilitates the assembly of functional and mature NPCs during interphase. At the same time, UBAP2L may exert its repair function to maintain the stability of existing NPC on NE. This dual regulatory mechanism integrates the cytoplasmic and the nuclear NPC assembly as well as the NE NPC repair signals and ensures efficient nuclear transport, adaptation to nutrient stress, and cellular proliferative capacity, highlighting the importance of NPC homeostasis at the intact NE. Source data are available for this figure: SourceData F10.

molecular mechanisms governing the spatiotemporal assembly of the Y-complex remained uncharacterized. Although our analysis lacks direct biochemical evidence and the conclusions are speculative at this point, our findings may provide some insights into this biological riddle identifying UBAP2L as a factor facilitating correct formation of the Y-complex and mature NPCs.

Indeed, IP experiments revealed reduced interaction of Y-complex Nups upon UBAP2L deletion (Fig. 5, G and H). It remains to be determined if the oligomerization status of the Y-complex and its interaction with other NPC structural elements can be also regulated by UBAP2L. Likewise, future ultrastructural approaches will be required to formally address the regulation of NPC symmetry by UBAP2L, which may explain why the organization of the NPC visualized by super-resolution microscopy appears to be altered in the absence of UBAP2L (Fig. 5, A and D). Because prepore structures display an eightfold arrangement already during the early steps of interphase NPC assembly (Otsuka et al., 2016), UBAP2L may act during the initial steps of nuclear pore formation or repair, prior to the described extrusion process. Indeed, UBAP2L protein shuttles in and out of the nucleus (Fig. S1, C–E), localizes to NPCs (Fig. 1, D–F), and interacts with several Nups (Fig. 2, A–C and Fig. 6 A). UBAP2L appears to be more frequently localized at the nuclear ring (Fig. 1, D and F), suggesting that it may be transported through existing mature NPCs to help the assembly and/or repair of new/damaged NPCs from the nuclear side. Future ultrastructural and biochemical experiments could shed some light on the presence of NPC assembly intermediates and their structural organization in UBAP2L KO cells.

NPC assembly in cells with rapid cell cycles can also involve AL, the cytoplasmic stacks of ER membranes with embedded and pre-assembled NPCs. AL can be inserted en bloc into the expanding NE in fly embryos (Hampoelz et al., 2016) and in higher eukaryotic cells (Ren et al., 2019). Interestingly, the splitSMLM analysis of UBAP2L KO cells revealed the presence of linearly organized cytoplasmic Nup assemblies where RanBP2 was distributed symmetrically and where Nup133-positive rings surrounded the central channel labeled by Nup62 (Fig. S2 A), suggesting that they may represent AL-like structures. The cytoplasmic Nup foci upon UBAP2L deletion did not contain Nup153 (Fig. 3 A), as reported previously for AL-NPCs (Hampoelz et al., 2016), and defects in DNAJB6, which interacted with UBAP2L (Fig. S5 Q), likewise induced AL (Kuiper et al., 2022), indicating that UBAP2L may, at least partially, contribute to the assembly of AL-NPC. Further experimental efforts will need to identify precise mechanisms of AL assembly or any possible links to reported molecular players such as Ran (Walther et al., 2003b) or ER- and NE-resident torsin AAA+ proteins (Rampello et al., 2020).

Importantly, the biological significance of the UBAP2L-mediated assembly of NPCs at the intact NE during early interphase is documented by defects in nucleocytoplasmic transport (Fig. 8) as well as by reduced proliferation capacity (Fig. S5, R–X) observed in UBAP2L-deficient cells. Although future studies will need to address whether UBAP2L-dependent regulation of NPC assembly can directly promote cell survival, one could speculate that the role of UBAP2L in NPC biogenesis may explain, at least to some extent, the reported oncogenic potential of UBAP2L. This role of UBAP2L might be further regulated to meet differential demands on NPC functionality which may operate during changing cellular conditions such as stress or nutrient availability. Interestingly, deletion of Y-complex Nups can selectively affect survival and proliferation of colon cancer cells in response to presence of nutrients (Sakuma et al., 2020), and UBAP2L is sufficient to restore the NPC density after nutrient deprivation (Fig. 9, L–Q), suggesting that UBAP2L-Nup pathway plays an important role under nutrient stress conditions, which have been previously implicated in the regulation of NPC numbers in fission yeast (Varberg et al., 2022).

Taken together, our findings identify a molecular pathway ensuring homeostasis of mature and functional NPCs at the intact envelope in human cells. We propose a detailed mechanism fueling the assembly of cytoplasmic Nups onto NE through regulation of FXR1 NE localization by UBAP2L (Fig. 7 and Fig. S5, C–I) and its interaction with the Y-complex and dynein (Fig. 5, G and H; and Fig. S2 K). It can be speculated that UBAP2L may remodel cytoplasmic FXR1 assemblies found in mitotic cells to promote the reported transport function of FXR1 by microtubule transport toward NE during early interphase (Agote-Aran et al., 2020). How UBAP2L can execute its "chaperone-like" function

on either FXR1 or Y-complex Nups and what are the upstream regulatory mechanisms of UBAP2L during early interphase remain the subjects of future investigations. It will be important to study any possible links of UBAP2L to torsin AAA+ proteins (Rampello et al., 2020), biomolecular condensation factors involved in the NPC assembly such as nuclear transport receptors (NTRs) (Kuiper et al., 2023), or a precise role of protein quality control factors such as DNAJB6 (Kuiper et al., 2022) in this pathway.

Interestingly, our data demonstrate that 19 arginines present within the RGG domain may mediate UBAP2L's function on FXR1 and Nups (Fig. 6). Globally, this mechanism appears to operate in an RNA-independent manner (Fig. S4, I and J); however, at this stage of analysis, it cannot be excluded that specific RNAs might be involved in the UBAP2L-dependent regulation of Nups and FXR1. It is also possible that specific factors such as NTRs bind to the RGG region and cooperate with UBAP2L to "chaperone" FXR proteins and Nups. Consistent with a previous report (Huang et al., 2020), mutation of 19 arginines to alanines also led to loss of the ADMA signal (Fig. 6 A). This raises an intriguing possibility, to be analyzed in the future, that ADMA or other arginine modifications of UBAP2L may regulate its function on Nups and their assembly into functional NPCs at the NE during early interphase.

## Materials and methods

### Antibodies
The following primary antibodies were used in this study: anti-UBAP2L obtained with synthetic peptide CRGNWEQPQNQ NQTQHKQRPQ-(C) (Proteogenix) coupled to ovalbumin and produced in rabbit (Antibody facility Institut de génétique et de biologie moléculaire et cellulaire [IGBMC]), mouse anti-FXR1+2 clone 3FX 2B12 obtained with synthetic peptide LKDPDSNPYSLLDNTESDQT-(C) coupled to ovalbumin (Antibody facility IGBMC [Khandjian et al., 1998]), rabbit polyclonal anti-FMRP (ab17722; Abcam), rabbit polyclonal anti-GFP (ab290; Abcam), mouse monoclonal anti-β-Actin (A2228; Sigma-Aldrich), rabbit polyclonal anti-GAPDH (G9545; Sigma-Aldrich), mouse monoclonal anti-α-Tubulin (T9026; Sigma-Aldrich), mouse monoclonal anti-FLAG M2 (F1804; Sigma-Aldrich), rabbit polyclonal anti-FLAG (F7425; Sigma-Aldrich), rabbit polyclonal anti-FXR1 (HPA018246; Sigma-Aldrich), rabbit polyclonal anti-Lamin A (C-terminal) (L1293; Sigma-Aldrich), rabbit monoclonal anti-Nup98 (C39A3) (2598; Cell Signaling Technology), rabbit polyclonal anti-PRMT1 (A33) (2449; Cell Signaling Technology), rabbit polyclonal anti-Tubulin (ab18251; Abcam), rabbit polyclonal anti-UBAP2L (ab138309; Abcam), rabbit monoclonal anti-Nup133 (ab155990; Abcam), mouse monoclonal anti-NPC Proteins (mAb414) (ab24609; Abcam), rabbit polyclonal anti-Nup153 (ab84872; Abcam), rabbit polyclonal anti-Nup188 (ab86601; Abcam), rabbit polyclonal anti-RanBP2 (ab64276; Abcam), rabbit polyclonal anti-Lamin B1 (ab16048; Abcam), rabbit monoclonal anti-NTF97/Importin beta (ab2811; Abcam), mouse monoclonal anti-Cyclin B1 (G-11) (sc-166757; Santa Cruz Biotechnology), mouse monoclonal anti-Cyclin E (HE12) (sc-247; Santa Cruz Biotechnology), mouse monoclonal anti-Nup133 (E-6) (sc-376763; Santa Cruz

Biotechnology), mouse monoclonal anti-TIA-1 (sc-166247; Santa Cruz Biotechnology), mouse monoclonal anti-FXR1 (03-176; Millipore), rabbit polyclonal anti-dimethyl-Arginine, asymmetric (ASYM24) (07-414; Millipore), mouse monoclonal anti-Mps1 (05-682; Millipore), rabbit polyclonal anti-G3BP1 (GTX112191; GeneTex), rabbit polyclonal anti-POM121 (GTX102128; Gene-Tex), rabbit polyclonal anti-Cyclin B1 (GTX100911; GeneTex), rabbit polyclonal anti-FXR2 (12552-1-AP; Proteintech), mouse monoclonal anti-Nucleoporin p62 (610497; BD Biosciences), mouse monoclonal anti-Ran (610340; BD Biosciences), rabbit monoclonal anti-SEC13 (MAB9055; R&D systems), rabbit polyclonal anti-UBAP2L (1025–1087 aa) (A300-534A; Bethyl), rabbit polyclonal anti-Nup85 (A303-977A; Bethyl), rabbit polyclonal anti-Nup160 (A301-790A; Bethyl), rabbit polyclonal anti-CRM1/Exportin 1 (NB100-79802; Novus) and rat monoclonal anti-GFP (3H9) (3h9-100; ChromoTek), rabbit polyclonal anti-LC3B (NB100-2331; Novus biological), guinea pig polyclonal anti-p62 (GP62-C; Progen), rabbit monoclonal anti-DNAJB6 (ab198995; Abcam), rabbit monoclonal anti-BiCD2 (HPA023013; Sigma-Aldrich), mouse monoclonal anti-p150^Glued (610473; BD Biosciences), and rabbit polyclonal anti-SNAP-tag (P9310S; New England Biolabs).

Secondary antibodies used were the following: goat polyclonal anti-Mouse CF680 (SAB4600199; Sigma-Aldrich), goat polyclonal anti-Chicken CF660C (SAB4600458; Sigma-Aldrich), goat polyclonal anti-Mouse AF647 (A-21236; Thermo Fisher Scientific), goat polyclonal anti-Mouse AF568 (A-11031; Thermo Fisher Scientific), goat polyclonal anti-Mouse AF555 (A-11029; Thermo Fisher Scientific), goat polyclonal anti-Mouse AF488 (A-21424; Thermo Fisher Scientific), goat polyclonal anti-Rabbit AF647 (A-21245; Thermo Fisher Scientific), goat polyclonal anti-Rabbit AF568 (A-11036; Thermo Fisher Scientific), goat polyclonal anti-Rabbit AF555 (A-21429; Thermo Fisher Scientific), goat polyclonal anti-Rabbit AF488 (A-11034; Thermo Fisher Scientific), goat Anti-Mouse IgG antibody (HRP) (GTX213111-01; GeneTex), goat Anti-Mouse IgG antibody (HRP) (GTX213110-01; GeneTex), and goat Anti-Rat IgG antibody (HRP) (7077S; Cell Signaling Technology).

### Generation of UBAP2L KO and SNAP-Nup85 stable cell lines
UBAP2L KO in HeLa cells was characterized previously (Guerber et al., 2023). UBAP2L KO in Nup96-GFP KI U2OS (CLS Cell Line Service, 300174; a generous gift of Arnaud Poterszman, IGBMC [Thevathasan et al., 2019]) cell lines were generated using CRISPR/Cas9 genome editing system (Jerabkova et al., 2020). Two guide RNAs were designed using the online software Benchling (https://www.benchling.com/), 5′-TGGCCAGAC GGAATCCAATG-3′ and 5′-GTGGTGGGCCACCAAGACGG-3′, and cloned into pX330-P2A-EGFP/RFP (Zhang et al., 2017) through ligation using T4 ligase (New England Biolabs). Nup96-GFP KI U2OS cells were transfected using X-tremeGENE 9 DNA Transfection Reagent (Roche), and 24 h after transfection, GFP and RFP double-positive cells were collected by FACS (BD FACS Aria II), cultured for 2 days, and seeded with FACS into 96-well plates. Obtained UBAP2L KO single-cell clones were validated by western blot and sequencing of PCR-amplified targeted fragments by Sanger sequencing (GATC). The following primers

were used for PCR amplification: 5′-TGCTGAGTGGAGAATGGT TA-3′ (forward) and 5′-AGACTGGTGGCAGTTGGTAG-3′ (reverse). Primers used for cloning and sequencing are described in Table S1.

SNAP and SNAP-Nup85 stable cell lines were generated in HeLa Kyoto cells by random integration of pSNAPf and pSNAPf-C1-hNup85 plasmids using Lipofectamine 2000 according to the manufacturer's instructions. Transfected cells were selected for 2–3 wk in a medium supplemented by G418 (400 µg/ml). Positive transgene-expressing clones were then isolated by FACS (BD FACS Aria II). Expression was assessed by western blot and immunofluorescence analysis.

## Cell culture
All cell lines were cultured at 37°C in 5% $CO_2$ humidified incubator. HeLa (Kyoto) and its derived UBAP2L KO cell lines were cultured in Dulbecco's Modified Eagle Medium (DMEM) (4.5 g/L glucose) supplemented with 10% fetal calf serum (FCS), 1% penicillin + 1% streptomycin. SNAP and SNAP-Nup85 HeLa (Kyoto) cell lines were cultured in DMEM (4.5 g/L glucose) w/ glutamax supplemented with 10% fetal calf serum (FCS) and 1% penicillin + 1% streptomycin. U2OS cells were cultured in DMEM (1 g/L glucose) supplemented with 10% FCS + gentamicin 40 µg/ml. Nup96-GFP KI U2OS and its derived UBAP2L KO cell lines were cultured in DMEM (1 g/l glucose) supplemented with 10% FCS, non-essential amino acids + sodium pyruvate 1 mM + gentamicin 40 µg/ml.

## Cell cycle synchronization treatments
Cells were synchronized in different stages of the cell cycle by double thymidine block and release (DTBR) protocol. Briefly, cells were treated with 2 mM thymidine for 16 h, washed out (three times with warm thymidine-free medium), then released in fresh thymidine-free culture medium for 8 h, treated with 2 mM thymidine for 16 h again, washed out, and then released in fresh thymidine-free culture medium for different time periods (0, 3, 6, 8, 9, 10, and 12 h). 0 h time point corresponds to G1/S phase, ~8–9 h to mitotic peak, 10 h to mitotic exit, and 12 h to early G1 phase. Cells were synchronized in the G1 phase using lovastatin for 16 h at 10 µM final concentration and in G0/G1 using Psoralidin (3,9-Dihydroxy-2-prenylcoumestan) for 24 h at 5 µM final concentration. Cells were synchronized in prometaphase using Nocodazole for 16 h at 100 ng/ml, STLC for 16 h at 5 µM, and monastrol for 16 h at 100 µM final concentration.

## Plasmids
All pEGFP-C1-UBAP2L WT (NCBI, NM_014847.4), pEGFP-C1-UBAP2L UBA (1–97 aa), pEGFP-C1-UBAP2L ΔUBA (Δ1–97 aa), pEGFP-C1-UBAP2L 98–430 aa, pEGFP-C1-UBAP2L 1–430 aa, pEGFP-C1-UBAP2L Δ1–429 aa, pEGFP-C1-UBAP2L Δ(UBA+RGG) (Δ1–195 aa), pEGFP-C1-FXR1 WT (NCBI, NM_001013438.3), and pSNAPf-C1-hNup85 plasmids were generated by Stephane Schmucker (IGBMC). pcDNA3.1-Flag-N-UBAP2L WT (NCBI, NM_014847.4) was generated by Evanthia Pangou (IGBMC). Primers used for cloning are described in Table S1. pEGFP-C1 was purchased from Clontech. pcDNA3.1-Flag-N was obtained from the IGBMC cloning facility, and pcDNA3.1-Flag-UBAP2L

R131-190A was a generous gift of Zhenguo Chen (Southern Medical University, Guangzhou, P.R. China) (Huang et al., 2020). pEGFP-C1-Nup85 was kindly provided by Valérie Doye (Institut Jacques Monod, Paris, France) (Loïodice et al., 2004), and pXRGG-GFP was kindly provided by Jan M. van Deursen (Mayo Clinic, Rochester, MN, USA) (Hamada et al., 2011; Love et al., 1998).

## Plasmid and siRNA transfections
Lipofectamine 2000 (Invitrogen), jetPEI-DNA transfection reagent (Polyplus-transfection), and X-tremeGENE 9 DNA Transfection Reagent (Roche) were used to perform plasmid transient transfection according to the manufacturer's instructions. Lipofectamine RNAiMAX (Invitrogen) was used to deliver siRNAs for gene knock-down according to the manufacturer's instructions at a final concentration of 10–20 nM siRNA. The following siRNA oligonucleotides were used: Non-targeting individual siRNA 5′-UAAGGCUAUGAAGAGAUAC-3′ (Dharmacon), UBAP2L siRNA 5′-CAACACAGCAGCACGUUAU-3′ (Dharmacon), FXR1 siRNA 5′-AAACGGAAUCUGAGCGUAA-3′ (Dharmacon), Nup153 siRNA-1 5′-GGACTTGTTAGATCTAGTT-3′ (Dharmacon), Nup153 siRNA-2 5′-AGTGTTCAGTATGCTGTGTTT CT-3′ (Dharmacon), and Nup214 siRNA 5′-GGTGAGAATCTTTGACTC C-3′ (Dharmacon).

## Protein preparation and western blotting
Cells were collected by centrifugation at 200 $g$ for 4 min at 4°C and washed twice with cold phosphate-buffered saline (PBS), and cell lysates for western blot were prepared using 1X radio-immunoprecipitation assay (RIPA) buffer (50 mM Tris-HCl, pH 7.5, 150 mM NaCl, 1% Triton X-100, 1 mM EDTA, 1 mM EGTA, 2 mM sodium pyrophosphate, 1 mM $Na_3VO_4$, and 1 mM NaF) supplemented with protease inhibitor cocktail (Roche) and incubated on ice for 30 min. After centrifugation at 16,000 $g$ for 15 min at 4°C, cleared supernatant was transferred to new clean Eppendorf tubes, and total protein concentration was measured using Bradford assay by Bio-Rad Protein Assay kit (Bio-Rad). Nuclear and cytoplasmic proteins were prepared using the NE-PER nuclear and cytoplasmic extraction reagent kit (78833; Thermo Fisher Scientific). Protein samples were boiled for 8 min at 95°C in 1X Laemmli buffer (LB) with β-mercaptoethanol (1610747; BioRad), resolved on 10% polyacrylamide gels or precast 4–12% Bis-Tris gradient gels (NW04120BOX; Thermo Fisher Scientific) or pre-cast NuPAGE 3–8% Tris-Acetate gradient Gels (EA0378BOX; Thermo Fisher Scientific), and transferred to a polyvinylidene difluoride membrane (IPFL00010; Millipore) using a semidry transfer unit (Amersham) or wet transfer modules (BIO-RAD Mini-PROTEAN Tetra System). Membranes were blocked in 5% non-fat milk powder, 5% bovine serum albumin (BSA; 160069; Millipore), or 5% non-fat milk powder mixed with 3% BSA and resuspended in TBS-T (Tris-buffered saline-T: 25 mM Tris-HCl, pH 7.5, 150 mM NaCl 0.05% Tween) for 1 h at room temperature, followed by incubation with antibodies diluted in TBS-T 5% BSA/5% milk. All incubations with primary antibodies were performed overnight at 4°C. TBS-T was used for washing the membranes. Membranes were developed using SuperSignal West Pico (Ref. 34580; Pierce) or Luminata Forte Western HRP substrate (Ref. WBLUF0500;

Merck Millipore). Western blotting images were acquired by GE Healthcare_Amersham Imager 600 or Invitrogen iBright 1500. The grayscale value of protein bands was quantified using ImageJ software.

## IPs

Cell lysates for IPs were prepared using 1X RIPA buffer supplemented with protease inhibitor cocktail and incubated on ice for 1 h. When indicated, 1X RIPA buffer was supplemented with RNase A or Benzonase. After centrifugation at 16,000 *g* for 15 min at 4°C, cleared supernatant was transferred to new clean Eppendorf tubes. Lysates were equilibrated to volume and concentration.

For endogenous IP experiments, IgG and target-specific antibodies as well as protein G sepharose 4 Fast Flow beads (GE Healthcare Life Sciences) were used. Samples were incubated with the IgG and target-specific antibodies overnight at 4°C with rotation. Beads were blocked with 3% BSA diluted in 1X RIPA buffer and incubated for 2 h at 4°C with rotation. Next, the IgG/specific antibodies-samples and blocked beads were incubated in 1.5 ml Eppendorf tubes to a final volume of 1 ml 4 h at 4°C with rotation. The incubated IgG/specific antibodies-samples-beads were washed with washing buffer (25 mM Tris-HCl, pH 7.5, 300 mM NaCl, 0.5% Triton X-100, 0.5 mM EDTA, 0.5 mM EGTA, 1 mM sodium pyrophosphate, 0.5 mM $Na_3VO_4$ and 0.5 mM NaF) or TBS-T supplemented with protease inhibitor cocktail 4–6 times for 10 min each at 4°C with rotation. Beads were pelleted by centrifugation at 200 *g* for 3 min at 4°C. The washed beads were directly eluted in 2X LB with β-Mercaptoethanol and boiled for 12 min at 95°C for western blot.

For GFP-IP/Flag-IP experiments, GFP-Trap A agarose beads (Chromotek) or Flag beads (Sigma-Aldrich) were used. Cells expressing GFP- or Flag-tagged plasmids for at least 24 h were used to isolate proteins using 1X RIPA buffer supplemented with protease inhibitor cocktail. Beads were blocked with 3% BSA diluted in 1X RIPA buffer and incubated for 2 h at 4 °C with rotation. Samples were incubated with the blocked beads for 2 h or overnight at 4°C with rotation, and the beads were washed and boiled as for endogenous IP.

## Immunofluorescence

Cells grown on glass coverslips (Menzel-Glaser) were washed twice in PBS and then fixed with 4% paraformaldehyde (PFA; 15710; Electron Microscopy Sciences) in PBS for 15 min at room temperature. For mitotic cells immunofluorescence, cells were collected from dishes with cell scrapers, centrifuged on Thermo Fisher Scientific Shandon Cytospin 4 Cytocentrifuge for 5 min at 1,000 rpm, and fixed immediately with 4% PFA for 15 min at room temperature. After fixation, cells were washed three times for 5 min in PBS and permeabilized with 0.5% NP-40 (Sigma-Aldrich) in PBS for 5 min. Cells were washed three times for 5 min in PBS and blocked with 3% BSA in PBS-Triton 0.01% (Triton X-100, T8787; Sigma-Aldrich) for 1 h. Cells were subsequently incubated with primary antibodies in blocking buffer (3% BSA in PBS-Triton 0.01%) for 1 h at room temperature, washed three times for 8 min in PBS-Triton 0.01% with rocking, and incubated with secondary antibodies in blocking buffer for

1 h at room temperature in the dark. After incubation, cells were washed three times for 8 min in PBS-Triton 0.01% with rocking in the dark, and glass coverslips were mounted on glass slides using Mowiol containing 0.75 µg/ml DAPI (Calbiochem) and imaged with a Plan-Apochromat 63× or 100×/1.4 oil objective using Zeiss epifluorescence microscope at room temperature.

For Nups immunofluorescence, cells grown on glass coverslips were washed twice in PBS and then fixed with 1% PFA in PBS for 10 min at room temperature, washed three times for 5 min in PBS, and permeabilized with 0.1% Triton X-100 and 0.02% SDS (EU0660; Euromedex) in PBS for 5 min. After permeabilization, cells were washed three times for 5 min in PBS and blocked with 3% BSA in PBS-Triton 0.01% for 1 h at room temperature or overnight at 4°C. Cells were subsequently incubated with primary antibodies in blocking buffer (3% BSA in PBS-T) for 1 h at room temperature, washed three times for 8 min with rocking in blocking buffer, and then incubated with secondary antibodies in blocking buffer for 1 h at room temperature in the dark. After incubation, cells were washed three times for 8 min with rocking in blocking buffer in the dark and then permeabilized again with 0.1% Triton X-100 and 0.02% SDS in PBS for 1 min and postfixed for 10 min with 1% PFA in PBS at room temperature in the dark. Then coverslips were washed twice in PBS for 5 min and mounted on glass slides using Mowiol containing 0.75 µg/ml DAPI. The mounted samples were imaged with a Plan-Apochromat 63×/1.4 oil objective using Zeiss epifluorescence microscope at room temperature.

An adapted protocol (Guerber et al., 2023) was used for the experiments presented in Fig. 1 C. After the appropriate synchronization using DTBR, the cytoplasm was extracted from the cells to remove the large cytoplasmic fraction of UBAP2L by incubating the coverslips in cold 0.01% Triton X-100 for 90 s. 4% PFA was immediately added to the coverslips after pre-extraction, and the standard immunofluorescence protocol was followed. The mounted samples were imaged (Z-stacks, 10.5 µm range, 1.5 µm step) with a Leica HCX PL APO 63×/1.4 oil immersion objective using inverted point scanning Leica TCS confocal SP8-UV confocal controlled with the LAS X software at room temperature. 405, 488, and 561 nm laser lines were used for the excitation of DAPI, α-Tubulin, and UBAP2L, respectively. Maximum projection processing of the z-stack was performed post-acquisition using the Fiji software.

## Sample preparation for SMLM

For splitSMLM, cells were plated on 35-mm glass-bottom dish with 14 mm microwell #1.5 cover glass (Cellvis). Cells were washed twice with PBS (2 ml/well) and then fixed with 1% PFA in PBS for 15 min at room temperature, washed three times for 5 min in PBS (store samples submerged in PBS at 4°C until use), and permeabilized with 0.1% Triton X-100 in PBS (PBS/Tx) for 15 min. Cells were blocked with 3% BSA in 0.1% PBS/Tx (PBS/Tx/B) for 1 h and then incubated with primary antibodies (optimal working concentration of primary antibody is 2 µg/ml) in PBS/Tx/B (200 µl/well) overnight at 4°C in a wet chamber. After incubation, cells were washed three times for 8 min with rocking in PBS/Tx/B and subsequently incubated with secondary antibodies (optimal working concentration of secondary

antibody is 4 µg/ml) in PBS/Tx/B (200 µl/well) for 2 h at room temperature in the dark. Immediately after, cells were washed three times for 8 min with rocking in PBS/Tx and postfixed for 10 min with 1% PFA at room temperature in the dark, and then cells were washed twice in PBS and kept in PBS in the dark.

The samples were imaged in a water-based buffer that contained 200 U/ml glucose oxidase, 1,000 U/ml catalase, 10% wt/vol glucose, 200 mM Tris-HCl, pH 8.0, 10 mM NaCl, and 50 mM Monoethanolamine (MEA). 2 mM cyclooctatetraene was added to the buffer for multicolor imaging (Andronov et al., 2022). The mixture of 4 kU/ml glucose oxidase (G2133; Sigma-Aldrich) and 20 kU/ml catalase (C1345; Sigma-Aldrich) was stored at –20°C in an aqueous buffer containing 25 mM KCl, 4 mM Tris(2-carboxyethyl)phosphine, 50% vol/vol glycerol, and 22 mM Tris-HCl, pH 7.0. MEA-HCl (30080; Sigma-Aldrich) was stored at a concentration of 1 M in $H_2O$ at –20°C. Cyclooctatetraene (138924; Sigma-Aldrich) was stored at 200 mM in dimethyl sulfoxide at –20°C. The samples were mounted immediately prior to imaging filling the cavity of the glass-bottom petri dishes with ~200 µl of the imaging buffer and placing a clean coverslip on top of it, which allowed imaging for ≥8 h without degradation of the buffer. After imaging, the samples were washed once with PBS and kept in PBS at +4°C.

## SMLM

The SMLM experiments were performed on a splitSMLM system (Andronov et al., 2022) that consisted of a Leica DMI6000B microscope, an HCX PL APO 160×/1.43 Oil CORR TIRF PIFOC objective, a 642-nm 500-mW fiber laser (MBP Communication Inc.) for fluorescence excitation and a 405-nm 50-mW diode laser (Coherent Inc.) for reactivation of fluorophores. The sample was illuminated through a Semrock FF545/650-Di01 dichroic mirror and the fluorescence was filtered with Semrock BLP01-532R and Chroma ZET635NF emission filters. For single-color imaging that was used for estimation of the NPC density at the NE, the fluorescence was additionally filtered with a Semrock BLP01-635R-25 long-pass filter and was projected onto an Andor iXon+ (DU-897D-C00-#BV) EMCCD camera.

For multicolor imaging, the fluorescence was split into two channels with a Chroma T690LPXXR dichroic mirror inside an Optosplit II (Cairn Research) image splitter. The short-wavelength channel was additionally filtered with a Chroma ET685/70m bandpass filter and both channels were projected side-by-side onto an Andor iXon Ultra 897 (DU-897U-CS0-#BV) EMCCD camera.

The SMLM acquisitions began with a pumping phase during which the sample was illuminated with the 642 nm laser but the fluorescence was not recorded due to a very high density of fluorophores in a bright state. The image collection started when the density of fluorophores dropped to a level that allowed observation of individual molecules, typically after ~10 s of pumping. Pumping and imaging were performed at 30–50% of the maximal power of the 642 nm laser. When the density of fluorophores in the bright state dropped further due to photobleaching, the sample started to be illuminated with the 405 nm laser for reactivation of fluorophores. The intensity of the 405 nm laser was increased gradually to account for the

photobleaching. For estimation of the NPC density, to increase speed, the pumping and imaging were performed at 100% laser power and the acquisitions were stopped after about 2 min of imaging.

## Processing of SMLM data

The fitting of single-molecule localizations was done in the Leica LAS X software with the "direct fit" method. For single-color imaging, the obtained localization tables were corrected for drift and reconstructed as 2D histograms with a pixel size of 15 nm in SharpViSu (Andronov et al., 2016). For multicolor imaging, the localizations were first unmixed in SplitViSu (Andronov et al., 2022). Next, they were corrected for drift and relocalizations in SharpViSu and reconstructed as 2D histograms with a pixel size of 5 nm.

For quantification of the rotational symmetry of the NPCs, individual NPCs were picked manually on the NE of each imaged cell. Only particles that were in focus and correct "top view" orientation were selected. For the analysis, the localizations within a radius of 130 nm from the manually picked center of each NPC were used. The obtained particles were aligned in smlm_datafusion2d with random rotation of every particle by $n \cdot 45°$, $n = [0, 7]$, after each alignment iteration (Heydarian et al., 2018). The aligned particles were then converted to polar coordinates, and localizations with radii from 50 to 70 nm were kept for further analysis. A sine function with a period of $\pi/4$ was fitted to the polar angle distribution of the sum of all aligned particles. The localizations were split into eight sectors using the minima of the sine function as the edges of the sectors. The number of localizations within each sector was calculated for each NPC. For a given NPC, a sector was considered occupied if the number of localizations within it was higher than half of the mean number of localizations per sector for this NPC. The quantified number of subunits of an NPC is the number of occupied sectors.

An adapted protocol (Andronov et al., 2022) was used to obtain the axial and radial profiles of the NPC. For radial profiles, the localizations of coimaged proteins were transformed using the alignment parameters of Nup96 after eightfold alignment in smlm_datafusion2d (Heydarian et al., 2018). For the "side view" profile, the axial profiles of individual particles were calculated in Fiji (Schindelin et al., 2012), averaging through the whole thickness of the NPC. The axial profiles of Nup96 particles were fitted with a sum of two Gaussians in Matlab. Nup96 particles and coimaged proteins were aligned using this fit of Nup96.

## Live cell imaging

For FXR1 mitotic granules assay, WT and UBAP2L KO HeLa cells expressing GFP-FXR1 were grown on 35/10 mm four-compartment glass bottom dishes (627871; Greiner Bio-One) and synchronized by double thymidine block, released for 8 h, and analyzed in time-lapse imaging experiments. Time-lapse imaging was performed with a spinning disk microscope composed of a Nikon Eclipse Ti2 microscope equipped with a Yokogawa CSU-X1 Confocal Scanner Unit, a Perfect Focus system module, and a Photometrics Prime 95B camera (1,200 × 1,200

pixels, pixel size 11 µm). Z-stacks (7 µm range, 1 µm step) were acquired every 5 min for 9 h using a piezo Stage (mulitposition mode). The system was controlled by the Metamorph Software. Objective was a Nikon CFI Plan Apochromat Lambda D 60× 1.42 NA Oil immersion. A Tokai stage top incubator was used to maintain the samples at 37°C, 5% $CO_2$, and 85% humidity condition. Image quantification analysis was performed using ImageJ software and movies were made with maximum intensity projection images for every time point shown at a speed of seven frames per second.

For protein import and export assay, WT and UBAP2L KO HeLa cells were grown on an eight-well chambered coverglass with non-removable wells (155411PK; Thermo Fisher Scientific) and transfected with the reporter plasmid XRGG-GFP for 30 h and incubated with full media with SiR-DNA 1:1,500 and Verapamil 1:1,000 for at least 1 h before filming. Then SiR-DNA and Verapamil were kept with media and cells were incubated in media with 0.01 µM dexamethasone. Dexamethasone-induced nuclear import of XRGG-GFP was recorded by Leica DMI8 microscope (Leica HC PL APO CS2 63× 1.4NA oil immersion objective) equipped with a Yokogawa CSU-W1 confocal scanner unit, an adaptative focus control option, and a Hamamatsu Orca Flash 2.0 camera (2,048*2,018 pixels with a pixel size of 6.5 µm) for 129 min (1 acquisition every 1 min, 12 µm range, 3 µm step). For nuclear export, dexamethasone was washed out at the 129-min time point with warm dexamethasone-free medium, cells were incubated with full media with SiR-DNA 1:6,000 and Verapamil 1:4,000, and nuclear export of XRGG-GFP was recorded for 170 min (1 acquisition every 1 min, 12-µm range, 3-µm step). The microscope was equipped with an OKOLAB incubation chamber to maintain an internal environment at 37°C, 5% $CO_2$, and 85% humidity condition. The system is controlled by the Metamorph Software. Image quantification analysis was performed using ImageJ software.

### NE intensity analysis of NPCs
A CellProfiler software pipeline was previously generated by Arantxa Agote-Arán (Agote-Aran et al., 2020) that automatically recognizes cell nuclei based on the DAPI fluorescent image. A threshold of nuclei size was applied to the pictures to exclude too small or too big nuclei, and nuclei edges were enhanced using the Prewitt edge-finding method. This allowed identification and measurement of the nuclei area, form factor, and nuclear mean intensity of desired channels. The parameters' measurements of the software were exported to an Excel file and statistically analyzed. At least 150 cells from three different biological replicates were measured.

### Colocalization analysis
To assess pixel colocalization/correlation, we used correlation measurement within CellProfiler. Briefly, "Measure Colocalization" modules were used to study the colocalization and correlation between intensities in different images (different color channels) on a pixel-by-pixel basis across an entire image. The number of cells measured per condition was listed in the corresponding figure's legend.

### Colony formation assay
500 WT and UBAP2L KO HeLa cells were seeded per well in 6-well plates and incubated at 37°C in 5% $CO_2$ for 7 days until colonies formed. Cells were washed with 1X PBS, fixed with 4% PFA, and stained with 0.1% Crystal Violet for 30 min. The number of colonies was first manually counted and then automatically quantified with Fiji software.

### Flow cytometry
For cell death analysis, HeLa cells were spun down and resuspended in cold PBS supplemented with 50 µg/ml PI (Ref. P4170; Sigma-Aldrich). PI-positive cells were analyzed by BD FACS Celesta Flow Cytometer.

### Experimental design, data acquisition, and statistical analysis
All experiments were done in a strictly double-blind manner. At least three independent biological replicates were performed for each experiment (unless otherwise indicated) and image quantifications were carried out in a blinded manner. Curves and graphs were made using GraphPad Prism and Adobe Illustrator software. Data were analyzed using a one-sample two-tailed $t$ test or two-sample unpaired two-tailed $t$ test (two-group comparison or folds increase relative to the control, respectively) or one-way ANOVA. A P value <0.05 (typically ≤0.05) was considered statistically significant and stars were assigned as follows: *$P < 0.05$, **$P < 0.01$, ***$P < 0.001$, ****$P < 0.0001$. In all graphs, results were shown as mean ± SD, and details for each graph were listed in the corresponding figure's legend.

### Online supplemental material
Fig. S1 shows that UBAP2L shuttles between cytoplasm and nucleus and regulates Nups localization in G1-arrested cells. Fig. S2 demonstrates that UBAP2L may inhibit formation of cytoplasmic AL. Fig. S3 shows that UBAP2L regulates localization of FXRP proteins in the cytoplasm. Fig. S4 provides evidence that 98–430 aa fragment of UBAP2L protein is required for the function of UBAP2L on Nups and FXR1. Fig. S5 shows that UBAP2L regulates FXRP proteins and promotes survival of HeLa cells. Table S1 describes the cloning primers used in the study. Table S2 describes other reagents and resources including bacterial stains, cell lines, chemicals, cDNAs, and software used in the study.

### Data availability
All data needed to evaluate the conclusions in the paper are present in the paper and/or the online supplemental material.

## Acknowledgments
We thank members of the I. Sumara and R. Ricci laboratories for helpful discussions on the manuscript and IGBMC facilities, especially the imaging center for their help.

Y. Liao was supported by PhD fellowship from the China Scholarship Council (CSC) and postdoctoral fellowship from SATT Conectus Alsace. X. Liu, J. Lin, M. Qu, and L. Ran were supported by PhD fellowships from the CSC, and A. Agote-Aran and L. Guerber were supported by PhD fellowships from the

IMC-Bio graduate school. E. Pangou was supported by postdoctoral fellowships from the Foundation pour la recherché Médicale, ANR-10-LABX-0030-INRT, and Fondation ARC pour la recherche sur le cancer Passerelle program. L. Andronov and B.P. Klaholz acknowledge support by Institut National du Cancer (INCa) and by the French Infrastructure for Integrated Structural Biology ANR-10-INSB-05-01, Instruct-ERIC, and iNEXT-Discovery. Research in the M. Gotta laboratory was supported by a grant from Schweizerischen Nationalfonds (310030_204267). Research in the I. Sumara laboratory was supported by a grant ANR-10-LABX-0030-INRT, a French State fund managed by the Agence Nationale de la Recherche (ANR) under the frame program Investissements d'Avenir ANR-10-IDEX-0002-02, Institut de génétique, biologie moléculaire et cellulaire, Centre National de la Recherche Scientifique, Australian Research Council, INCa (PLBIO 2022-082), ANR (AAPG2022, NICE4Nups), Sanofi iAward Europe and Programme Fédérateur Aviesan, Plan Cancer, and national collaborative project "NANOTUMOR." Open Access funding provided by Université de Genève.

Author contributions: Y. Liao and L. Andronov designed and performed experiments and helped write the manuscript. X. Liu, J. Lin, L. Guerber, L. Lu, A. Agote-Aran, E. Pangou, L. Ran, C. Kleiss, M. Qu, S. Schmucker, and L. Cirillo performed experiments. Z. Zhang helped perform experiments. D. Riveline, M. Gotta, and B.P. Klaholz helped design the experiments and supervise. I. Sumara supervised the project, designed experiments, and wrote the manuscript with input from all authors.

Disclosures: The authors declare no competing interests exist.

Submitted: 3 October 2023

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

# Supplemental material

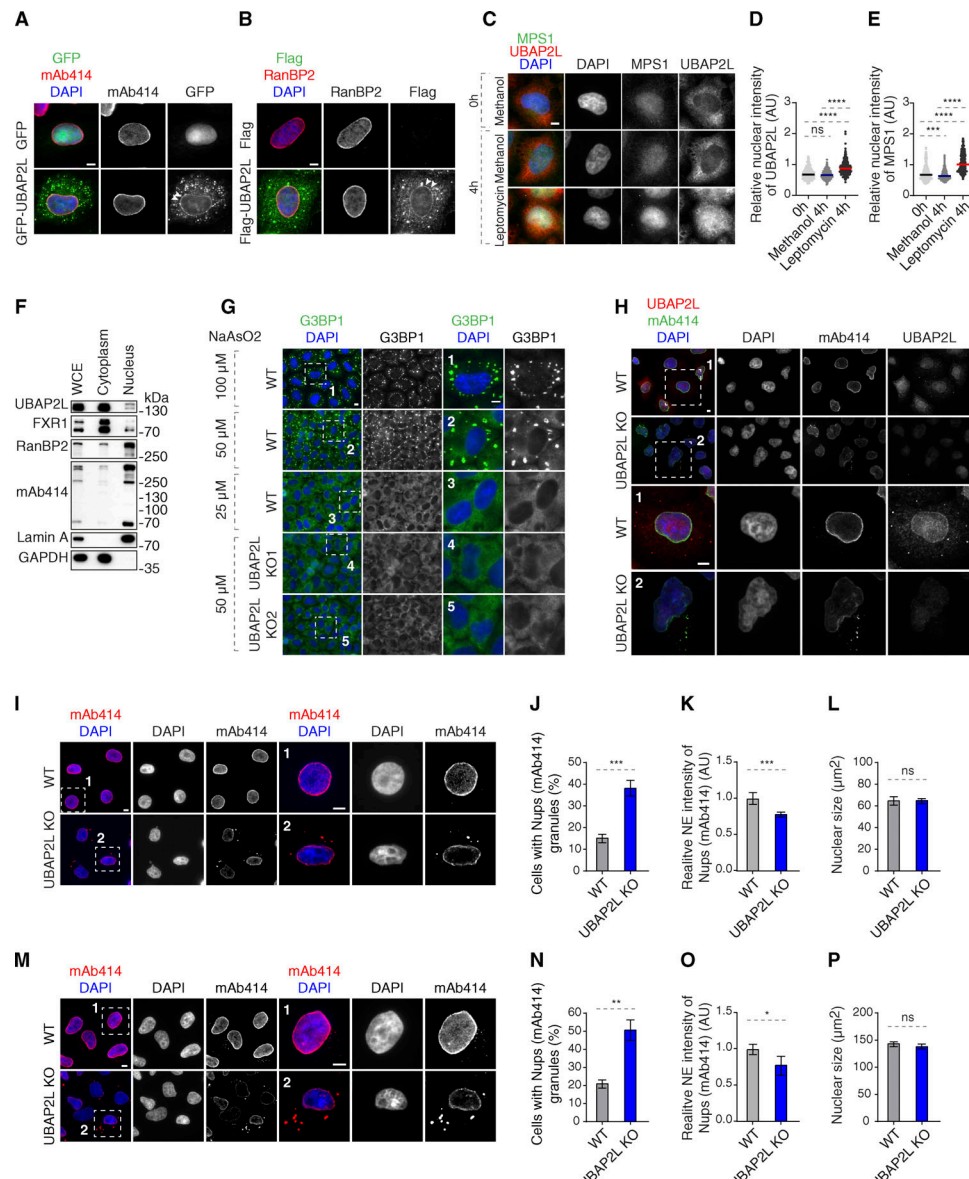

Figure S1.  **UBAP2L shuttles between cytoplasm and nucleus. (A)** Representative immunofluorescence images depicting the localization of UBAP2L and Nups (mAb414) in HeLa cells expressing GFP alone or GFP-UBAP2L. The arrowheads indicate the NE localization of GFP-tagged UBAP2L. Scale bar, 5 μm. **(B)** Representative immunofluorescence images depicting the localization of UBAP2L and Nups (RanBP2) in HeLa cells expressing Flag alone or Flag-UBAP2L. The arrowheads indicate the NE localization of Flag-tagged UBAP2L. Scale bar, 5 μm. **(C–E)** Representative immunofluorescence images depicting the cytoplasmic and nuclear localization of UBAP2L and MPS1 (also known as protein kinase TTK) after treatment with the Leptomycin B (inhibitor of nuclear export factor Exportin 1) (100 ng/ml) for 4 h. Nuclei were stained with DAPI. Scale bar, 5 μm (C). The relative nuclear intensity (AU) of UBAP2L (D) and MPS1 (E) shown in C was quantified. At least 150 cells per condition were analyzed (mean ± SD, ns: not significant, ***P < 0.001, ****P < 0.0001, unpaired two-tailed *t* test, *n* = 3 independent experiments). **(F)** Protein levels of UBAP2L, FXR1, and Nups were analyzed by western blot in the whole cell extract (WCE) and in nuclear and cytoplasmic fractions of HeLa cells. **(G)** Representative immunofluorescence images of WT and UBAP2L KO HeLa cells depicting formation of SGs labeled by G3BP1 at indicated arsenite concentrations. The magnified framed regions are shown in the corresponding numbered panels. Nuclei were stained with DAPI. Scale bars, 5 μm. **(H)** Representative immunofluorescence images depicting the localization of Nups and UBAP2L in asynchronously proliferating WT and UBAP2L KO HeLa cells visualized by mAb414 and UBAP2L antibodies. Nuclei were stained with DAPI. The magnified framed regions are shown in the corresponding numbered panels. Note that UBAP2L signal is absent in UBAP2L-deleted cells. Scale bars, 5 μm. **(I–L)** Representative immunofluorescence images depicting the localization and NE intensity of Nups (mAb414) and nuclear size in WT and UBAP2L KO HeLa cells synchronized in G1 phase by lovastatin (10 μM) for 16 h. Nuclei were stained with DAPI. The magnified framed regions are shown in the corresponding numbered panels. Scale bars, 5 μm (I). The cells with Nups (mAb414) granules (J), the NE intensity of Nups (mAb414) (K), and the nuclear size (L) shown in I were quantified. At least 150 cells per condition were analyzed (mean ± SD, ns: not significant, ***P < 0.001, unpaired two-tailed *t* test, *n* = 4 independent experiments). **(M–P)** Representative immunofluorescence images depicting the localization and NE intensity of Nups (mAb414) and nuclear size in WT and UBAP2L KO HeLa cells synchronized in G0/G1 phase by Psoralidin (5 μM) for 24 h. Nuclei were stained with DAPI. The magnified framed regions are shown in the corresponding numbered panels. Scale bars, 5 μm (M). The cells with Nups (mAb414) granules (N), the NE intensity of Nups (mAb414) (O), and the nuclear size (P) shown in M were quantified. At least 200 cells per condition were analyzed (mean ± SD, ns: not significant, *P < 0.05, **P < 0.01, unpaired two-tailed *t* test, *n* = 3 independent experiments). Source data are available for this figure: SourceData FS1.

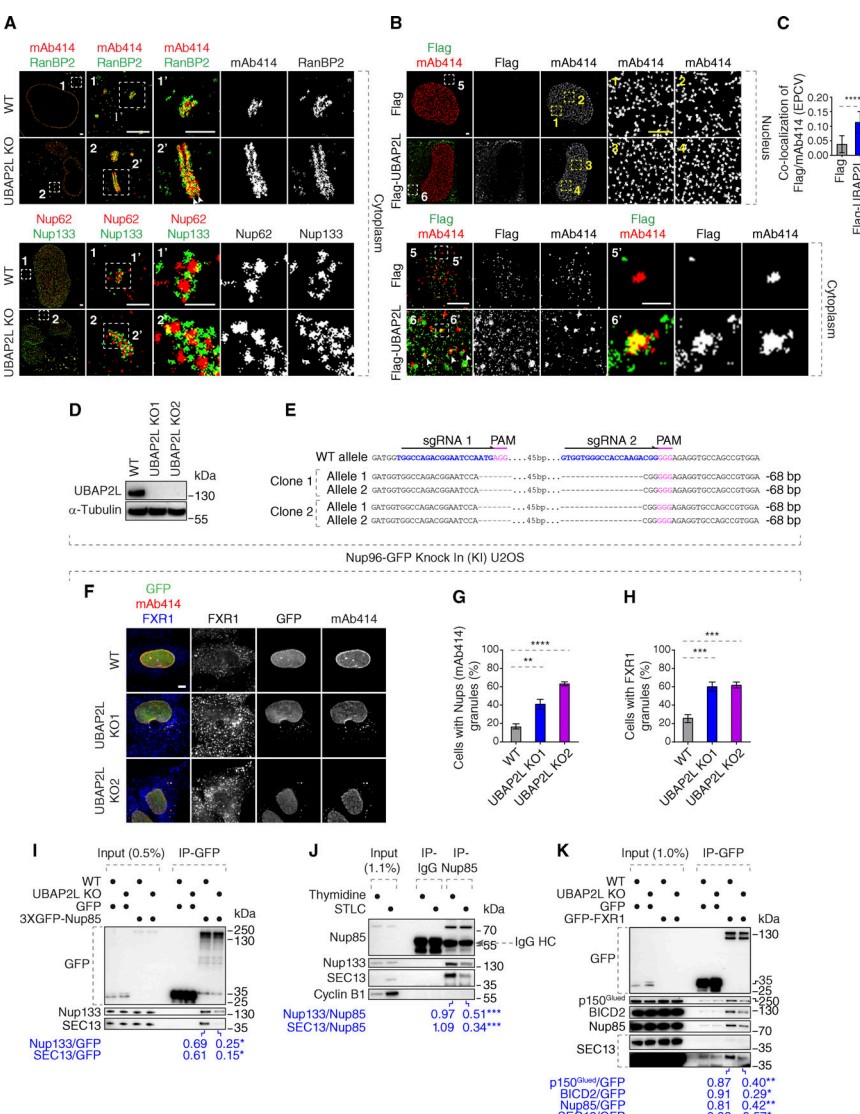

Figure S2. **UBAP2L may inhibit formation of cytoplasmic AL or AL-like Nup assemblies. (A)** Representative splitSMLM immunofluorescence images depicting the localization of NPC components corresponding to the central channel (Nups labeled by mAb414) and cytoplasmic filaments (RanBP2) at the NE and in the cytoplasm, as well as the localization of NPC components corresponding to the central channel (FG-Nup Nup62) and the outer ring (Y-complex Nup133) in the cytoplasm in WT and UBAP2L KO HeLa cells synchronized in interphase by DTBR at 12 h. Note that unlike at the NE where RanBP2 can localize exclusively to the cytoplasmic side of the NPCs (Fig. 5 A), deletion of UBAP2L leads to the accumulation of the Nup assemblies in the cytoplasm with a symmetric distribution of RanBP2. Moreover, similar to the nuclear surface, in the cytoplasm, Nup62 signal is surrounded by Nup133 ring-like structures in both WT and UBAP2L KO cells. The magnified framed regions are shown in the corresponding numbered panels. Scale bars, 1,000, 300, and 150 nm, respectively. **(B and C)** Representative SMLM immunofluorescence images of FG-Nups (mAb414) at the nuclear surface and in the cytoplasm in interphase HeLa cells expressing Flag alone or Flag-UBAP2L for 35 h and synchronized by DTBR at 12 h. The magnified framed regions are shown in the corresponding numbered panels and corresponding quantification is shown in Fig. 5 C. The arrowheads indicate the cytoplasmic co-localization of FLAG-UBAP2L and mAb414-reactive Nups, which were highlighted in the corresponding magnified regions. Scale bars, 1,000 and 500 nm, respectively (B). The colocalization (EPCV, events per cell cytoplasmic view) of cytoplasmic mAb414 with Flag and Flag-UBAP2L in B was measured by CellProfiler (mean ± SD, ****P < 0.0001, unpaired two-tailed t test; counted 35 cells for Flag and 32 cells for Flag-UBAP2L) (C). **(D and E)** Validation of CRISPR/Cas9-mediated UBAP2L KO Nup96-GFP KI U2OS cell clones by western blot (D) and Sanger sequencing (E). **(F–H)** Representative immunofluorescence images of the localization of Nups (GFP-Nup96 and mAb414) and FXR1 in WT and in two UBAP2L KO Nup96-GFP KI U2OS clonal cell lines in interphase cells synchronized by DTBR at 15 h. Nuclei were stained with DAPI. Scale bar, 5 μm (F). The percentage of cells with cytoplasmic granules of Nups (mAb414) (G) and of FXR1 (H) shown in F were quantified. At least 200 cells per condition were analyzed (mean ± SD, **P < 0.01, ***P < 0.001, ****P < 0.0001, unpaired two-tailed t test, n = 3 independent experiments). **(I)** Lysates of WT and UBAP2L KO Hela cells expressing GFP alone or 3XGFP-Nup85 for 27 h and synchronized in G1/S phase by Thymidine 16 h were immunoprecipitated using agarose GFP-Trap A beads (GFP-IP), analyzed by western blot, and signal intensities were quantified (shown a mean value, *P < 0.05, unpaired two-tailed t test; n = 3 independent experiments). **(J)** HeLa cells lysates of cells synchronized in interphase (Thymidine 16 h) and of cells synchronized in mitosis (STLC 16 h) were immunoprecipitated using Nup85 antibody or IgG, analyzed by western blot, and signal intensities were quantified (shown a mean value, ***P < 0.001, unpaired two-tailed t test; n = 3 independent experiments). **(K)** Lysates of interphase WT and UBAP2L KO HeLa cells expressing GFP alone or GFP-FXR1 for 27 h were immunoprecipitated using agarose GFP-Trap A beads (GFP-IP), analyzed by western blot, and signal intensities were quantified (shown a mean value, *P < 0.05, **P < 0.01, unpaired two-tailed t test; n = 3 independent experiments). Source data are available for this figure: SourceData FS2.

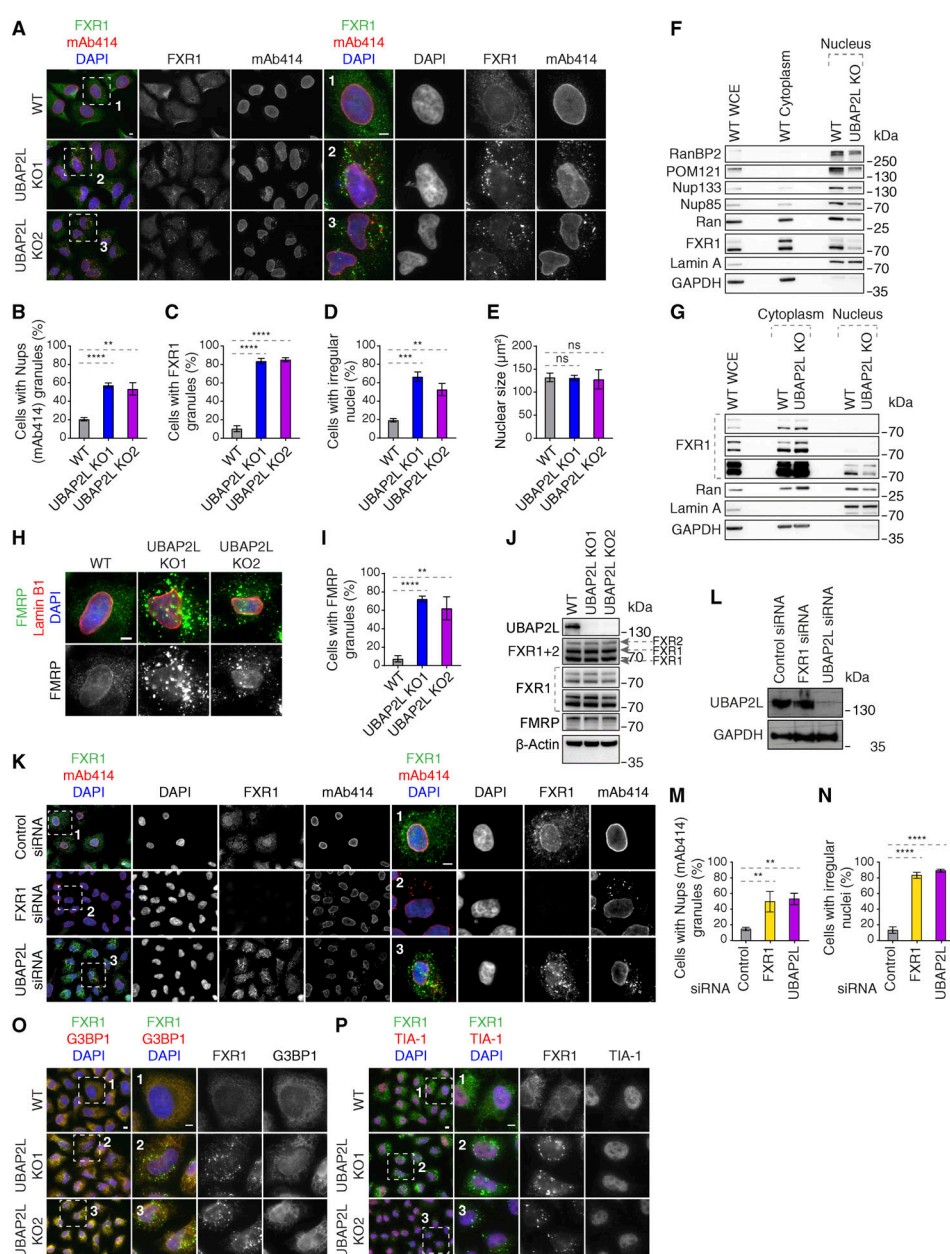

Figure S3. **UBAP2L regulates FXRP proteins in the cytoplasm. (A–E)** Representative immunofluorescence images depicting the nuclear shape and localization of Nups (mAb414) and FXR1 in WT and UBAP2L KO HeLa cells in interphase cells synchronized by DTBR at 12 h. The magnified framed regions are shown in the corresponding numbered panels. Nuclei were stained with DAPI. Scale bars, 5 μm (A). The percentage of cells with cytoplasmic granules of Nups (mAb414) (B) and of FXR1 (C) and with irregular nuclei (D) and the nuclear size (E) shown in A were quantified. At least 250 cells per condition were analyzed (mean ± SD, ns: non-significant, **P < 0.01, ***P < 0.001, ****P < 0.0001, unpaired two-tailed t test, n = 3 independent experiments). **(F)** The nuclear and cytoplasmic protein levels of Nups and NPC transport-associated factors in WT and UBAP2L KO HeLa cells synchronized as in A were analyzed by western blot. WCE indicates whole cell extract. **(G)** The nuclear and cytoplasmic protein levels of Nups and NPC transport-associated factors in asynchronously proliferating WT and UBAP2L KO HeLa cells were analyzed by western blot. WCE indicates whole cell extract. **(H and I)** Representative immunofluorescence images depicting the localization of FMRP and Lamin B1 in WT and UBAP2L KO HeLa cells synchronized in interphase by DTBR at 12 h. Nuclei were stained with DAPI. Scale bar, 5 μm (H). The percentage of cells with the cytoplasmic granules containing FMRP shown in H was quantified (I). At least 200 cells per condition were analyzed (mean ± SD, **P < 0.01, ****P < 0.0001, unpaired two-tailed t test, n = 3 independent experiments). **(J)** The protein levels of FXRP proteins in WT and UBAP2L KO HeLa cells synchronized in interphase by DTBR at 12 h were analyzed by western blot. **(K–N)** Representative immunofluorescence images depicting localization of FXR1 and Nups (mAb414) and the nuclear shape in the HeLa cells treated with indicated siRNAs and synchronized in interphase by DTBR at 12 h. The magnified framed regions are shown in the corresponding numbered panels. Nuclei were stained with DAPI. Scale bars, 5 μm (K). UBAP2L protein levels in K were analyzed by western blot (L). The percentage of cells with the cytoplasmic granules of Nups (mAb414) (M) and irregular nuclei (N) shown in K were quantified. At least 200 cells per condition were analyzed (mean ± SD, **P < 0.01, ****P < 0.0001, unpaired two-tailed t test, n = 3 independent experiments). **(O and P)** Representative immunofluorescence images of WT and UBAP2L KO HeLa cells synchronized in interphase by DTBR at 12 h under non-stress conditions depicting localization of FXR1 (O and P), G3BP1 (O), and TIA-1 (P). The magnified framed regions are shown in the corresponding numbered panels. Nuclei were stained with DAPI. Scale bars, 5 μm. Note that FXR1-containing granules present in non-stressed UBAP2L KO HeLa cells do not colocalize with SG components. Source data are available for this figure: SourceData FS3.

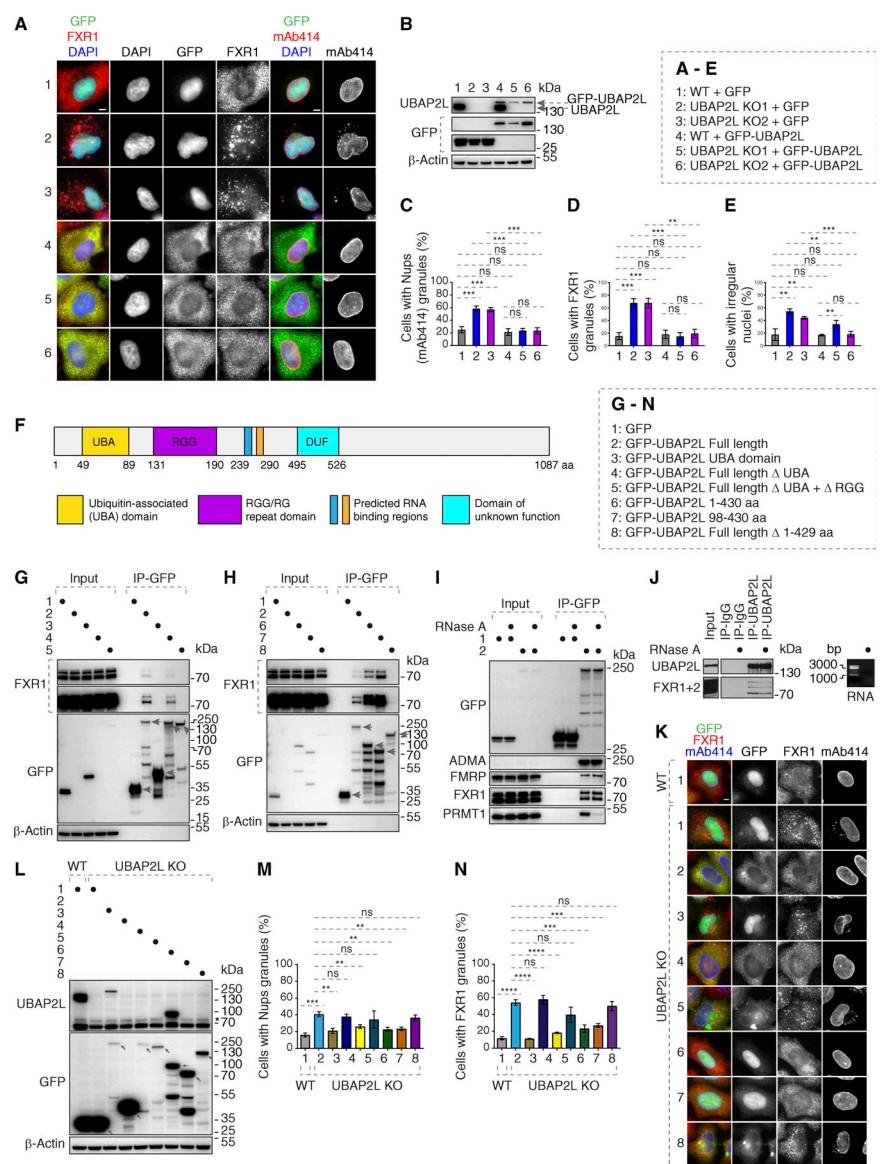

Figure S4. **98–430 aa fragment of UBAP2L protein is required for the function of UBAP2L on Nups and FXR1. (A–E)** Representative immunofluorescence images depicting the nuclear shape and localization of FXR1 and Nups (mAb414) in WT and UBAP2L KO HeLa cells expressing GFP alone or GFP-UBAP2L for 60 h and synchronized in interphase by DTBR at 12 h. Nuclei were stained with DAPI. Scale bars, 5 μm. Note that ectopic expression of GFP-UBAP2L but not GFP can rescue the nuclear and localization phenotypes in both UBAP2L KO HeLa cell lines (A). The protein levels of endogenous UBAP2L, GFP, and GFP-UBAP2L of cells shown in A were analyzed by western blot (B). The percentage of cells with the cytoplasmic granules of Nups (mAb414) (C) and of FXR1 (D) and with irregular nuclei (E) shown in A were quantified. At least 200 cells per condition were analyzed (mean ± SD, ns: not significant, **P < 0.01, ***P < 0.001, unpaired two-tailed *t* test, *n* = 3 independent experiments). **(F)** Domain organization of UBAP2L depicting UBA domain, RGG domain, two predicted RNA binding regions, and the domain of unknown function (DUF). **(G and H)** Lysates of HeLa cells expressing GFP alone or GFP-UBAP2L–derived constructs (full-length [FL], UBA, ΔUBA, or Δ(UBA+RGG) fragments) for 27 h were immunoprecipitated using agarose GFP-Trap A beads (GFP-IP) and analyzed by western blot (G). Lysates of HeLa cells expressing GFP alone or several GFP-UBAP2L–derived constructs (FL, 1–430 aa, 98–430 aa, or Δ1–429 aa fragments) for 27 h were immunoprecipitated using agarose GFP-Trap A beads (GFP-IP) and analyzed by western blot (H). The arrows indicate the bands corresponding to the expressed GFP proteins while the remaining bands are non-specific. **(I and J)** Interphase HeLa cells expressing GFP alone or GFP-UBAP2L for 27 h and cell lysates were treated with RNase A, immunoprecipitated using agarose GFP-Trap A beads (GFP-IP), and analyzed by western blot. Note that RNase treatment can abolish interaction with PRMT1 but not with FXRPs (I). IPs from cell lysates of HeLa cells treated with RNase A using UBAP2L antibody or IgG were analyzed by western blot. The efficiency of the RNase treatment was confirmed by imaging of mRNAs by agarose gel electrophoresis and ethidium bromide staining (J). **(K–N)** Representative immunofluorescence images depicting localization of FXR1 and Nups (mAb414) in WT and UBAP2L KO HeLa cells expressing GFP alone or GFP-UBAP2L-derived fragments (FL, UBA, ΔUBA, Δ(UBA+RGG), 1–430 aa, 98–430 aa, or Δ1–429 aa) for 60 h and synchronized in interphase by DTBR at 12 h. Scale bar, 5 μm (K). Note that the UBAP2L 98–430 aa protein fragment containing the RGG domain is required for the function of UBAP2L on Nups and FXR1. The protein levels of endogenous UBAP2L, GFP, and GFP-UBAP2L-derived versions (FL, UBA, ΔUBA, Δ(UBA+RGG), 1–430 aa, 98–430 aa, or Δ1–429 aa) of cells shown in K were analyzed by western blot. The arrows indicate the bands corresponding to the expressed GFP proteins while the remaining faster migrating bands are either non-specific or degradation products (L). The percentage of cells with the cytoplasmic granules of Nups (mAb414) (M) and of FXR1 (N) shown in K were quantified. At least 200 cells per condition were analyzed (mean ± SD, ns: not significant, **P < 0.01, ***P < 0.001, ****P < 0.0001, unpaired two-tailed *t* test, *n* = 3 independent experiments). Source data are available for this figure: SourceData FS4.

Liao et al.
Nuclear pore complex homeostasis in interphase

Journal of Cell Biology    S5

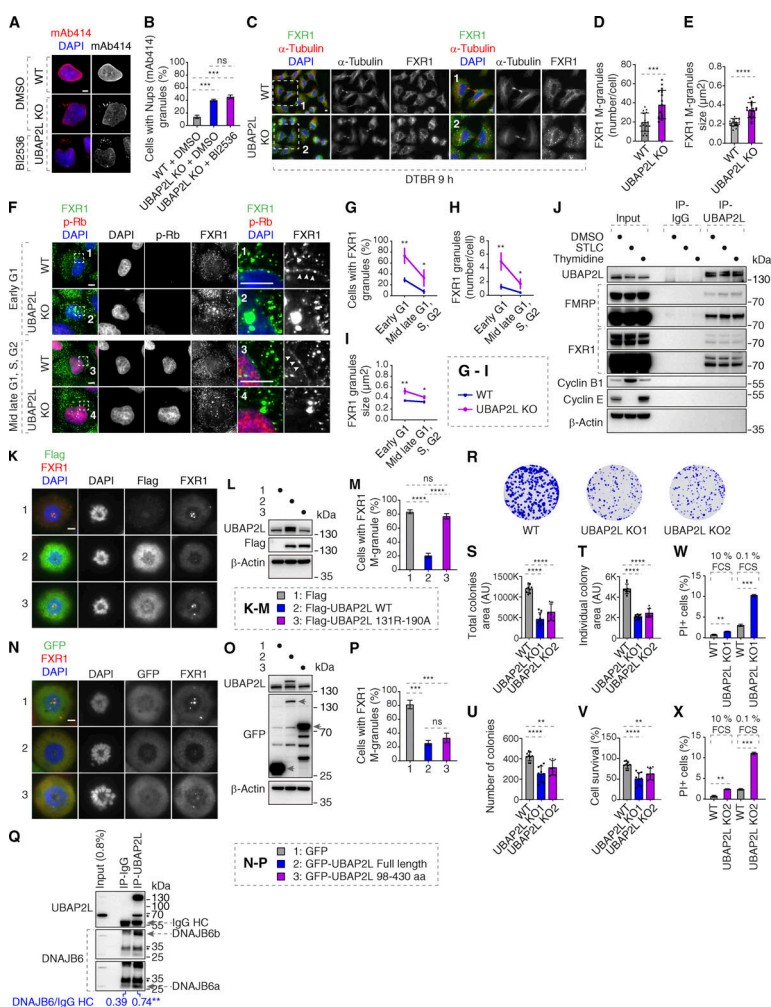

**Figure S5.** **UBAP2L regulates FXRP proteins and promotes survival of HeLa cells. (A and B)** Representative immunofluorescence images depicting the localization of Nups (mAb414) in WT and UBAP2L KO HeLa cells synchronized in interphase by DTBR at 12 h. PLK1 inhibitor BI 2536 (or solvent control) was used at a concentration of 100 nM for 45 min prior to sample collection. Nuclei were stained with DAPI. Scale bar, 5 µm (A). The percentage of cells with the cytoplasmic granules containing Nups (mAb414) shown in A was quantified (B). At least 150 cells per condition were analyzed (mean ± SD, ns: not significant, ***P < 0.001, unpaired two-tailed t test, n = 3 independent experiments). **(C–E)** Representative immunofluorescence images depicting the localization of FXR1 in WT and UBAP2L KO HeLa cells synchronized by DTBR 9 h in late telophase. Nuclei were stained with DAPI. The magnified framed regions are shown in the corresponding numbered panels. Scale bars, 5 µm (C). The number of FXR1 granules per cell (number/cell) (D) and the size of FXR1 granules (granule ≥ 0.105 µm²) (E) shown in C were quantified (mean ± SD, ***P < 0.001, ****P < 0.0001, unpaired two-tailed t test. 17 WT and 18 UBAP2L KO HeLa cells were counted, respectively). **(F–I)** Representative immunofluorescence images depicting the localization of FXR1 in different cell cycle stages in asynchronously proliferating WT and UBAP2L KO HeLa cells. p-Rb was used to distinguish between early G1 (p-Rb–negative cells) and mid-late G1, S, and G2 (p-Rb–positive cells) stages. Nuclei were stained with DAPI. The arrowheads indicate the NE localization of endogenous FXR1. Scale bars, 5 µm (F). The percentage of cells with cytoplasmic FXR1 granules (G), the number of FXR1 granules per cell (number/cell) (H), and the size of FXR1 granules (granule ≥ 0.2109 µm²) (I) shown in F were quantified. At least 200 cells per condition were analyzed (mean ± SD, *P < 0.05, **P < 0.01, unpaired two-tailed t test, n = 3 independent experiments). **(J)** IPs from HeLa cells lysates of asynchronously proliferating cells (DMSO 16 h), cells synchronized in mitosis (STLC 16 h) or in interphase (thymidine 16 h) using UBAP2L antibody or IgG were analyzed by western blot. **(K–M)** HeLa cells expressing Flag, Flag-UBAP2L WT, or Flag-UBAP2L R131–190A for 27 h were synchronized in prometaphase using STCL for 16 h and representative immunofluorescence images depicting localization of FXR1 are shown in K. Chromosomes were stained with DAPI. Scale bar, 5 µm. The protein levels of Flag-UBAP2L and endogenous UBAP2L in K were analyzed by western blot (L). The percentage of cells with FXR1-granules shown in K was quantified (M). At least 200 cells per condition were analyzed (mean ± SD, ns: not significant, ****P < 0.0001, unpaired two-tailed t test, n = 3 independent experiments). **(N–P)** Representative immunofluorescence images depicting the localization of FXR1 in HeLa cells expressing GFP, GFP-UBAP2L FL, or GFP-UBAP2L 98–430 aa for 27 h synchronized in prometaphase using STCL for 16 h. Chromosomes were stained with DAPI. Scale bar, 5 µm (N). The protein levels of GFP-UBAP2L and endogenous UBAP2L in N were analyzed by western blot (O). The percentage of cells with FXR1-granules shown in N was quantified (P). At least 200 cells per condition were analyzed (mean ± SD, ns: not significant, ***P < 0.001, unpaired two-tailed t test, n = 3 independent experiments). **(Q)** HeLa cells lysates were immunoprecipitated using UBAP2L antibody or IgG, analyzed by western blot, and signal intensities were quantified (shown a mean value, **P < 0.01, unpaired two-tailed t test; N = 3). The arrows indicate the bands corresponding to the IgG heavy chain (HC). **(R–V)** Representative images of colony formation assays of WT and UBAP2L KO HeLa cells maintained in culture for 7 days (R). Total colony area (S), individual colony area (T), average number of colonies (U), and cell survival (V) of cells shown in R were quantified using the Fiji software (mean ± SD, **P < 0.01, ****P < 0.0001; one-way ANOVA, n = 3 independent experiments). **(W and X)** The percentage of PI-positive cells in WT and UBAP2L KO HeLa cells cultured in the indicated concentrations of serum for 72 h were quantified by fluorescence activated cell sorting (mean ± SD, **P < 0.01, ***P < 0.001, unpaired two-tailed t test, n = 3 independent experiments). Source data are available for this figure: SourceData FS5.

Provided online are two tables. Table S1 describes the cloning primers used in the study. Table S2 describes other reagents and resources including bacterial stains, cell lines, chemicals, cDNAs, and software used in the study.

