## [Peer Review File · The Journal of Cell Biology]

UBAP2L ensures homeostasis of nuclear pore complexes at the intact nuclear envelope

Yongrong Liao, Leonid Andronov, Xiaotian Liu, Junyan Lin, Lucile Guerber, Linjie Lu, Arantxa Agote-Aran, Evanthia Pangou, Li Ran, Charlotte Kleiss, Mengdi Qu, Stephane Schmucker, Luca Cirillo, Zhirong Zhang, Daniel Riveline, Monica Gotta, Bruno Klaholz, and Izabela Sumara

Corresponding Author(s): Izabela Sumara, IGBMC

Review Timeline:

Submission Date:	2023-10-03
Editorial Decision:	2023-11-17
Revision Received:	2024-02-15
Editorial Decision:	2024-03-02
Revision Received:	2024-03-08

Monitoring Editor: Michael Rout

Scientific Editor: Dan Simon

Transaction Report:

DOI: <https://doi.org/10.1083/jcb.202310006>

November 17, 2023

Re: JCB manuscript #202310006

Dr. Izabela Sumara
IGBMC
1, rue Laurent Fries
ILLKIRCH 67400
France

Dear Dr. Sumara,

Thank you for submitting your manuscript entitled "UBAP2L drives scaffold assembly of nuclear pore complexes at the intact nuclear envelope." The manuscript was assessed by expert reviewers, whose comments are appended to this letter. We invite you to submit a revision if you can address the reviewers' key concerns, as outlined here.

You will see that overall, the reviewers are enthusiastic about your work and feel that the fundamental conclusion that UBAP2L is crucial for nuclear pore complex assembly and stability is well-founded and as such the manuscript will represent a valuable addition to the Journal of Cell Biology, after appropriate additions and edits. The reviewers agree that the biochemistry and microscopy in the manuscript is of high quality, utilizing state-of-the-art methods such as single molecule and super-resolution imaging, CRISPR editing, and molecular interaction analysis. The data is meticulously quantified and well-presented, and despite its length, the manuscript is readable with clear, concise conclusions that accurately reflect the data. The quantification is thorough and supports publication. However, some conclusions lack direct biochemical evidence, making them somewhat speculative. Reviewer #1 asks for more definitive evidence to support the conclusion that UBAP2L functions specifically in de novo NPC assembly and not in NPC repair. We agree that this is an important issue that must be resolved with new data. Reviewer #1 also notes that the claim that UBAP2L mediates Y-complex formation and interaction with Nup153 is not well supported and should be toned down in the absence of biochemical evidence. This is an interesting question, but we agree with the reviewer that new experiments to address this are not essential for this paper. The remaining reviewer comments ask for assays of an inner ring Nup, additional controls, and for a more thorough integration of the findings with existing literature on NPC assembly. These should also be addressed in full.

GENERAL GUIDELINES:

Text limits: Character count for an Article is < 40,000, not including spaces. Count includes title page, abstract, introduction, results, discussion, and acknowledgments. Count does not include materials and methods, figure legends, references, tables, or supplemental legends.

Figures: Articles may have up to 10 main text figures. Figures must be prepared according to the policies outlined in our Instructions to Authors, under Data Presentation, <https://jcb.rupress.org/site/misc/ifora.xhtml>. All figures in accepted manuscripts will be screened prior to publication.

Supplemental information: There are strict limits on the allowable amount of supplemental data. Articles may have up to 5 supplemental figures. Up to 10 supplemental videos or flash animations are allowed. A summary of all supplemental material should appear at the end of the Materials and methods section.

Please note that JCB now requires authors to submit Source Data used to generate figures containing gels and Western blots with all revised manuscripts. This Source Data consists of fully uncropped and unprocessed images for each gel/blot displayed in the main and supplemental figures. Since your paper includes cropped gel and/or blot images, please be sure to provide one Source Data file for each figure that contains gels and/or blots along with your revised manuscript files. File names for Source Data figures should be alphanumeric without any spaces or special characters (i.e., SourceDataF#, where F# refers to the associated main figure number or SourceDataFS# for those associated with Supplementary figures). The lanes of the gels/blots should be labeled as they are in the associated figure, the place where cropping was applied should be marked (with a box), and molecular weight/size standards should be labeled wherever possible.

Source Data files will be made available to reviewers during evaluation of revised manuscripts and, if your paper is eventually

published in JCB, the files will be directly linked to specific figures in the published article.

The typical timeframe for revisions is three to four months. While most universities and institutes have reopened labs and allowed researchers to begin working at nearly pre-pandemic levels, we at JCB realize that the lingering effects of the COVID-19 pandemic may still be impacting some aspects of your work, including the acquisition of equipment and reagents. Therefore, if you anticipate any difficulties in meeting this aforementioned revision time limit, please contact us and we can work with you to find an appropriate time frame for resubmission. Please note that papers are generally considered through only one revision cycle, so any revised manuscript will likely be either accepted or rejected.

Thank you for this interesting contribution to Journal of Cell Biology. You can contact us at the journal office with any questions at cellbio@rockefeller.edu.

Sincerely,

Michael Rout, PhD
Monitoring Editor
Journal of Cell Biology

Dan Simon, PhD
Scientific Editor
Journal of Cell Biology

Reviewer #1 (Comments to the Authors (Required)):

Uncovering mechanisms governing nuclear pore complex (NPC) assembly has been a central pursuit of the nuclear transport field over several decades. Here, Liao et al. implicate UBAP2L as a potential factor that contributes to de novo NPC biogenesis in interphase. The key data supporting such a role includes super-resolution microscopy demonstrating the localization of UBAP2L at least a subset of NPCs, IP experiments demonstrating biochemical interactions with nups and loss of function experiments that lead to a reduction in NPC density at the nuclear envelope alongside the accumulation of nups in cytosolic aggregates. As IP experiments in the context of UBAP2L deletion result in a reduction of Y-complex components affinity purified with binding partners, there is an effort to establish that UBAP2L plays a role in "driving" the assembly of the Y-complex in the cytoplasm. Similar efforts try to connect UBAP2L to FXR1, another factor this group has suggested plays a role in NPC assembly. Although in general the data presented are of high quality and rigorously quantified, the data supporting the latter two conclusions is open to interpretation due to a lack of direct biochemical evidence supporting any physical interactions. Further, the manuscript is a bit challenging to navigate as it is quite long, there are many supplemental figures, and there is a tendency to veer off topic at the end of the work. Nonetheless, the core conclusion that UBAP2L plays a role in NPC assembly and/or stability is on solid footing and will make an important contribution appropriate for JCB.

Major Points

1) There have been many genetic perturbations described in the literature where nups are mislocalized but the key challenge is to define whether mislocalized nups are a product of defective assembly or NPC instability. There is no definitive evidence presented in this work that UBAP2L specifically impacts NPC assembly or that the observed cytosolic aggregates are an intermediate in NPC assembly. In fact, as the cycloheximide experiment presented in Figure 9 results in the persistence of aggregates, it suggests that they may be derived from assembled NPCs. This experiment should be performed in the context of the siRNA mediated knockdown of UBAP2L under unstressed conditions. In lieu of more definitive data establishing a role for UBAP2L in de novo NPC assembly, the title and conclusions should be amended to reflect this uncertainty as to UBAP2L's role.

2) The model that UBAP2L "drives the formation of Y complex" or that "UBAP2L mediates the interaction of Y complex Nups with Nup153" should be clearly written as being speculative as there is no direct data supporting these assertions. Instead, this conclusion rests on IP data in Figure 5G and H where overexpressed GFP-tagged nups are affinity purified with lower nup levels of other nups in the UBAP2L knockout. This is interpreted as suggesting that UBAP2L contributes to Y complex assembly. A more likely interpretation is that the GFP-tagged nups are mislocalized to the cytosolic aggregates where they may not robustly

interact with nup partners. To be able to conclude that there is an active "driving" role for UBAP2L in forming nup subcomplexes, direct biochemical reconstitution experiments would be required that would be beyond the scope of the current study.

3) General lack of scholarship. NPC assembly has been investigated for over 30 years in many models including yeast and *Xenopus* and mammalian cell lines. There is limited appreciation of this deep literature by the authors in particular in the introduction where there is an emphasis on their prior work. The strong suggestion is to include at least a discussion of the other assembly factors that have been identified including nuclear transport receptors/Ran, membrane remodeling proteins, lipid biosynthetic factors and chaperones/AAA proteins like Torsin. Such an effort might also lead to a better integration of UBAP2L into the known NPC assembly pathway and may provide a more critical lens to evaluate their proposed model. It might also mean more consideration of alternative models like, for example, that UBAP2L is engaging with NTRs through its RGG motif. As NTRs are well established NPC assembly factors and play roles as chaperones in modulating biomolecular condensation, this analysis may provide fruitful avenues for future work.

Minor Points:

- 1) Figure 1: Whether or not UBA2L associates with all or just a subset of NPCs is important for interpreting its function in, for example, the nuclear transport experiments. Can the authors provide these data? And if not, the caveats for providing this information should be discussed.
- 2) Figure 2: There is no sense of the complexity of the bound fractions of these IPs as only Western blots are used. Either this data (e.g. a coomassie stain) should be presented or a more explicit evaluation of non-specific interactions with abundant proteins would help.
- 3) Figure 3 and others - there is no investigation of any inner ring nups. As these are central to NPC assembly, testing the localization/binding of at least one would support a more comprehensive analysis.
- 4) Figure 4: Can the authors interpret why the Aurora B staining is so different in the UBAP2L Kos for the reader?
- 5) Figure 5: That there might be differences in the rotational symmetry of NPCs in the UBAP2L KO is interesting but may not make much sense. As it is often difficult to be certain whether there is comprehensive labeling or consistent blinking of fluorophores with SMS approaches, some caveats to these data should be introduced. Ultimately, ultrastructural approaches would be needed to be definitive here.
- 6) There are many FG-nups that are not recognized by mAb414. Instead of using "FG-nups" in the text the term "414-reactive nups" or something equivalent would be more accurate.

Reviewer #2 (Comments to the Authors (Required)):

The manuscript is well-suited for JCB.

The quality of work is excellent. Adds to the field and is a stimulating set of work.

I believe it is worthy of acceptance.

Title: UBAP2L drives scaffold assembly of nuclear pore complexes at the intact nuclear envelope

Authors claim to have uncovered a direct role of ubiquitin-associated protein 2-like (UBAP2L) in the biogenesis of adequately organized and functional NPCs at the intact nuclear envelope (NE) in human cells. UBAP2L-mediated biogenesis of NPCs ensures nuclear transport, adaptation to nutrient stress, and cellular proliferation.

UBAP2L localizes to the nuclear pores and drives the formation of the Y-complex, an essential scaffold component of the NPC, and its localization to the NE. UBAP2L localizes to the NPCs and interacts with Nups and NPC assembly factors. Taken together, the interaction of UBAP2L with Y-complex Nups, as well as with the nuclear and cytoplasmic NPC assembly factors, suggests a possible function of UBAP2L on Nups assembly and NPC biogenesis.

UBAP2L facilitates the interaction of the Y-complex with POM121 and Nup153, the critical upstream factors in a well-defined sequential order of Nups assembly onto NE during interphase. UBAP2L and BAP2L regulate the localization of the Nup transporting factor FXR1. UBAP2L regulates Nups localization but not in postmitotic cells.

The duality of UBAP2L is that it localizes to the NE, and NPCs drive the formation of the Y-complex. They can chaperon FXRP proteins to restrict their timely localization to the NE and their interaction with the Y-complex.

Figures 1&2 show that UBAP2L localizes to the NPCs and interacts with Nups and NPC assembly factors. Figure 3 shows that UBAP2L regulates Nups localization but not in postmitotic cells. Figure 5 shows UBAP2L mediates the assembly of the NPC scaffold elements and the biogenesis of NPCs. Figure 10 summarizes the steps UBAP2L takes to regulate the biogenesis of NPCs at the intact nuclear envelope.

The recommendation is for acceptance.

Notes:

What measurements have been made to determine Co-localization? Needs to be clearly stated in methods and results.

Fig.5 UBAP2L mediates the assembly of the NPC scaffold elements and the biogenesis of NPCs.

Panel G is that a doublet of Nup153? Not seen in Fig.2 or Fig.6? MAb 414 was used? Is there any Nup214? Or other FG Nups?

Fig. S2. Localization of UBAP2L during cell cycle progression. Panels E & I, Gray bar has no {plus minus} SD bars?

Why?

Fig. S3. UBAP2L may inhibit formation of cytoplasmic annulate lamellae (AL) or AL like Nup assemblies. Isn't crystal clear.

Panel B Please provide more details, especially on co-localization and increased magnification panels (close-ups) like other figures.

Reviewer #3 (Comments to the Authors (Required)):

This is a well written paper describing roles for a protein called UBAP2L in allowing Y-complex components to become incorporated into NPCs during interphase. This manuscript builds upon an extensive body of work that the Sumara lab have produced in recent years outlining roles for FXR1 and FMRP in this pathway.

I thought that the quality of the biochemistry, microscopy and data presentation were high. The experimental approaches, including single molecule and super-resolution imaging, CRISPR editing and molecular interaction are state of the art and well performed. The manuscript was long (but easy to read) and the conclusions drawn were clear, concise and for the most part accurately represented the data presented. The quantification of data was clear, careful and robust. I'm supportive of publication.

The results show that in unstressed conditions, UBAP2L localises to the NE and that it can interact with Y-complex NUPs. In the absence of UBAP2L, Nups and components of the nuclear transport machinery are relocalised to cytoplasmic puncta and that these effects are seen in G1 nuclei. In UBAP2L ko cells, NPC components are disorganised and NUPs are relocalised to cytoplasmic punctae - I thought the now symmetric annulate lamellae described in the SI was really striking. Mechanistic information is gleaned through rescue experiments invoking an Arg-rich domain of UBAP2L in allowing NUP incorporation, and its operation (and interaction) with FXRP in this pathway. Functional information is gleaned by examining nucleocytoplasmic transport and the response to serum starvation, suggesting that incorporation of NUPs into the NE can be boosted by UBAP2L overexpression and that UBAP2L is necessary for long term proliferation of cancer cells. The manuscript was mostly observational, but the observations are clear and interesting.

For the majority of the points above, the data are strongly supportive. My only slight concern relates to the interpretation of the serum starvation results in Fig 9 and Fig S12 in that 'serum starvation' isn't the same as 'nutrient stress'. Can you examine what happens to UBAP2L-dependent NPC density in amino acid deprived media? Serum starvation removes the GF needed for cell cycle progression and can lead to R-point arrest. Given your data suggesting UBAP2L operates in early G1 to control NPC incorporation, are the cells just arrested in a cell cycle phase where UBAP2L isn't operating? Finally, is NPC incorporation further reduced in the serum starved UBAP2L-ko cells?

Minor comment:

1. The term post-mitotic can mean those cells that have exited the cell cycle. I think you're using this term to describe cells that are completing mitosis (e.g., Fig 4 legend). Would mitotic-exit be a better descriptor here?
2. Fig. 1F could do with an orientation bar to show readers where you're measuring from.
3. In the GFP-NUP85 overexpressing cells, levels of UBAP2L are increased (Fig 2A) - is this evidence of an increased need for this pathway under conditions of overexpression?
4. In 3G, 3H, S2C and S2E, the relocalisation of mAb414 label from NE to cytoplasmic puncta is dramatic and perhaps a little undersold from the quantification. Not essential, but is there a way to represent the percentage of mAb414 signal in puncta, rather than just whether there are punctae?

* Point added from the consultative review - I think that R1 and R2 were fair. I agree with R1 about GFP-NUP85 overexpression and the need to tone down the 'UBAP2L-drives the formation of the Y-complex' messages.

Response to Reviewers Comments on manuscript JCB #202310006

Reviewer #1

Uncovering mechanisms governing nuclear pore complex (NPC) assembly has been a central pursuit of the nuclear transport field over several decades. Here, Liao et al. implicate UBAP2L as a potential factor that contributes to *de novo* NPC biogenesis in interphase. The key data supporting such a role includes super-resolution microscopy demonstrating the localization of UBAP2L at least a subset of NPCs, IP experiments demonstrating biochemical interactions with nups and loss of function experiments that lead to a reduction in NPC density at the nuclear envelope alongside the accumulation of nups in cytosolic aggregates. As IP experiments in the context of UBAP2L deletion result in a reduction of Y-complex components affinity purified with binding partners, there is an effort to establish that UBAP2L plays a role in "driving" the assembly of the Y-complex in the cytoplasm. Similar efforts try to connect UBAP2L to FXR1, another factor this group has suggested plays a role in NPC assembly. Although in general the data presented are of high quality and rigorously quantified, the data supporting the latter two conclusions is open to interpretation due to a lack of direct biochemical evidence supporting any physical interactions. Further, the manuscript is a bit challenging to navigate as it is quite long, there are many supplemental figures, and there is a tendency to veer off topic at the end of the work. Nonetheless, the core conclusion that UBAP2L plays a role in NPC assembly and/or stability is on solid footing and will make an important contribution appropriate for JCB.

We thank this reviewer for a very positive assessment of our manuscript and his/her appreciation of the importance of the presented findings and the quality of the experimental work. We are also thankful for helpful suggestions which we addressed in full and which greatly improved the study. We agree that direct biochemical evidence is missing in order to strongly support the role of UBAP2L in Y-complex assembly. We therefore strongly toned down the conclusions on scaffold assembly throughout the text and modified the title of the manuscript. We also thank this reviewer for a suggestion to consider a possible role of UBAP2L in repair and/or stability of NPC. As outlined below in detail (Major point 1), we added new data suggesting that indeed UBAP2L may play a more general role in NPC homeostasis, facilitating both *de novo* assembly as well as stability of existing NPCs at the intact nuclear envelope. All modifications following these recommendations are described below and the changes in the manuscript text are labeled in red.

Major Points

1) There have been many genetic perturbations described in the literature where nups are mislocalized but the key challenge is to define whether mislocalized nups are a product of defective assembly or NPC instability. There is no definitive evidence presented in this work that UBAP2L specifically impacts NPC assembly or that the observed cytosolic aggregates are an intermediate in NPC assembly. In fact, as the cycloheximide experiment presented in Figure 9 results in the persistence of aggregates, it suggests that they may be derived from assembled NPCs. This experiment should be performed in the context of the siRNA mediated knockdown of UBAP2L under unstressed conditions. In lieu of more definitive data establishing a role for UBAP2L in *de novo* NPC assembly, the title and conclusions should be amended to reflect this uncertainty as to UBAP2L's role.

We thank this reviewer for his/her suggestions and agree with this point. We included entire new Figure 10 and a results chapter (line 475) to present additional data supporting a hypothesis that UBAP2L might be involved in both NPC assembly and stability/repair as predicted by reviewer.

As suggested, we used siRNA-mediated downregulation of UBAP2L in the absence or presence of CHX and showed that downregulation of UBAP2L led to significant increase in the Nups granules in the cytoplasm and reduction of Nup intensity at the NE as observed in the KO cells (Fig. 10, A-D). CHX partially decreased the presence of Nup foci in UBAP2L-depleted cells but not in control cells (Fig. 10, A and C) as well as moderately decreased the NE Nups levels in both groups (Fig. 10, A and D), suggesting that UBAP2L-mediated regulation of Nups is partially dependent on the production of new Nups and that UBAP2L may also be involved in NPC stability and or NPC repair. To corroborate these observations, we generated stably integrated SNAP-Nup85 version in HeLa cells, which allowed for a pulse labelling of the “old” pool of existing Nup85 prior to siRNA transfections (Fig. 10G). As a positive control, we used downregulation of Nup153 previously implicated in *de novo* interphase NPC assembly. Relative to control siRNA, downregulation of UBAP2L or Nup153 led to increase in cytoplasmic Nups assemblies and decrease in Nups levels at the NE (Fig. 10, H-K). Although we cannot fully exclude the possibility that, even after extensive washes of the fluorescent SNAP, some labeling of new Nup85 pool took place during the course of this experiment, these results suggest that in addition to their role in the NPC assembly *de novo*, UBAP2L, and

unexpectedly Nup153, may also regulate stability or repair of existing NPCs at the NE during interphase.

To adequately integrate these additional observations and to correctly describe the role of UBAP2L in NPC homeostasis we adapted and modified the title and the entire manuscript whenever required.

2) The model that UBAP2L "drives the formation of Y complex" or that "UBAP2L mediates the interaction of Y complex Nups with Nup153" should be clearly written as being speculative as there is no direct data supporting these assertions. Instead, this conclusion rests on IP data in Figure 5G and H where overexpressed GFP-tagged nups are affinity purified with lower nup levels of other nups in the UBAP2L knockout. This is interpreted as suggesting that UBAP2L contributes to Y complex assembly. A more likely interpretation is that the GFP-tagged nups are mislocalized to the cytosolic aggregates where they may not robustly interact with nup partners. To be able to conclude that there is an active "driving" role for UBAP2L in forming nup subcomplexes, direct biochemical reconstitution experiments would be required that would be beyond the scope of the current study.

As stated above, we agree that direct biochemical reconstitution experiments are missing in order to strongly support the role of UBAP2L in Y-complex assembly. We highlighted the fact that in one set of our immunoprecipitation experiments, an endogenously tagged Nup96 has been used (Fig. 5H, line 260) but otherwise as suggested by reviewer we strongly toned down the conclusions on scaffold assembly throughout the entire text (i.e. replaced "drives" with "facilitate", "promotes", "may facilitate" etc) and modified the title of the manuscript removing the strong statement about the NPC scaffold assembly. We also clearly stated in the discussion part (line 538) that our proposed hypothesis on NPC scaffold assembly remains speculative at this stage of analysis.

3) General lack of scholarship. NPC assembly has been investigated for over 30 years in many models including yeast and *Xenopus* and mammalian cell lines. There is limited appreciation of this deep literature by the authors in particular in the introduction where there is an emphasis on their prior work. The strong suggestion is to include at least a discussion of the other assembly factors that have been identified including nuclear transport receptors/Ran, membrane remodeling proteins, lipid biosynthetic factors and chaperones/AAA proteins like Torsin. Such an effort might also lead to a better integration of UBAP2L into the known NPC assembly

pathway and may provide a more critical lens to evaluate their proposed model. It might also mean more consideration of alternative models like, for example, that UBAP2L is engaging with NTRs through its RGG motif. As NTRs are well established NPC assembly factors and play roles as chaperones in modulating biomolecular condensation, this analysis may provide fruitful avenues for future work.

We thank reviewer for this suggestion and we apologize for not having included sufficient scientific background on NPC assembly pathways in the previous version of the manuscript due to space limitations. We included additional short paragraph in the introduction section (line 85) describing some known non-Nup factors involved in the NPC assembly. We have also included a brief discussion part that these factors may co-operate with UBAP2L on assembly of annulate lamellae (line 599) or to promote UBAP2L-dependent “chaperone-like” function on Nups and FXR1 (lines 637, 646).

Minor Points:

1) Figure 1: Whether or not UBA2L associates with all or just a subset of NPCs is important for interpreting its function in, for example, the nuclear transport experiments. Can the authors provide these data? And if not, the caveats for providing this information should be discussed.

We agree that this is an important point but due to technical limitations of our imaging instruments, we are currently unable to perform superresolution 3D imaging of the entire cell/nucleus in order to conclude whether UBAP2L can be localized to all or just a subset of NPCs. We briefly discussed this limitation in the results section (line 144). Nevertheless, with an attempt to address this point, we analyzed a pixel co-localization/correlation from immunofluorescence microscopy pictures using correlation measurement within CellProfiler as described in the methods section (line 970). This analysis showed that a portion of endogenous UBAP2L co-localized with the Nups detected by the monoclonal antibody mAb414 (Fig. 1B).

2) Figure 2: There is no sense of the complexity of the bound fractions of these IPs as only Western blots are used. Either this data (e.g. a coomassie stain) should be presented or a more explicit evaluation of non-specific interactions with abundant proteins would help.

We strongly agree with this point and thank the reviewer for this suggestion. The quantifications and statistical analysis were included as a result of previous revision in another journal and we would be very happy to remove them should this be required for this reviewer. For now, we have additionally provided western blots of abundant proteins commonly used as negative controls. The results show no interaction with UBAP2L pathway components (Fig. 2A-C).

3) Figure 3 and others - there is no investigation of any inner ring nups. As these are central to NPC assembly, testing the localization/binding of at least one would support a more comprehensive analysis.

As suggested by reviewer, we have analyzed the localization of the inner ring Nup205 (Fig. 3A, E). The new results show that similar to Nups from other NPC sub-complexes (line 178), Nup205 accumulated in cytoplasmic foci following inactivation of UBAP2L.

4) Figure 4: Can the authors interpret why the Aurora B staining is so different in the UBAP2L Kos for the reader?

We previously described a role of UBAP2L during mitotic exit (Guerber et al EMBO reports, 2023). However, unlike for PLK1, we did not observe major changes in the localization of Aurora B, although some minor spindle midzone changes could be occasionally detected. We therefore included a better representative picture of Aurora B at the midzone in Fig. 4A. Related to this point, we showed in Fig. S5A, B that inhibition of PLK1 was not able to reverse the Nup localization observed upon UBAP2L deletion (line 579).

5) Figure 5: That there might be differences in the rotational symmetry of NPCs in the UBAP2L KO is interesting but may not make much sense. As it is often difficult to be certain whether there is comprehensive labeling or consistent blinking of fluorophores with SMS approaches, some caveats to these data should be introduced. Ultimately, ultrastructural approaches would be needed to be definitive here.

We agree with this point. We discussed the limitations of possible insufficient labeling in both experimental sets and the need for future ultrastructural analysis in the revised results section (lines 247 and 548).

6) There are many FG-nups that are not recognized by mAb414. Instead of using "FG-nups" in the text the term "414-reactive nups" or something equivalent would be more accurate.

We apologize for this mistake. We modified "FG-Nups" to "mAb414-reactive Nups" in the entire manuscript text as suggested by reviewer.

Reviewer #2

The manuscript is well-suited for JCB.

The quality of work is excellent. Adds to the field and is a stimulating set of work.

I believe it is worthy of acceptance.

Title: UBAP2L drives scaffold assembly of nuclear pore complexes at the intact nuclear envelope

Authors claim to have uncovered a direct role of ubiquitin-associated protein 2-like (UBAP2L) in the biogenesis of adequately organized and functional NPCs at the intact nuclear envelope (NE) in human cells. UBAP2L-mediated biogenesis of NPCs ensures nuclear transport, adaptation to nutrient stress, and cellular proliferation.

UBAP2L localizes to the nuclear pores and drives the formation of the Y-complex, an essential scaffold component of the NPC, and its localization to the NE. UBAP2L localizes to the NPCs and interacts with Nups and NPC assembly factors. Taken together, the interaction of UBAP2L with Y-complex Nups, as well as with the nuclear and cytoplasmic NPC assembly factors, suggests a possible function of UBAP2L on Nups assembly and NPC biogenesis.

UBAP2L facilitates the interaction of the Y-complex with POM121 and Nup153, the critical upstream factors in a well-defined sequential order of Nups assembly onto NE during interphase. UBAP2L and BAP2L regulate the localization of the Nup transporting factor FXR1. UBAP2L regulates Nups localization but not in postmitotic cells.

The duality of UBAP2L is that it localizes to the NE, and NPCs drive the formation of the Y-complex. They can chaperon FXRP proteins to restrict their timely localization to the NE and their interaction with the Y-complex.

Figures 1&2 show that UBAP2L localizes to the NPCs and interacts with Nups and NPC assembly factors. Figure 3 shows that UBAP2L regulates Nups localization but not in postmitotic cells. Figure 5 shows UBAP2L mediates the assembly of the NPC scaffold elements

and the biogenesis of NPCs. Figure 10 summarizes the steps UBAP2L takes to regulate the biogenesis of NPCs at the intact nuclear envelope.

The recommendation is for acceptance.

We are extremely grateful to this reviewer for his/her enthusiastic support of our manuscript and recognition of the quality and importance of our findings. We are also thankful for helpful suggestions which we addressed in full and which further improved the study. All modifications following these recommendations are described below and the changes in the manuscript text are labeled in red.

Notes:

What measurements have been made to determine Co-localization? Needs to be clearly stated in methods and results.

Thank you for this question. As explained in response to the minor point 1 of reviewer 1, we analyzed a pixel co-localization/correlation on immunofluorescence (and superresolution, please see below) microscopy pictures using correlation measurement within CellProfiler as mentioned in the result section (line 138) and described in methods (line 970). This analysis showed that a portion of endogenous UBAP2L co-localized with the Nups detected by the monoclonal antibody mAb414 (Fig. 1B).

Fig.5 UBAP2L mediates the assembly of the NPC scaffold elements and the biogenesis of NPCs.

Panel G is that a doublet of Nup153? Not seen in Fig.2 or Fig.6? MAb 414 was used? Is there any Nup214? Or other FG Nups?

We thank reviewer for pointing this out. Indeed, we observed in our hands that a specific band corresponding to Nup153 migrates at around 250 kDa size and is recognized by both Nup153 and mAb414 antibodies. Although we cannot really explain this phenomenon, we suspect that the usage of the pre-cast NuPAGE™ 3-8% Tris-Acetate gradient Gels (Thermo Scientific, EA0378BOX, please see the methods section) could influence this atypical migration pattern of Nup153. To confirm the specificity of the recognized band we provide below the full scans of Western blot analysis of control-, Nup153- and Nup214 siRNA- treated HeLa cell extracts (Fig. 1 for reviewer 2), which should demonstrate that molecular weight markers are assigned

correctly in the corresponding figures throughout the entire manuscript. We marked a non-specific, faster migrating band in the Fig. 5G with an asterisk and we included a short note in the results section to describe this evident Nup153 size discrepancy (line 158).

Figure 1 for reviewer 2. Analysis of the specificity of the Nup153 and mAb414 antibodies. Cell lysates of HeLa cells treated with indicated siRNAs were analysed by Western blotting with indicated antibodies. Please note that a specific band of around 250 kDa size was recognized by Nup153 and mAb414 antibodies corresponding to Nup153, as indicated in the Figures 2A and B; 3G; 5G and H; 6A.

Fig. S2. Localization of UBAP2L during cell cycle progression. Panels E & I, Gray bar has no {plus minus} SD bars?

Why?

We apologize for this mistake. The experiments presented in the old Fig. S2 are now shown in Fig. S1I-P due to fusion of several supplementary figures to fit the requirements of the maximum of five at JCB. We have now included the missing error bars in the corresponding panels K and O as requested.

Fig. S3. UBAP2L may inhibit formation of cytoplasmic annulate lamellae (AL) or AL like Nup assemblies. Isn't crystal clear. Panel B Please provide more details, especially on co-localization and increased magnification panels (close-ups) like other figures.

Thank you for this suggestion. As requested, we provided increased magnification panels for co-localization of Flag-UBAP2L and mAb414 signal in the cytoplasm in the new Fig. S2B in addition to the co-localization quantification analysis in the new Fig. S2C performed as described above. In the new panel A, we have shown additional magnified pictures of the cytosolic pre-assembled NPC-like complexes observed in UBAP2L KO cells, which appear to represent AL-like structures. We discuss the possibility that UBAP2L may regulate AL-NPC assembly in the revised manuscript (lines 239, 596).

Reviewer #3

This is a well written paper describing roles for a protein called UBAP2L in allowing Y-complex components to become incorporated into NPCs during interphase. This manuscript builds upon an extensive body of work that the Sumara lab have produced in recent years outlining roles for FXR1 and FMRP in this pathway.

I thought that the quality of the biochemistry, microscopy and data presentation were high. The experimental approaches, including single molecule and super-resolution imaging, CRISPR editing and molecular interaction are state of the art and well performed. The manuscript was long (but easy to read) and the conclusions drawn were clear, concise and for the most part accurately represented the data presented. The quantification of data was clear, careful and robust. I'm supportive of publication.

The results show that in unstressed conditions, UBAP2L localises to the NE and that it can interact with Y-complex NUPs. In the absence of UBAP2L, Nups and components of the nuclear transport machinery are relocalised to cytoplasmic puncta and that these effects are seen in G1 nuclei. In UBAP2L ko cells, NPC components are disorganised and NUPs are relocalised to cytoplasmic punctae - I thought the now symmetric annulate lamellae described in the SI was really striking. Mechanistic information is gleaned through rescue experiments invoking an Arg-rich domain of UBAP2L in allowing NUP incorporation, and its operation (and interaction) with FXRP in this pathway. Functional information is gleaned by examining nucleocytoplasmic transport and the response to serum starvation, suggesting that incorporation of NUPs into the NE can be boosted by UBAP2L overexpression and that UBAP2L is necessary for long term proliferation of cancer cells. The manuscript was mostly observational, but the observations are clear and interesting.

We are extremely grateful to this reviewer for his/her strong support of our manuscript and recognition of the high quality our findings, methods used and clear writing. Likewise, we are thankful for helpful suggestions which we addressed in full and which further strongly improved the study. All modifications following these recommendations are described below and the changes in the manuscript text are labeled in red.

For the majority of the points above, the data are strongly supportive. My only slight concern relates to the interpretation of the serum starvation results in Fig 9 and Fig S12 in that 'serum starvation' isn't the same as 'nutrient stress'. Can you examine what happens to UBAP2L-dependent NPC density in amino acid deprived media?

We apologize for using the term nutrient stress inappropriately. We corrected this mistake accordingly whenever referred to serum starvation experiments (from line 449 on). In addition, we provided new data suggesting that both deprivation of serum (new Fig. 9, N and O) and amino acids (new Fig. 9, P and Q) could induce the formation of the cytoplasmic Nup foci and can be rescued by Flag-UBAP2L overexpression, suggesting UBAP2L-mediated NPC regulation may occur under more general nutrient stress conditions (line 465).

Serum starvation removes the GF needed for cell cycle progression and can lead to R-point arrest. Given your data suggesting UBAP2L operates in early G1 to control NPC incorporation, are the cells just arrested in a cell cycle phase where UBAP2L isn't operating? Finally, is NPC incorporation further reduced in the serum starved UBAP2L-ko cells?

Thank you very much for these suggestions. As suggested by the reviewer, we performed additional experiments to address these points. We observed that although serum starvation further potentiated inhibition of cell viability in UBAP2L-dependent manner (new Fig. S5, W and X), it did not lead to more severe Nups defects in UBAP2L KO cells (new Fig. 9, F and G). This suggests that additional pathways may contribute to UBAP2L-dependent cell survival but not to UBAP2L-mediated Nups regulation under serum poor conditions (line 456).

We agree that it was important to exclude the possibility that serum starvation induced cell cycle arrest where UBAP2L is not operational. Therefore, we analyzed the effects of serum deprivation on Nups localization in both early G1 as well as in phospho-Rb-positive cells (mid-late G1, S and G2 phases) as described also for non-treated WT and UBAP2L KO cells in Fig.

4, B, E and F. All cell cycle stages analysed displayed increased cytoplasmic Nup foci in response to serum deprivation (Fig. 9, H-J) similar to the results obtained in UBAP2L KO cells and despite reduced percentage of phospho-Rb-positive cells upon serum starvation (Fig. 9K) (line 456).

Minor comment:

1. The term post-mitotic can mean those cells that have exited the cell cycle. I think you're using this term to describe cells that are completing mitosis (e.g., Fig 4 legend). Would mitotic-exit be a better descriptor here?

We agree with the reviewer on this point and corrected the term post-mitotic when referring to the cells exiting mitosis (line 207, chapter title and Fig. 4 legend), as suggested by reviewer. We used the word “postmitotic” solely as the established term for one type of the NPC assembly pathway.

2. Fig. 1F could do with an orientation bar to show readers where you're measuring from.

Thank you for this useful suggestion. We included orientation bars in the quantification panels of the superresolution images showing UBAP2L localization to NPC and indicated the NPC center (central channel middle point) and NPC periphery in the radial distribution of localization from the top view (new Fig. 1E) as well as NPC center and cytoplasm and nucleus orientation in the averaged side view profiles (new Fig. 1F). The figure legend has been adapted accordingly.

3. In the GFP-NUP85 overexpressing cells, levels of UBAP2L are increased (Fig 2A) - is this evidence of an increased need for this pathway under conditions of overexpression?

This could be a very interesting point but after careful analysis (please see Figure 2 for reviewer 3 below), we did not observe any reproducible effects of GFP-Nup85 overexpression on UBAP2L protein levels. Accordingly, we replaced the UBAP2L blot in the Fig. 2A panel by a more representative picture. In the future, it will be important to further study the factors and the conditions that regulate UBAP2L expression or protein levels as shown for instance for the serum deprivation in Fig. 9A, D and E and that could contribute to the UBAP2L-mediated NPC regulation.

Figure 2 for reviewer 3. Protein levels of UBAP2L are not affected by Nup85 overexpression. Three independent replicates of the experiments showing cell lysates of HeLa cells expressing GFP or 3XGFP-Nup85 analysed by Western blotting with indicated antibodies.

4. In 3G, 3H, S2C and S2E, the relocalisation of mAb414 label from NE to cytoplasmic puncta is dramatic and perhaps a little undersold from the quantification. Not essential, but is there a way to represent the percentage of mAb414 signal in puncta, rather than just whether there are punctae?

We replaced the panels in the new Fig 3H and new Fig S1I, M by less dramatic and more representative images to better fit the corresponding quantifications (Fig. 3I and Fig. S1K, O). In this case, an automatic subcellular segmentation analysis is difficult and complicated by the fact that frequently and depending on the focal view of the 2D images, the Nups foci overlay the nuclear area (please see some examples in the Fig. 3A), although through careful microscopic analysis, these granules reside clearly in the cytosol.

* Point added from the consultative review - I think that R1 and R2 were fair. I agree with R1 about GFP-NUP85 overexpression and the need to tone down the 'UBAP2L-drives the formation of the Y-complex' messages.

Thank you. We agree with these points and addressed the suggestions of reviewer 1 and 2 in full.

March 2, 2024

RE: JCB Manuscript #202310006R

Dr. Izabela Sumara
IGBMC
1, rue Laurent Fries
ILLKIRCH 67400
France

Dear Dr. Sumara,

Thank you for submitting your revised manuscript entitled "UBAP2L ensures homeostasis of nuclear pore complexes at the intact nuclear envelope." We would be happy to publish your paper in JCB pending final revisions necessary to meet our formatting guidelines (see details below).

A. MANUSCRIPT ORGANIZATION AND FORMATTING:

1) Text limits: Character count for Articles is < 40,000, not including spaces. Count includes title page, abstract, introduction, results, discussion, and acknowledgments. Count does not include materials and methods, figure legends, references, tables, or supplemental legends.

2) Figure formatting: Articles may have up to 10 main text figures. Avoid pairing red and green for images and graphs to ensure legibility for color-blind readers. If red and green are paired for images, please ensure that the particular red and green hues used in micrographs are distinctive with any of the colorblind types. If not, please modify colors accordingly or provide separate images of the individual channels.

Scale bars must be present on all microscopy images, including inset magnifications. Please add scale bars to magnifications in Figures 7E/F, 8E, 9L/N, S2A/B, & S5F.

Molecular weight or nucleic acid size markers must be included on all gel electrophoresis. In order for readers to accurately assess the size of the proteins shown, the cropped blot images must extend to include a region containing at least one of the molecular weight size markers that were run on the gel. Please increase the size of the cropped images of b-actin blots in Figures 2B, 3G, 9D, 10B/E, S3J, & S5L/O and the p62 blot in Figure 9D to include the nearest marker. Also add size markers to the RNA agarose gel in Figure S4J.

3) Statistical analysis: Error bars on graphic representations of numerical data must be clearly described in the figure legend. The number of independent data points (n) represented in a graph must be indicated in the legend. Please, indicate whether 'n' refers to technical or biological replicates (i.e. number of analyzed cells, samples or animals, number of independent experiments). If independent experiments with multiple biological replicates have been performed, we recommend using distribution-reproducibility SuperPlots (please see Lord et al., JCB 2020) to better display the distribution of the entire dataset, and report statistics (such as means, error bars, and P values) that address the reproducibility of the findings.

Statistical methods should be explained in full in the materials and methods. For figures presenting pooled data the statistical measure should be defined in the figure legends. Please also be sure to indicate the statistical tests used in each of your experiments (both in the figure legend itself and in a separate methods section) as well as the parameters of the test (for example, if you ran a t-test, please indicate if it was one- or two-sided, etc.). Also, if you used parametric tests, please indicate if the data distribution was tested for normality (and if so, how). If not, you must state something to the effect that "Data distribution was assumed to be normal but this was not formally tested."

4) Materials and methods: Should be comprehensive and not simply reference a previous publication for details on how an experiment was performed. Please provide full descriptions (at least in brief) in the text for readers who may not have access to referenced manuscripts. The text should not refer to methods "...as previously described." Please also indicate the acquisition and quantification methods for immunoblotting/western blots.

5) For all cell lines, vectors, constructs/cDNAs, etc. - all genetic material: please include database / vendor ID (e.g., Addgene,

ATCC, etc.) or if unavailable, please briefly describe their basic genetic features, even if described in other published work or gifted to you by other investigators (and provide references where appropriate). Please be sure to provide the sequences for all of your oligos: primers, si/shRNA, RNAi, gRNAs, etc. in the materials and methods. You must also indicate in the methods the source, species, and catalog numbers/vendor identifiers (where appropriate) for all of your antibodies, including secondary. If antibodies are not commercial, please add a reference citation if possible. Please add either references or more information describing how antibodies were made at the IGBMC antibody facility.

6) Microscope image acquisition: The following information must be provided about the acquisition and processing of images:

- a. Make and model of microscope
- b. Type, magnification, and numerical aperture of the objective lenses
- c. Temperature
- d. Imaging medium
- e. Fluorochromes
- f. Camera make and model
- g. Acquisition software
- h. Any software used for image processing subsequent to data acquisition. Please include details and types of operations involved (e.g., type of deconvolution, 3D reconstitutions, surface or volume rendering, gamma adjustments, etc.).

7) References: There is no limit to the number of references cited in a manuscript. References should be cited parenthetically in the text by author and year of publication. Abbreviate the names of journals according to PubMed. JCB formatting does not allow for supplemental references, please remove these and add any non-duplicate references to the main reference list.

8) Supplemental materials: Articles generally may have up to 5 supplemental figures and 10 videos.

Please also note that tables, like figures, should be provided as individual, editable files. A summary of all supplemental material should appear at the end of the Materials and methods section. Please include one brief sentence per item.

9) eTOC summary: A ~40-50 word summary that describes the context and significance of the findings for a general readership should be included on the title page. The statement should be written in the present tense and refer to the work in the third person. It should begin with "First author name(s) et al..." to match our preferred style.

10) Conflict of interest statement: JCB requires inclusion of a statement in the acknowledgements regarding competing financial interests. If no competing financial interests exist, please include the following statement: "The authors declare no competing financial interests." If competing interests are declared, please follow your statement of these competing interests with the following statement: "The authors declare no further competing financial interests."

11) A separate author contribution section is required following the Acknowledgments in all research manuscripts. All authors should be mentioned and designated by their first and middle initials and full surnames. We encourage use of the CRediT nomenclature (<https://casrai.org/credit/>).

12) ORCID IDs: ORCID IDs are unique identifiers allowing researchers to create a record of their various scholarly contributions in a single place. Please note that ORCID IDs are required for all authors. At resubmission of your final files, please be sure to provide your ORCID ID and those of all co-authors.

13) JCB requires authors to submit Source Data used to generate figures containing gels and Western blots with all revised manuscripts. This Source Data consists of fully uncropped and unprocessed images for each gel/blot displayed in the main and supplemental figures. Since your paper includes cropped gel and/or blot images, please be sure to provide one Source Data file for each figure that contains gels and/or blots along with your revised manuscript files. File names for Source Data figures should be alphanumeric without any spaces or special characters (i.e., SourceDataF#, where F# refers to the associated main figure number or SourceDataFS# for those associated with Supplementary figures). The lanes of the gels/blots should be labeled as they are in the associated figure, the place where cropping was applied should be marked (with a box), and molecular weight/size standards should be labeled wherever possible. Source Data files will be directly linked to specific figures in the published article.

Please add a Source Data image for the RNA agarose gel in Figure S4J.

14) Journal of Cell Biology now requires a data availability statement for all research article submissions. These statements will be published in the article directly above the Acknowledgments. The statement should address all data underlying the research presented in the manuscript. Please visit the JCB instructions for authors for guidelines and examples of statements at (<https://rupress.org/jcb/pages/editorial-policies#data-availability-statement>).

B. FINAL FILES:

Thank you for your attention to these final processing requirements. Please revise and format the manuscript and upload materials within 7 days. If you need an extension for whatever reason, please let us know and we can work with you to determine a suitable revision period.

Thank you for this interesting contribution, we look forward to publishing your paper in Journal of Cell Biology.

Sincerely,

Michael Rout, PhD
Monitoring Editor
Journal of Cell Biology

Dan Simon, PhD
Scientific Editor
Journal of Cell Biology

Reviewer #1 (Comments to the Authors (Required)):

The authors did a thorough job addressing all of my concerns. I look forward to seeing this work in print.

Reviewer #3 (Comments to the Authors (Required)):

I think the authors revisions are important and well performed, the text changes to tone down significance are fine by me. I have no further concerns and happy to recommend publication.